# Mast cells link immune sensing to antigen-avoidance behaviour

Thomas Plum[1,14 ✉], Rebecca Binzberger[1,2,14], Robin Thiele[1,2], Fuwei Shang[1,3], Daniel Postrach[1,2], Candice Fung[4], Marina Fortea[4], Nathalie Stakenborg[5], Zheng Wang[5], Anke Tappe-Theodor[6], Tanja Poth[7], Duncan A. A. MacLaren[8], Guy Boeckxstaens[5], Rohini Kuner[6], Claudia Pitzer[9], Hannah Monyer[8], Cuiyan Xin[10], Joseph V. Bonventre[10], Satoshi Tanaka[11], David Voehringer[12], Pieter Vanden Berghe[4], Jessica Strid[13], Thorsten B. Feyerabend[1] & Hans-Reimer Rodewald[1 ✉]

The physiological functions of mast cells remain largely an enigma. In the context of barrier damage, mast cells are integrated in type 2 immunity and, together with immunoglobulin E (IgE), promote allergic diseases. Allergic symptoms may, however, facilitate expulsion of allergens, toxins and parasites and trigger future antigen avoidance[1–3]. Here, we show that antigen-specific avoidance behaviour in inbred mice[4,5] is critically dependent on mast cells; hence, we identify the immunological sensor cell linking antigen recognition to avoidance behaviour. Avoidance prevented antigen-driven adaptive, innate and mucosal immune activation and inflammation in the stomach and small intestine. Avoidance was IgE dependent, promoted by Th2 cytokines in the immunization phase and by IgE in the execution phase. Mucosal mast cells lining the stomach and small intestine rapidly sensed antigen ingestion. We interrogated potential signalling routes between mast cells and the brain using mutant mice, pharmacological inhibition, neural activity recordings and vagotomy. Inhibition of leukotriene synthesis impaired avoidance, but overall no single pathway interruption completely abrogated avoidance, indicating complex regulation. Collectively, the stage for antigen avoidance is set when adaptive immunity equips mast cells with IgE as a telltale of past immune responses. On subsequent antigen ingestion, mast cells signal termination of antigen intake. Prevention of immunopathology-causing, continuous and futile responses against per se innocuous antigens or of repeated ingestion of toxins through mast-cell-mediated antigen-avoidance behaviour may be an important arm of immunity.

Mast cells are haematopoietic cells residing in barrier tissues exposed to internal and external environments[6]. Mast cells are best known for their roles in immunoglobulin E (IgE)-mediated allergies, which affect up to 40% of the world's population[7]. Type 2 immune responses are mounted in the context of barrier disruption and entry of infectious agents, including parasites, or tissue-damaging or innocuous protein antigens, collectively termed allergens or toxins. Type 2 immune responses, notably involving interleukin (IL)-4, drive immunoglobulin class switch recombination to antigen-specific IgE, which is bound to the mast-cell-expressed high-affinity IgE receptor (FcεRI). On reexposure to the antigen, mast cells release mediators, including proteases, histamine, serotonin and leukotrienes, which can contribute to allergic pathology. The role of mast cells in IgE-mediated allergies is often viewed as a consequence of an overreactive response using immune pathways originally directed against parasites[8]. In contrast to this view, Margie Profet[1] proposed the hypothesis that acute allergic reactions, for instance, sneezing, coughing, vomiting and diarrhoea, may serve to rapidly expel toxins and allergens. Moreover, these symptoms would help with recognition of the source, enabling future allergen and toxin avoidance[1]. Although evidence for antigen-avoidance behaviour has been reported[4,5], and this concept[1] has been broadened[2,3], no 'immunology of avoidance' has entered the field. Moreover, the cellular and molecular underpinnings, notably the role of mast cells, in this behavioural adaptation are poorly understood.

[1]Division for Cellular Immunology, German Cancer Research Center, Heidelberg, Germany. [2]Faculty of Biosciences, Heidelberg University, Heidelberg, Germany. [3]Faculty of Medicine, Heidelberg University, Heidelberg, Germany. [4]Laboratory for Enteric NeuroScience Translational Research Center for Gastrointestinal Disorders, KU Leuven, Leuven, Belgium. [5]Laboratory for Intestinal Neuroimmune Interactions, Department of Chronic Diseases, Metabolism and Ageing, Translational Research Center for Gastrointestinal Disorders, KU Leuven, Leuven, Belgium. [6]Pharmacology Institute, Heidelberg University, Heidelberg, Germany. [7]Center for Model System and Comparative Pathology, Institute of Pathology, Heidelberg University Hospital, Heidelberg, Germany. [8]Department of Clinical Neurobiology of the Medical Faculty of Heidelberg University and German Cancer Research Center, Heidelberg, Germany. [9]Interdisciplinary Neurobehavioral Core, Heidelberg University, Heidelberg, Germany. [10]Division of Renal Medicine and Division of Engineering in Medicine, Department of Medicine, Brigham and Women's Hospital, Harvard Medical School, Boston, MA, USA. [11]Laboratory of Pharmacology, Division of Pathological Sciences, Kyoto Pharmaceutical University, Kyoto, Japan. [12]Department of Infection Biology, University Hospital Erlangen and Friedrich-Alexander University Erlangen-Nuremberg, Erlangen, Germany. [13]Department of Immunology and Inflammation, Imperial College London, London, UK. [14]These authors contributed equally: Thomas Plum, Rebecca Binzberger. ✉e-mail: t.plum@dkfz.de; hr.rodewald@dkfz.de

Here, we show that mast cells and IgE are key in promoting protein-avoidance behaviour. Mucosal mast cells lining the stomach and small intestine rapidly respond to ingested antigens. Mice harbouring mast cells but not mast-cell-deficient mice subsequently avoid antigen uptake when given a free choice of water with or without antigen under unperturbed behavioural conditions. Our findings indicate an important protective role for mast cells to signal avoidance behaviour, which, when heeded, prevents or reduces inflammation driven by repeated confrontations between the immune system and innocuous substances. The marked conservation of mast cells in animal evolution, even before the advent of IgE, indicates that immunity of avoidance may be a fundamental mode of immune defence.

## Mast cells are essential for antigen-avoidance behaviour

We adapted the drink avoidance test from Cara et al.[4] and made use of genetically mast-cell-deficient mice[9]. We induced a systemic immune response to a model protein antigen, ovalbumin (OVA), by immunizing wild-type BALB/c $Cpa3^{+/+}$ and mast-cell-deficient BALB/c $Cpa3^{Cre/+}$ mice[9] by intraperitoneal injection on days 0 and 14 with OVA in complex with aluminium hydroxide (alum) as an adjuvant (OVA-alum) (Fig. 1a). In BALB/c mice, which are T helper type 2 prone, this induces robust OVA-specific IgE antibody responses (Fig. 1f). Control animals received alum only. Beginning on day 20, mice were subjected to the avoidance test, which is based on the preference of mice for egg white water (8% sucrose plus 20% egg white as OVA source in water) over plain water.

As behavioural assays are sensitive to environmental influences and the social wellbeing of mice, we housed large cohorts of mice (13–16 mice per cage), composed of mixed experimental groups, in Intelli-Cages (Fig. 1b). This system allows for the uninterrupted assessment of unbiased natural behaviour (Methods). The cages were equipped with eight drinking bottles, of which four contained egg white water, and four contained plain water (Fig. 1b). Throughout the experiments, mice had the free choice of egg white water and normal water. Individual drinking preferences were continuously recorded. Non-immunized mice (alum), whether they had mast cells ($Cpa3^{+/+}$) or not ($Cpa3^{Cre/+}$), showed a strong preference for the egg white water during the 12 days of measurements (Fig. 1c–e). By contrast, between days 1 and 5, immunized $Cpa3^{+/+}$ mice began to avoid the egg white water and afterwards drank almost exclusively normal water. Hence, in support of previous data[4], $Cpa3^{+/+}$ mice choose egg white water when non-immunized but normal water when immunized. Avoiding mice showed normal locomotion (number of cage corner visits; not shown) and showed no signs of disease. In marked contrast to $Cpa3^{+/+}$ mice, mast-cell-deficient $Cpa3^{Cre/+}$ mice failed to avoid the egg white water even when immunized (Fig. 1c–e), demonstrating an exclusive function of mast cells, that is, one that cannot be compensated by other immune or non-immune cells, for antigen-avoidance behaviour.

In summary, non-immunized mice preferred egg white water over water, whereas immunized mice shunned egg white water, and, notably, this antigen-avoidance behaviour was mast cell dependent. In addition, we observed higher egg white water preference among $Cpa3^{Cre/+}$ alum versus $Cpa3^{Cre/+}$ OVA-alum mice, indicating that a minor component of the OVA-avoidance response may be immunization dependent and mast cell independent (Fig. 1d).

Mast cells have been linked to anxiety-like behaviour[10] that may interfere with the behavioural analysis of $Cpa3^{Cre/+}$ mutant mice. To control for such potential deficits, we assessed anxiety-related and general behaviour in the elevated plus maze (Extended Data Fig. 1a–d), open field test (Extended Data Fig. 1e–i), light–dark test (Extended Data Fig. 1j–m) and recorded behaviour for 24 h in a home cage monitoring system (Extended Data Fig. 1n–s). None of these assays distinguished mast-cell-deficient mice from their wild-type littermates, indicating

that the $Cpa3^{Cre/+}$ mice had no behavioural deficits measurable by these assays that could have confounded the drink avoidance experiments.

The absence of OVA avoidance in immunized mast-cell-deficient $Cpa3^{Cre/+}$ mice was not due to diminished OVA-specific IgE and IgG1 antibody titres compared with wild-type littermates (Fig. 1f,g). Development of aversion was associated with a pronounced accumulation of mast cells in the stomachs and small intestines of wild-type mice (Fig. 1h–j). The strongest increases were found for small intestine intraepithelial mast cells, whereas the increase in number of small intestine lamina propria mast cells was less pronounced (Fig. 1i,j). As expected, in $Cpa3^{Cre/+}$ mice, numbers of mast cells in these tissues were negligible (Fig. 1h–j).

In addition to the absence of mast cells, basophil numbers are reduced to about 40% of normal in $Cpa3^{Cre/+}$ mice[9]. To address the role of basophils, we analysed basophil-deficient $Mcpt8$-$Cre$ mice (Extended Data Fig. 2). Mice were offered two bottles, one with egg white water and one with plain water (two-bottle test). Avoidance of egg white water was unimpaired in $Mcpt8$-$Cre$ mice (Extended Data Fig. 2a,b), which remained basophil deficient after immunization (Extended Data Fig. 2c,d). Except for reduced numbers of stomach mast cells, all parameters (mast cells in small intestine, anti-OVA IgG1 and IgE) were indistinguishable between basophil-deficient and wild-type mice (Extended Data Fig. 2e–h). Hence, basophils are not involved in the avoidance behaviour.

## Role of IgE in antigen-avoidance behaviour

Mast cells can be activated not only by antigen and IgE through the high-affinity IgE receptor (FcεRI) but also by non-IgE stimuli[11]. We tested the role of IgE in antigen-avoidance behaviour. BALB/c wild-type ($Igh$-$7^{+/+}$) and IgE-deficient ($Igh$-$7^{-/-}$)[11] mice were immunized (Fig. 1a) with OVA-alum or alum alone and subjected to the avoidance assay (Fig. 1b). In contrast to alum-immunized $Igh$-$7^{+/+}$ and $Igh$-$7^{-/-}$ mice, OVA-alum-immunized $Igh$-$7^{+/+}$ mice avoided the egg white water. Notably, IgE-deficient $Igh$-$7^{-/-}$ mice showed no avoidance (Fig. 2a–c). As expected, $Igh$-$7^{-/-}$ mice failed to generate anti-OVA IgE but produced anti-OVA IgG1 (Fig. 2d,e). Hence, IgE and mast cells are both essential for antigen-avoidance behaviour.

To test whether IgE is sufficient in the absence of immunization, we injected monoclonal anti-OVA IgE into BALB/c mice. This led to partial sensitization of wild-type but not mast-cell-deficient mice (Fig. 2f,g). Four of seven wild-type mice showed variable degrees of avoidance (Fig. 2f,g). Of note, after transfer of IgE, mast cell levels in stomach and small intestine were not elevated (Fig. 2h,i for passive sensitization; Fig. 1h,i for active immunization). In line with the partial avoidance response (Fig. 2f,g), only half of the mice showed activation of stomach mast cells on OVA contact (Fig. 2j), whereas none of them showed activation of small-intestinal epithelial mast cells (Fig. 2k). Moreover, active immunization but not IgE transfer sensitized BALB/c mice for anaphylaxis (based on temperature drop) on intragastric OVA application (Fig. 2l). Hence, monoclonal IgE transfer can partially induce avoidance behaviour; however, immunization may be required to attain full mast cell activation and avoidance.

## Th2 cytokines promote antigen-avoidance behaviour

We analysed a Th1-biased strain with (C57BL/6 $Cpa3^{+/+}$) or without (C57BL/6 $Cpa3^{Cre/+}$)[9] mast cells with respect to its ability to mount antigen-avoidance behaviour responses. Owing to their elevated sensitivity to sucrose[12], even at only 1% (as opposed to 8% in the BALB/c experiments), all but one of the OVA-alum-immunized C57BL/6 $Cpa3^{+/+}$ mice preferred egg white water (Extended Data Fig. 3a,d). However, at lower sucrose concentration (0.25%), avoidance behaviour became evident in C57BL/6 $Cpa3^{+/+}$ mice (Extended Data Fig. 3b,e), as well as its mast cell dependency in the C57BL/6 $Cpa3^{Cre/+}$ mice (Extended

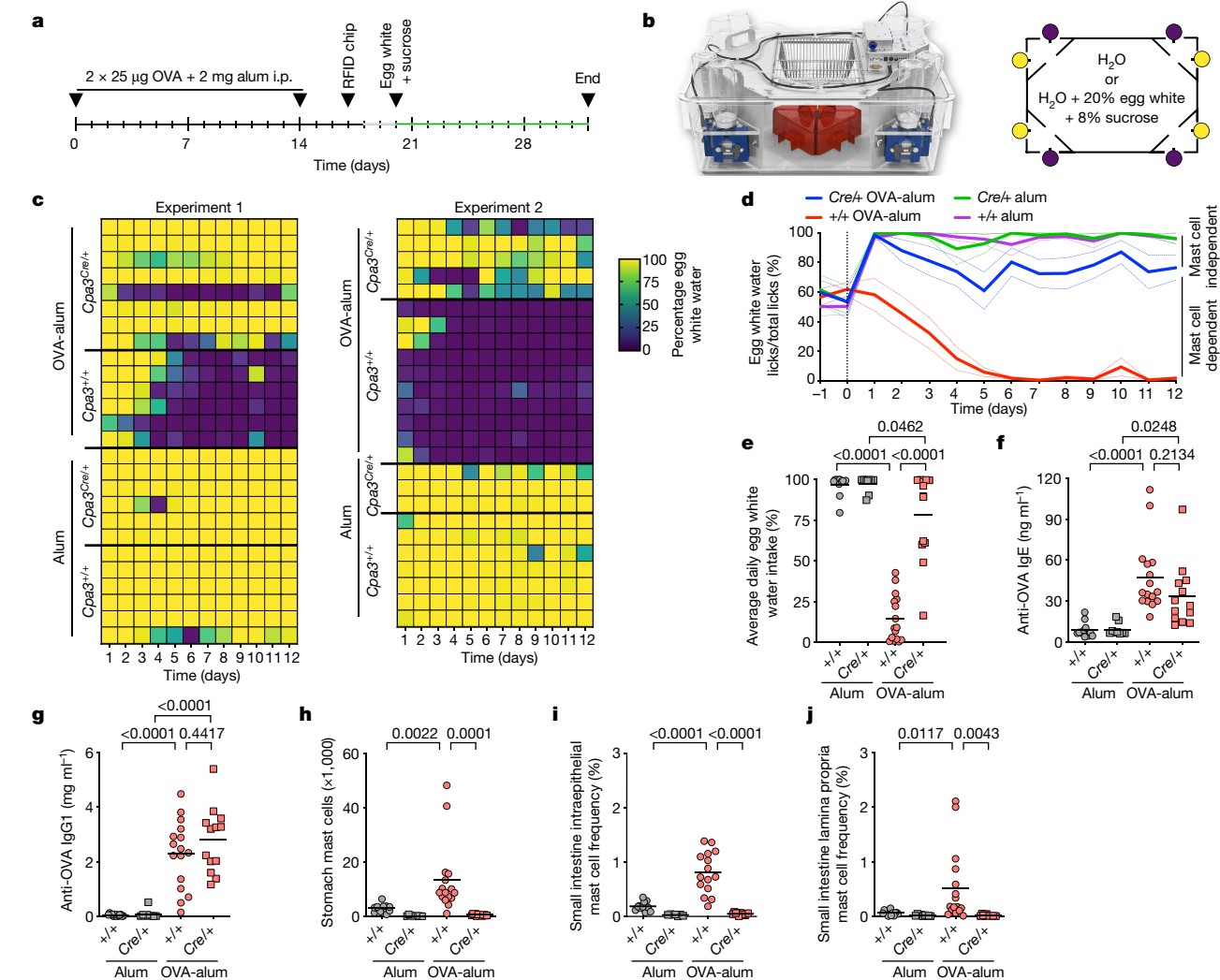

**Fig. 1 | Mast cells are essential for antigen-avoidance behaviour in type 2 immunized mice. a,b**, Type 2 immunization scheme, experimental timeline for avoidance test (**a**) and IntelliCage setup with bottle positioning (**b**). Purple symbols indicate water, yellow symbols indicate egg white water with sucrose. **c**, Egg white water preferences of alum-immunized and OVA-alum-immunized BALB/c *Cpa3^{+/+}* and *Cpa3^{Cre/+}* mice displayed as percentage of egg white water intake over total water intake (colour scale indicates percentages) during the course of experiments 1 and 2. Each row corresponds to an individual mouse. **d**, Egg white water preference displayed as number of egg white water licks over total number of water licks versus time. Data are presented as mean ± s.e.m. **e**, Egg white water preference displayed as the fraction of egg white water intake over total water intake as an average per day for the duration of the experiment. **f,g**, Serum amounts of anti-OVA IgE (**f**) and anti-OVA IgG1 (**g**) measured at the end of the IntelliCage experiments. **h–j**, Absolute numbers of stomach mast cells (**h**), frequencies of small intestine intraepithelial mast cells (**i**) and small intestine lamina propria mast cells (**j**) among total live cells, measured at the end of the IntelliCage experiments. Bars represent mean values, and each dot corresponds to a single mouse. In **c–j**, *Cpa3^{+/+}* alum (*n* = 13 mice for **c–h**, *n* = 12 for **i** and *n* = 13 for **j**); *Cpa3^{+/+}* OVA-alum (*n* = 16 mice for **c–h**, *n* = 15 for **i** and *n* = 16 for **j**); *Cpa3^{Cre/+}* alum (*n* = 9 mice for **c–h**, *n* = 7 for **i** and *n* = 9 for **j**); *Cpa3^{Cre/+}* OVA-alum (*n* = 13 mice for **c–h**, *n* = 11 for **i** and *n* = 13 for **j**). Statistical analysis was performed using one-way analysis of variance with Tukey multiple-comparison test (**e–j**). Exact *P* values are shown. i.p., intraperitoneal.

Data Fig. 3b,e). Antigen avoidance was not observed in C57BL/6 mice immunized with alum only (Extended Data Fig. 3a,b,d,e).

C57BL/6 mice were injected with a Th2-promoting and mast-cell-promoting cytokine cocktail 2 days after each round of immunization (Methods). This treatment enhanced the avoidance response; however, this response remained dependent on mast cells and OVA immunization (Extended Data Fig. 3c,f). Compared with that of BALB/c mice, the avoidance response of C57BL/6 mice remained less pronounced even under cytokine stimulation (Fig. 1c versus Extended Data Fig. 3c). Whereas OVA-specific IgE levels and stomach mast cell counts were comparable (Fig. 1f,h versus Extended Data Fig. 3g,h), levels of small intestine intraepithelial mast cells were approximately eight-fold higher in BALB/c mice (Fig. 1i) compared with cytokine-stimulated C57BL/6 mice (Extended Data Fig. 3i), indicating a mast cell compartment with

potential relevance for avoidance. Although mast cells in both stomach and small intestine were activated on oral OVA administration in immunized C57BL/6 mice (Extended Data Fig. 3j–l; measurement of activation is provided in the context of Fig. 4), systemic anaphylaxis required intravenous OVA injection (Extended Data Fig. 3m). In summary, Th2 cytokines promote mast-cell-dependent avoidance behaviour in OVA-immunized, Th1-prone C57BL/6 mice. However, in contrast to the BALB/c strain (Fig. 2l), OVA-immunized C57BL/6 mice were resistant to anaphylaxis by oral antigen challenge.

## Antigen avoidance prevents inflammation

We examined immunological consequences in the gastrointestinal tract, focusing on the stomach and small intestine, under avoidance

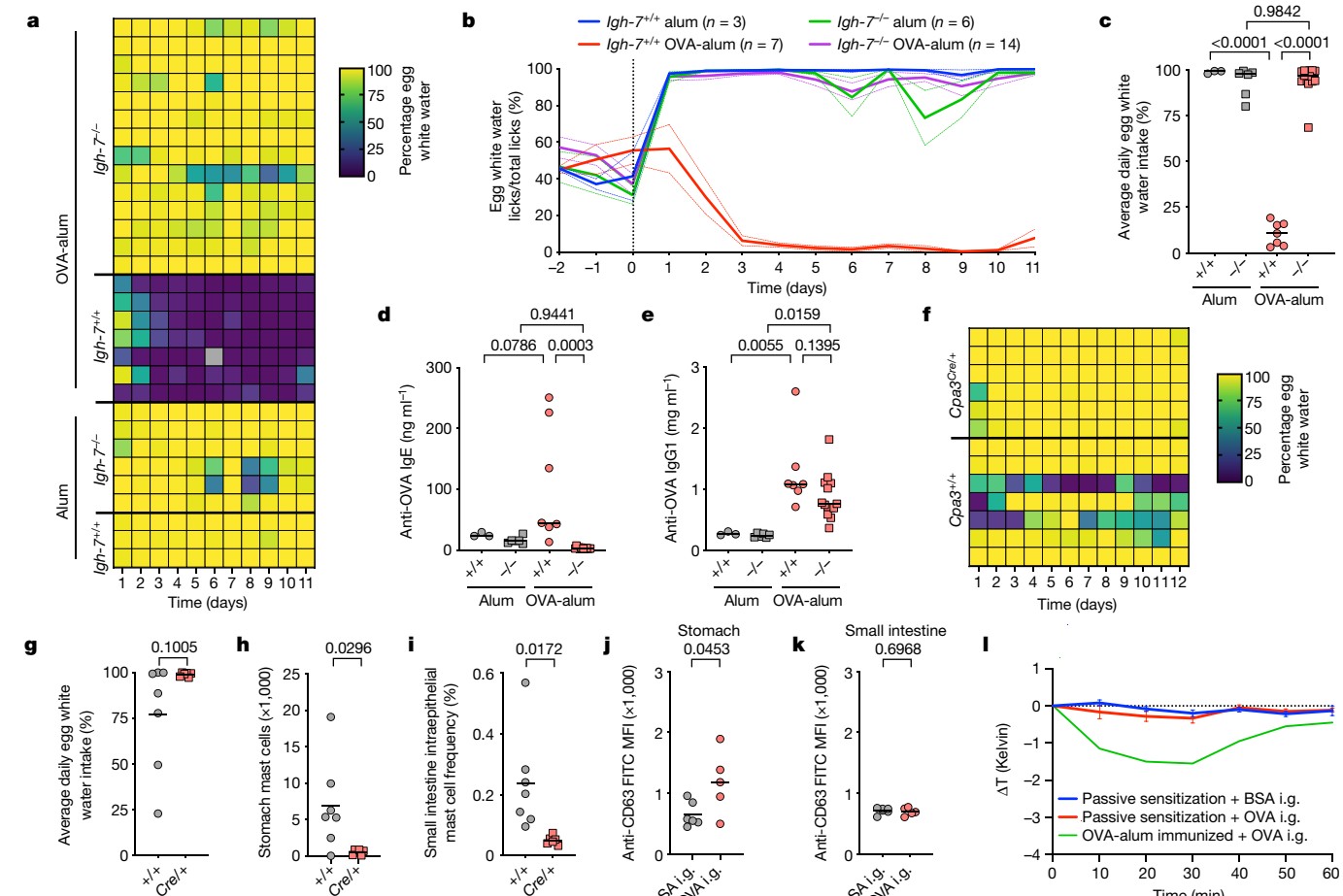

**Fig. 2 | Role of IgE in immunity of avoidance. a–c**, Egg white water preference of alum-immunized and OVA-alum-immunized BALB/c *Igh-7^+/+* and *Igh-7^−/−* mice displayed as percentage of egg white water intake over total water intake (colour scale indicates percentages) during the course of the IntelliCage experiment (**a**). Each row corresponds to an individual mouse. The grey field indicates a mouse that had lost the transponder for 1 day, precluding measurement. **b**, Data from **a** displayed as egg white water licks over total water licks versus time. Data are presented as mean ± s.e.m. **c**, Data from **a** displayed as the fraction of egg white water intake over total water intake as an average per day for the duration of the experiment. **d,e**, Serum amounts of anti-OVA IgE (**d**) and anti-OVA IgG1 (**e**) at the end of the experiment. **f,g**, BALB/c *Cpa3^+/+* and *Cpa3^Cre/+* mice received mouse anti-OVA monoclonal IgE antibody on days 0 and 9 and were subjected to the avoidance test (IntelliCage as in Fig. 1b) starting on day 10 (Methods). **f**, Egg white water preference is displayed as in **a**. Each row represents an individual mouse. **g**, Egg white water preference is displayed as in **c**. **h,i**, Absolute numbers of stomach mast cells (**h**) and frequency of small intestine intraepithelial mast cells (i) among total live cells at the end of the experiment. **j,k**, CD63 expression, an indicator of mast cell activation, on stomach (**j**) and small intestine intraepithelial (**k**) mast cells from IgE-sensitized BALB/c *Cpa3^+/+* mice after intragastric gavage (i.g.) with OVA or control protein (BSA). **l**, Rectal temperatures after indicated challenges in IgE-sensitized (passive sensitization) or OVA-immunized (OVA-alum) BALB/c mice. Data are presented as mean ± s.e.m. For **a–e**, *Igh-7^+/+* alum (*n* = 3 mice); *Igh-7^+/+* OVA-alum (*n* = 7); *Igh-7^−/−* alum (*n* = 6); *Igh-7^−/−* OVA-alum (*n* = 14). For **f–i**, *Cpa3^+/+* (*n* = 7); *Cpa3^Cre/+* (*n* = 6). For **j–l**, *Cpa3^+/+* IgE/BSA (*n* = 6); *Cpa3^+/+* IgE/OVA (*n* = 5); *Cpa3^+/+* OVA-alum/OVA (*n* = 2). Statistical analysis was performed using one-way analysis of variance with Tukey multiple-comparison test (**c–e**) and two-sided Student's *t* tests (**g–k**). Exact *P* values are shown. MFI, mean fluorescence intensity.

and non-avoidance conditions. To this end, wild-type BALB/c mice were immunized and subjected to either the two-bottle test ('avoidance') or OVA gavage ('non-avoidance') (Methods) for up to 16 days (Extended Data Fig. 4a). Mast-cell-deficient BALB/c mice were subjected to the same protocol. As expected, wild-type mice avoided whereas mast-cell-deficient mice preferred egg white water in the two-bottle test (Extended Data Fig. 4b). Wild-type mice forced to ingest OVA by gavage developed diarrhoea, whereas all but one of the mast-cell-deficient mice remained healthy after OVA gavage (Extended Data Fig. 4c), consistent with the role of mast cells in models of food allergy[13]. Occurrence of diarrhoea indicated induction of inflammatory and immunological processes under non-avoidance conditions. Indeed, the percentages of neutrophils in stomach and small intestine increased, and serum (hence systemic) levels of IL-4 and IL-6 were elevated (Extended Data Fig. 4d–g).

We analysed changes in gene expression by RNA sequencing (RNA-seq) in whole lysates from stomach and small intestine taken

on days 7, 11 and 16 after the beginning of the tests (Extended Data Fig. 4a). We reasoned that RNA-seq would yield comprehensive, high-resolution data on possible immune pathway activation and inflammation, comparing non-avoidance with avoidance conditions. Samples were subjected to principal component analysis (Extended Data Fig. 5a–d) based on the 500 most variable genes, corresponding to approximately 45% of variance. The results indicated the greatest differences between wild-type mice under avoidance (two-bottle test) and non-avoidance (OVA gavage) conditions (Extended Data Fig. 5a,b). This distinction was less obvious (in the small intestine) or absent (in the stomach) in mast-cell-deficient mice, which were non-avoiding in both the two-bottle test and under gavage (Extended Data Fig. 5c,d).

We characterized differential gene expression in the stomach and small intestine, comparing avoidance and non-avoidance conditions (Extended Data Fig. 5e,f). We first compared wild-type mice under avoidance (two-bottle test) and non-avoidance (gavage) conditions, to

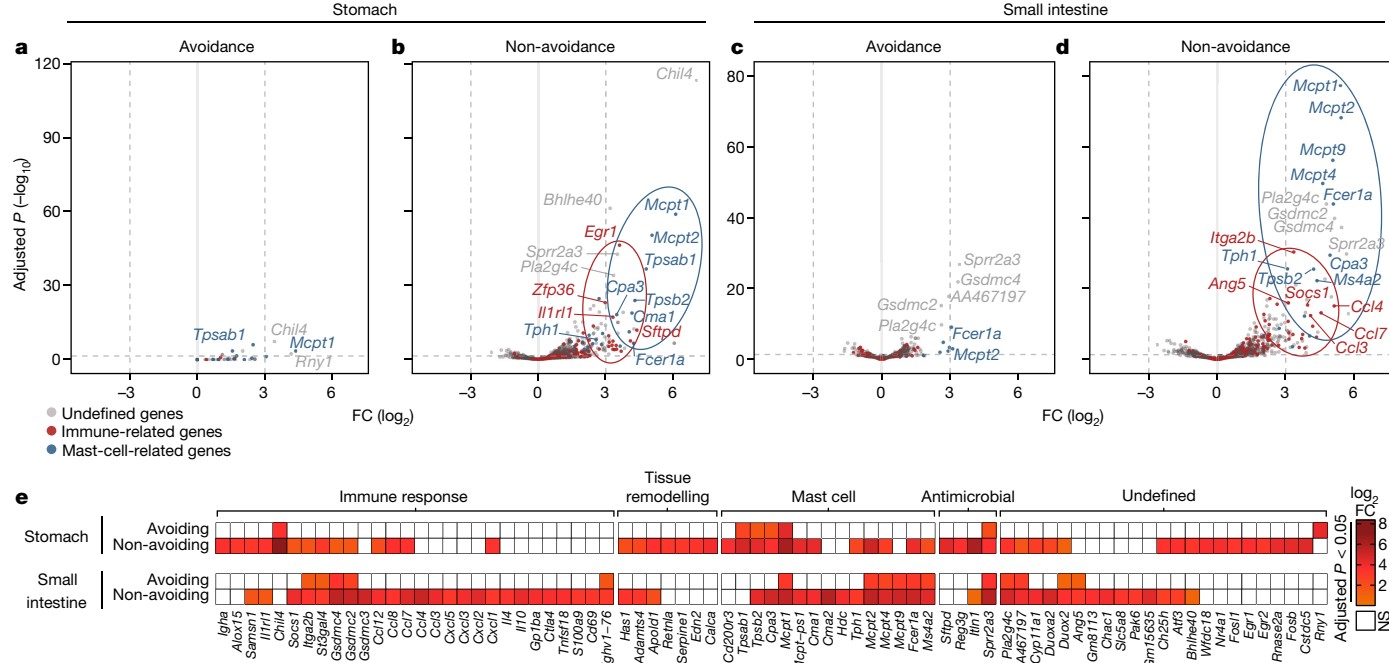

**Fig. 3 | Antigen avoidance prevents immune activation and inflammation in gastrointestinal tissues. a**–**d**, Volcano plots of DEG in stomach (**a**,**b**) and small intestine (**c**,**d**) tissues comparing avoiding mice (7-day two-bottle test; stomach $n = 4$, intestine $n = 3$) with mice receiving only water (stomach $n = 2$, intestine $n = 3$) (**a**,**c**) and non-avoiding mice (7-day gavage; stomach $n = 3$, intestine $n = 3$) with mice receiving only water (stomach $n = 2$, intestine $n = 3$) (**b**,**d**). The experimental outline is shown in Extended Data Fig. 4a. Red dots represent genes associated with immunity-related GO terms (Methods and Extended Data Fig. 5e, f). Blue dots represent manually annotated mast cell genes. **e**, Heatmaps of $\log_2$ fold change in gene expression in stomachs and small intestines of mice under avoidance and non-avoidance conditions on day 7. Genes shown are taken from **b** and **d**, satisfying $\log_2$ fold change >3 and $P \le 0.05$. Fold changes are indicated by scale: white indicates a gene not significantly regulated. All data are from wild-type BALB/c mice. Statistical comparisons were done using the DESeq2 package (Methods). $P$ values were adjusted for multiple comparisons using the Benjamini–Hochberg algorithm. FC, fold change; NS, not significant.

shed light on any gastrointestinal antigen-driven immunity and inflammation that is prevented by antigen-avoidance behaviour in mice with a normal immune system. For all time points and tissues, expression of genes indicative of adaptive, innate and mucosal immunity as well as chemotaxis was elevated under non-avoidance conditions (Extended Data Fig. 5e,f and Supplementary Tables 1 and 2).

To identify differentially expressed genes (DEG) characteristic of avoidance or non-avoidance, we analysed fold changes in gene expression versus $P$ values using volcano plots (Fig. 3a–d). In this manner, we compared avoiding mice (two-bottle test, where mice still drink small amounts of the OVA solution) with mice receiving only water (Fig. 3a,c). We also compared mice receiving gavage (non-avoidance) with mice drinking only water (Fig. 3b,d). Under non-avoidance conditions, DEG composed of mast-cell-related and immunity-related genes (for definitions, see Methods) emerged in the stomach and small intestine (Fig. 3b,d) (Supplementary Table 3). No transcriptional response of this scope and magnitude was observed in avoiding mice (Fig. 3a,c). Notably, even avoiding mice showed some gene activation compared with water-only mice, including activation of mast-cell-related genes (Fig. 3a,c). This gene activation, which only became evident by comparison of water-only versus avoiding mice, may define a threshold for the amount of voluntary antigen uptake in antigen-sensitized mice (Fig. 3a versus Fig. 3b and Fig. 3c versus Fig. 3d).

For visualization of tissue-specific signatures and annotation of genes in addition to those related to mast cells and immunity, we generated heatmaps of signature genes (Fig. 3e; for gene filtering, see Methods). We identified genes associated with tissue remodelling and antimicrobial functions, which, notably, showed differences between

the stomach and small intestine (Fig. 3e). Finally, we also probed our DEG versus a published inflammatory gene expression list[14]. In both stomach and small intestine, the inflammatory signature was evident under non-avoidance but not avoidance conditions in wild-type mice (Extended Data Fig. 5g,h and Supplementary Table 4). Collectively, our comparison of gastrointestinal gene expression at a tissue level between mice under avoidance and non-avoidance conditions showed broad immune activation and inflammation in stomach and small intestine, nearly all of which was prevented by antigen-avoidance behaviour.

## Immune activation in non-avoiding $Cpa3^{Cre/+}$ mice

Next, we asked how much of the non-avoidance induced immune-related and inflammation-related response was mast cell dependent. In analogy to the analysis in Extended Data Fig. 5e,f, we compared gene expression in BALB/c wild-type (avoiding) versus mast-cell-deficient (non-avoiding) mice in the two-bottle test (following the experimental outline in Extended Data Fig. 6a). In mast-cell-deficient mice, the induction of immune response genes was much reduced and its kinetics were altered (Extended Data Fig. 6b,c), compared with wild-type mice under non-avoidance conditions (Extended Data Fig. 5e,f). In the stomachs of mast-cell-deficient mice, enhanced immune gene expression was only detected on days 11 and 16 (Extended Data Fig. 6b). The intestines of mast-cell-deficient mice showed elevated immune gene expression only on day 7 (Extended Data Fig. 6c). Hence, the immunological and inflammatory response that is prevented by mast-cell-mediated antigen-avoidance behaviour is largely but not exclusively driven by mast cells.

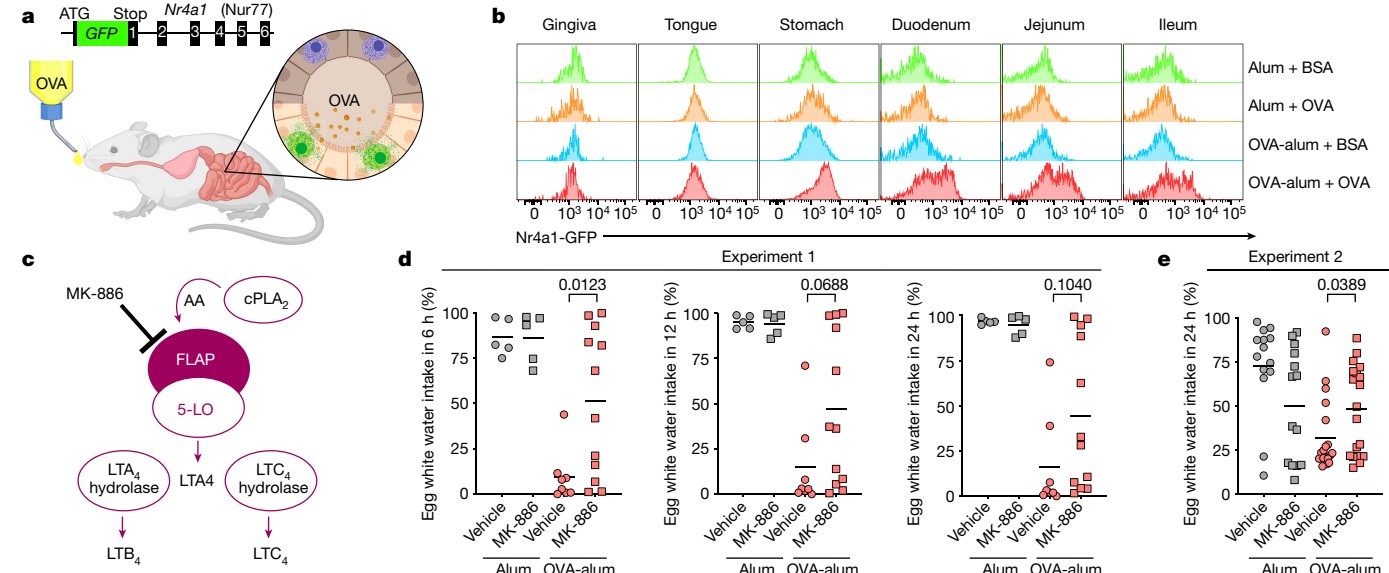

**Fig. 4 | In situ tracing of mast cell activation and role of FLAP in antigen avoidance. a**, Model for detection of FcεRI-activated mast cells using the *Nr4a1-GFP* reporter allele. In vivo OVA-responsive mast cells expressed GFP. **b**, (BALB/c x C57BL/6)F1 *Nr4a1-GFP* mice immunized with OVA-alum or alum only were given drinking water with OVA or BSA (Methods). Representative histograms (for data shown in Extended Data Fig. 7a) of GFP expression in tissue mast cells isolated ex vivo after drinking OVA- or BSA-containing water for 3 h. **c**, Model displaying the generation of leukotrienes and FLAP inhibition by MK-886. **d,e**, BALB/c wild-type mice were immunized with OVA-alum (as in Fig. 1a), and, 1 h before the IntelliCage (**d**), or two-bottle (**e**) test, mice were treated with PBS or MK-886 (Methods). Preference is displayed as the fraction of egg white water intake over total water intake for 6, 12 or 24 h. Bars represent mean values, and each dot corresponds to a single mouse. Alum vehicle (*n* = 5 mice), alum MK-886 (*n* = 5), OVA-alum vehicle (*n* = 8), OVA-alum MK-886 (*n* = 12) (**d**), alum vehicle (*n* = 14), alum MK-886 (*n* = 14), OVA-alum vehicle (*n* = 19), OVA-alum MK-886 (*n* = 19) (**e**). Statistical analysis was performed by two-sided Student's *t* tests in **d** and **e**. Exact *P* values are shown. AA, arachidonic acid. Figure 4a created with BioRender.com.

## Mast cell sensing of ingested antigens

To identify OVA-responsive mast cells along the route from the oral cavity to the gastrointestinal tract, we immunized *Nr4a1*-green fluorescent protein (*Nr4a1-GFP*) reporter mice. This enabled in vivo tracing of mast cells activated through FcεRI[15] (Fig. 4a). Three hours after OVA-immunized *Nr4a1-GFP* mice had drunk 25% OVA water, we isolated cell suspensions and analysed mast cells by flow cytometry for activation. Mast cells from the oral cavity (gingiva and tongue) remained GFP-negative, whereas stomach and small-intestinal mast cells showed pronounced signalling reporter expression (Fig. 4b and Extended Data Fig. 7a). We also analysed the oesophagus and colon but detected no mast cells (oesophagus) or only minute numbers (colon), precluding further analysis (not shown). In a direct demonstration of their antigen reactivity, freshly isolated stomach mast cells from immunized but not control mice (BALB/c) showed specific increases in intracellular Ca$^{2+}$ in response to OVA stimulation in vitro (Extended Data Fig. 7b,c). Collectively, these results show that oral antigen exposure rapidly activates tissue-resident stomach and small-intestinal mast cells.

## Role of leukotrienes in antigen-avoidance behaviour

After mucosal antigen contact, mast cells become activated to release de novo synthesized lipid mediators and preformed mediators including biogenic amines, proteoglycans and proteases from their granules[16]. Lipid mediators, such as leukotrienes and prostaglandins, can activate sensory neurons[17].

Leukotrienes are generated from phospholipids by cytosolic phospholipase A2 and 5-lipoxygenase (5-LO); the latter requires the 5-LO-activating protein (FLAP), which is thus essential for the generation of leukotrienes[18] (Fig. 4c). We sought to specifically test the contribution of leukotrienes to OVA avoidance by pharmacological inhibition of FLAP. To this end, BALB/c wild-type mice were alum or OVA-alum immunized and treated, 1 h before the avoidance tests, with FLAP inhibitor MK-886 (ref. 19). FLAP inhibition reduced avoidance over the course of 24 h, with *P* values reaching significance at 6 h (Fig. 4d, Experiment 1) or 24 h (Fig. 4e, Experiment 2) compared with vehicle-treated controls. This inhibitor effect was seen in around half of the mice. These data indicate that leukotrienes may be involved at least in initial antigen-avoidance behaviour.

Next, we tested the roles in antigen avoidance of preformed mast cell mediators, including carboxypeptidase A3 (Cpa3) using *Cpa3*$^{Y356L,E378A}$ mutant mice (Extended Data Fig. 8a), mast cell protease 5 (Mcpt5) using *Cpa3*$^{-/-}$ (double deficient for Cpa3 and Mcpt5) mice (Extended Data Fig. 8b), mast cell tryptase (Mcpt6) using *Mcpt6*$^{-/-}$ mice (Extended Data Fig. 8c) and histamine using histidine decarboxylase (Hdc) *Hdc*$^{-/-}$ mice (Extended Data Fig. 8d). Immunized mice of these strains developed OVA avoidance to the same extent as wild-type mice (Extended Data Fig. 8a–d), excluding functions for Cpa3, Mcpt5, Mcpt6 and histamine in OVA-avoidance behaviour.

One of the avoidance signature genes enriched in mice force fed with antigens is *Tph1* (Fig. 3b,d,e), the enzyme for serotonin (5-hydroxytryptamine; 5-HT) synthesis. A large fraction of stomach mast cells harboured 5-HT (Extended Data Fig. 8e,f). Mast-cell-derived 5-HT may signal through its ionotropic serotonin 3 receptor (5-hydroxytryptamine receptor 3; 5-HTR3), a key regulator of visceral malaise, nausea and emesis[20]. We tested the role of 5-HTR3 in antigen avoidance by treatment of mice with the specific inhibitor palonosetron[21]. Palonosetron did not significantly decrease antigen-avoidance behaviour (Extended Data Fig. 8g). In summary, of all the preformed or newly generated mast cell mediators that we tested, only leukotriene synthesis blockade inhibited antigen-avoidance behaviour.

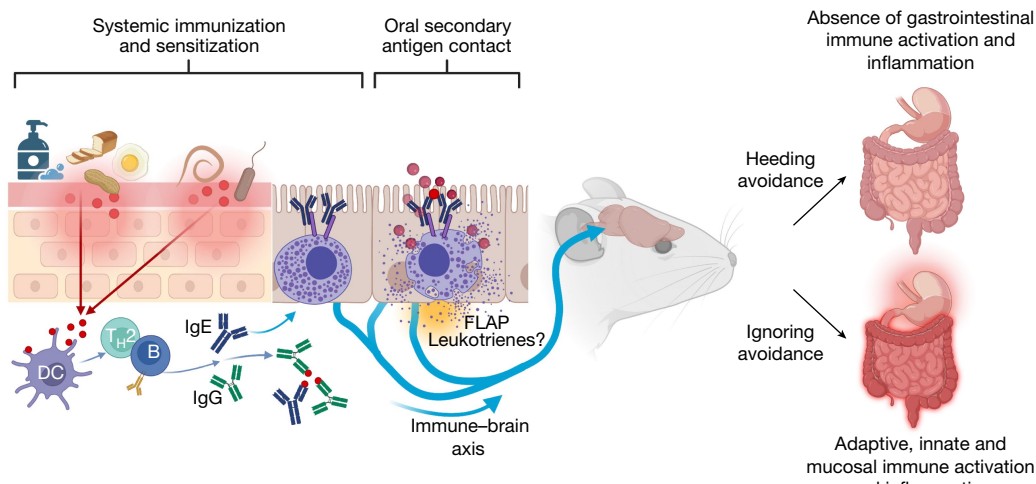

**Fig. 5 | Mast cells and IgE promote immunity of avoidance.** Model of mast-cell-mediated avoidance behaviour in the framework of type 2 immunity. Barrier damage facilitates entry of antigens (for instance, occupational and innocuous proteins such as those from flour, eggs or peanuts, as well as from pathogens), leading to a type 2 immune response. Adaptive immunity generates specific IgG and IgE antibodies towards antigen neutralization. Antigen-specific IgE binds to mast cells, which, on reencounter with antigens, signal avoidance behaviour. Inhibition of FLAP-dependent leukotrienes impairs avoidance, indicating that this mediator may contribute to the immune–brain axis. Mice avoiding antigen (heeding avoidance) are largely protected from developing gastrointestinal immune activation and inflammation, which occur when avoidance is ignored. Immunity of avoidance is dependent on the presence of mast cells and IgE (not shown). B, B cell; DC, dendritic cell. Created with BioRender.com.

## Interrogation of neural pathways

The fast avoidance reaction of some immunized mice (29% of wild-type mice within the first day and some even immediately after their first licks) (Extended Data Fig. 9a,b) could be due to rapidly acting humoral factors in the blood circulation or direct signalling from mast cells to neurons. The small-intestinal epithelium is innervated by intrinsic primary afferent neurons residing in submucosal and myenteric plexi of the enteric nervous system or by extrinsic neurons located in the dorsal root ganglia and vagal ganglia[22] (Extended Data Fig. 10a). We investigated possible mast cell signalling to intrinsic neurons by ex vivo calcium imaging of intestine segments from immunized mice expressing a genetically encoded calcium sensor in submucosal plexus (Extended Data Fig. 10b,c) and myenteric plexus (Extended Data Fig. 10d–f) neurons. Avoidance signalling through extrinsic vagal neurons was assessed by subdiaphragmatic vagotomy of immunized mice (Extended Data Fig. 10g,h). In addition, we tested the effect of resiniferatoxin (RTX)-mediated depletion of extrinsic Trpv1-expressing vagal and dorsal root ganglion sensory neurons on antigen avoidance (Extended Data Fig. 10i,j). None of these experiments revealed evidence for a neuronal route transmitting the avoidance signal (Extended Data Fig. 10b–j). As Trpv1-expressing (RTX-sensitive) neurons represent only a subset of all dorsal root ganglion neurons[23], a function for RTX-insensitive dorsal root ganglion neurons cannot be ruled out.

## Discussion

We show here that type 2 immunized animals sense antigens rapidly by mast cells lining the stomach and small intestines. Antigen-avoidance behaviour was found to be dependent on mast cells and IgE and was amplified by Th2 cytokines. Comparison of voluntary with no antigen uptake demonstrated a threshold for antigen consumption, resulting in apparently 'acceptable' low-level immune activation in the gastrointestinal tract. Ingestion of antigen above this threshold, which would normally be prevented by avoidance behaviour, resulted in strong gene activation indicative of adaptive, innate and mucosal immunity, as well as chemotaxis. Such broad and sustained (for at least 16 days) elevation of expression of immune-related and inflammation-related gene programmes indicates a profound immune response to OVA, akin to food allergy. The gastrointestinal epithelium forms an important barrier controlling nutrient uptake and preventing entry of microbes across the intestinal wall. Activated mast cells may induce intestinal leakiness by secretion of proteases Mcpt1 (ref. 24) and Mcpt4 (ref. 25), and indeed the *Mcpt1* and *Mcpt4* genes were strongly upregulated under non-avoidance conditions (Fig. 3e). The non-avoidance signature also contained genes associated with antimicrobial function, including *Sprr2a3*, *Itnr* and *Reg3g*, and tissue remodelling, including *Has1*, *Adamts4* and *Apold1*, which may indicate impaired barrier function. Hence, avoidance behaviour prevents antigen-driven immune activation, ensures barrier integrity and protects against allergic pathology[13,26,27]. The extent of immune gene induction was lower in non-avoiding mast-cell-deficient mice compared with force-fed wild-type mice, indicating that immune activation may be largely but not completely mast cell dependent.

The conservation of mast cells and IgE over millions of years in mammalian species strongly implies that these immune components provided a robust evolutionary advantage. Experimentally, when anaphylaxis was discovered in 1901 by Richet and Portier[28], it was interpreted as a fatal, 'non-protecting' response against the toxin with which dogs were repeatedly inoculated. The immunological 'purpose' of allergic responses has been a matter of long-lasting debate[1,2,8]. Whereas mast cells and IgE contribute to allergic pathology, there are few examples for protective Th2 responses through mast cells[29] and IgE[29,30]. Beneficial functions of mast cells and IgE are commonly invoked in parasite immunity[31]. However, parasite infections may be unperturbed[32] or only delayed in the absence of mast cells[33], and functions of mast cells and IgE can be redundant with those of other cells (for instance, eosinophils[34]) or antibody isotypes[35,36]. Given the complexities in understanding the actual roles of mast cells in immunological protection, the role for mast cells observed here in antigen-avoidance behaviour is intriguing. The non-redundant function of mast cells in the avoidance reaction may represent an evolutionary advantage of preventing recurrent immune responses to non-infectious antigens (allergens) and intoxications. Although allergens are frequently innocuous, non-toxic substances[2] (see, for example, the long list of occupational allergens[37]) and remain

non-immunogenic in the presence of intact barriers, they can elicit type 2 responses in the context of barrier defects in gut, lung or skin. Irritants such as detergents or solvents can damage the integrity of inner and outer surfaces[38], and antigen-specific IgE production can be enhanced in the wake of barrier defects[39]. Whether a substance is both barrier damaging and immunogenic, for instance, papain[40] or Derp1 (ref. 41), or whether a physical or chemical insult or an infection[26] damages barriers and opens the door to immunity against innocuous substances is irrelevant to the outcome: type 2 immunity, IgE production and loading of mast cells with antigen-specific IgE (Fig. 5). For both toxins and innocuous substances, avoidance prevents repeated contact, which, as we show here for a protein antigen, results in local and systemic immune activation and inflammation.

Owing to their location in barrier tissues and their close proximity to sensory neurons[42], mast cells are poised to sense protein antigens and signal their presence to the central nervous system. The compounds secreted by activated mast cells include leukotrienes[16]. Leukotrienes can activate sensory neurons, as has been shown for LTC4, which activates spinal neuron pathways through $CysLT_2R$ to induce itch[17]. It is thus conceivable that leukotrienes released from mast cells following antigen sensing signal avoidance within the first 24 h via dorsal root ganglion and spinal neuron pathways. Our analysis of antigen consumption kinetics in the IntelliCage shows that avoidance frequently develops on first oral antigen contact. A function for leukotrienes at this stage would be consistent with our experiments using an inhibitor of leukotriene production[43] (Fig. 4).

Aversive signals are processed in the brain by the parabrachial nucleus (PBN), which receives input from the spinal cord and brain stem[44]. Activation of neurons in the PBN is sufficient to drive conditioned taste avoidance after experimentally pairing a novel food (for instance, a sweetened solution) with injection of malaise-inducing cisplatin[45]. In this setting, cisplatin injection causes systemic release of growth and differentiation factor 15 (GDF-15) into the circulation[45], which signals avoidance of the paired food to the PBN through GFRAL-positive brain stem neurons[45–47]. Florsheim et al. propose that GDF-15 is induced downstream of mast cells and leukotrienes to promote antigen avoidance[43]. GDF-15 blockade effectively reduced avoidance during the second but not the first drink test. In this phase, systemic release of GDF-15 and signalling through brain stem neurons may sustain avoidance.

The sensing of environmental antigens by mast cells extends immunity beyond the neutralization and destruction of pathogens to the prevention of inflammatory disease by antigen avoidance. Despite their bone marrow origin, mast cells are only distantly related to other immune lineages[48,49]; it is thus not surprising that they have separate, unique functional properties. The present work identifies them as sensor cells linking antigen recognition elicited by type 2 immune responses to behaviour. In this manner, at long last[50], mast cells emerge with an important and non-redundant new function.

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

# Methods

## Mice

BALB/c $Cpa3^{Cre/+}$ (ref. 9), C57BL/6 $Cpa3^{Cre/+}$ (ref. 9), BALB/c $Mcpt8$-$Cre^{51}$, BALB/c $Cpa3^{-/-}$ (ref. 52), BALB/c $Cpa3^{Y356L,E378A}$ (ref. 53), BALB/c $Hdc^{-/-}$ (ref. 54), BALB/c $Mcpt6^{-/-}$ (ref. 55) and $Nr4a1$-$GFP^{56}$ mice on C57BL/6 or (BALB/c × C57BL/6)F1 background were maintained in the mouse breeding facilities of the DKFZ Heidelberg. BALB/c $Igh$-$7^{-/-}$ (ref. 11) mice were bred at Imperial College London and used at the Interdisciplinary Neurobehavioral Core, Heidelberg University, for experiments. BALB/c wild-type mice and C57BL/6 Wnt1|GCaMP3 ($Wnt1$-$Cre;R26R$-$GCaMP3$)[57,58] mice were maintained in the mouse breeding facilities of the KU Leuven. Mice were housed with a 12-h day–night cycle in a controlled environment at 20–24 °C and 45–65% humidity. All behavioural tests were performed on adult (>7 weeks old) female and male mice. Controls for $Cpa3^{Cre/+}$, $Mcpt8$-$Cre$, $Cpa3^{Y356L,E378A}$, $Cpa3^{-/-}$ and $Mcpt6^{-/-}$ mice were gender-matched and age-matched wild-type littermates. Controls for $Igh$-$7^{-/-}$ and $Hdc^{-/-}$ mice were gender-matched and age-matched wild-type mice housed in the same animal room. All animal experiments were performed in accordance with institutional and governmental regulations. Experiments in Heidelberg were approved by the Regierungspräsidium Karlsruhe, Germany. Experiments in Leuven were approved by the Animal Care and Animal Experiments Committee of the KU Leuven, Leuven, Belgium.

## Immunizations

Mice were actively immunized by intraperitoneal injection with 25 µg OVA (Sigma-Aldrich) complexed with 2 mg Al(OH)$_3$ (alum) (Invivo-Gen) on days 0 and 14, and avoidance tests started on day 20. For Th2 cytokine treatment, mice were injected intraperitoneally with a cocktail composed of 2 µg IL-3 (Peprotech), 12 µg anti-mouse IL-3 (MP2-8F8, BioLegend), 2 µg IL-4 (Peprotech), 12 µg anti-mouse IL-4 (11B11, BioLegend) and 0.4 µg IL-9 (Peprotech) on days 2 and 16. Mixing IL-3 with the anti-IL-3 antibody MP2-8F8 and IL-4 with the anti-IL-4 antibody 11B11 generates cytokine–antibody complexes that show increased activity in vivo[59], which we exploited here to increase the magnitude and duration of cytokine effects in vivo. No such effects have been described for IL-9; hence no anti-IL-9 antibody was used. For passive sensitization, mice were injected intraperitoneally on day 0 and intravenously on day 9 with 10 µg anti-OVA IgE monoclonal antibody (E-C1, Chondrex), and intragastric challenge (anaphylaxis) or avoidance test started on day 10.

## IntelliCage

Keeping mice in IntelliCages prevents stress induced by handling and thus enables observation of natural behaviour[60]. For systemic immunization, mice received two injections of OVA-alum on days 0 and 14 as described above. On day 17, mice received a subcutaneous implant of a unique radio-frequency identification (RFID) transponder into the nape under isoflurane anaesthesia and were placed in an IntelliCage apparatus (TSE Systems) in groups of 13–16 animals. Mice were kept on a 12-h light–dark cycle with ad libidum access to chow and water. After acclimatization (1–3 days), the water in four of eight bottles was exchanged for 20% (v/v) egg white water containing either 0.25% (w/v), 1% (w/v) (C57BL/6) or 8% (w/v) (BALB/c) sucrose. Drinking behaviour was analysed for up to 14 days. Data were collected using the IntelliCage Plus software (NewBehavior AG).

For passive sensitization, mice received two injections of monoclonal anti-OVA-IgE on days 0 and 9 as described above. On day 11, mice received a subcutaneous implant of a unique RFID transponder into the nape under isoflurane anaesthesia and were placed in the IntelliCage apparatus. Mice were kept on a 12-h light–dark cycle with ad libidum access to chow and water. After acclimatization, the water in four of eight bottles was exchanged for 20% (v/v) egg white water containing 8% (w/v) sucrose. Drinking behaviour was analysed for up to 14 days. Data were collected using the IntelliCage Plus software (NewBehavior AG).

## Egg white preparation

Egg white (20% v/v) was prepared as follows: chicken egg white was separated from its yolk, diluted in water and strained through filter paper (grade 3 hw; 65 g m$^{-2}$; Ahlstrom Munksjö). The 20% egg white solutions used for experiments contained on average 0.78 ± 0.17 endotoxin units per millilitre, which is substantially lower than that of standard mouse chow[61].

## Two-bottle test

One week after the second immunization, BALB/c $Cpa3^{+/+}$ and BALB/c $Cpa3^{Cre/+}$ or knockout mice and their corresponding littermates were individually housed. Cages were equipped with two identical bottles, one containing water and the other containing 20% egg white water with 8% sucrose. According to the results of sodium dodecyl sulfate polyacrylamide gel electrophoresis, the 20% egg white water solution contained approximately 40 mg ml$^{-1}$ OVA (not shown). Every 24 h, bottles were weighed, and the positions of the bottles were changed to control for a side preference in drinking behaviour. The avoidance tests were run for up to 7 days.

## Multiple OVA gavage treatments

Mice received 50 mg OVA by intragastric gavage every 2–3 days (for 7 days treatment, 4× gavage; for 11 days, 6× gavage; for 16 days, 8× gavage). Mice were inspected for diarrhoea 1 h after each gavage. By comparison to this OVA gavage, voluntary consumption by mast-cell-deficient BALB/c $Cpa3^{Cre/+}$ mice in IntelliCage experiments (Fig. 1) was on average 586 mg OVA per day. Hence, the amount of antigen given by gavage was not higher than the amount of antigen consumed voluntarily by mast-cell-deficient mice.

## Pharmacological inhibition of FLAP and 5-HTR3

Mice were immunized with OVA-alum as described above. On day 20, mice were deprived of drinking water overnight. Palonosetron (Sigma-Aldrich) was injected intraperitoneally at a dose of 0.5 mg kg$^{-1}$ 12 h before the avoidance test. MK-886 (Abcam) was administered by intragastric gavage at a dose of 10 mg kg$^{-1}$ 1 h before the avoidance test. As repeated dosing with MK-886 can alter mouse behaviour[62], we only gave the mice a single dose, followed by a 1-day observation. IntelliCage (MK-886) and two-bottle tests (MK-886; palonosetron) were run for 24 h.

## Analysis of mast cell activation in $Nr4a1$-$GFP$ mice

Mice were immunized with OVA-alum as described above. On day 20, $Nr4a1$-$GFP$ mice were housed individually. After a 12-h period of water deprivation, cages were equipped with a bottle containing 25% OVA (Sigma-Aldrich) in water. Control mice were given a bottle containing 25% bovine serum albumin (BSA, Roth) in water. After 3 h, the bottles were weighed to check for consumption from the bottles, and mice were euthanized for further analysis. Activation of purified mast cells was monitored by flow cytometry for GFP expression.

For analysis of anaphylaxis in $Nr4a1$-$GFP$ mice, mice were immunized with OVA-alum as described above. On day 21, mice received intragastric gavage with 50 mg OVA (Sigma) or 50 mg BSA (Sigma), and their body temperature was monitored using a rectal thermometer. After 3 h, mice were euthanized for further analysis. Activation of purified mast cells was monitored as described above.

## Pharmacological ablation of Trpv1-positive sensory neurons

Mice were immunized with OVA-alum as described above. Between the OVA immunizations, starting on day 10, resiniferatoxin (Cayman Chemicals) was injected subcutaneously into the flank in three escalating doses (30, 70 and 100 µg kg$^{-1}$) on consecutive days. Control mice were treated with vehicle (dimethyl sulfoxide in phosphate-buffered

saline (PBS)). On day 20, Trpv1 denervation was confirmed by prolonged withdrawal latency of mice to noxious heat (52 °C) applied to the tail (tail flick test; data not shown). Beginning on the next day, OVA avoidance was analysed by the two-bottle test.

## Tissue digestion

Gingival single-cell suspensions were prepared as previously described[63]. In brief, the palate and mandible were isolated, and tissues were digested for 1 h at 37 °C in RPMI supplemented with 10% fetal calf serum (FCS; Sigma-Aldrich), 0.15 µg DNase I and 3.2 mg ml$^{-1}$ collagenase IV (all enzymes from Sigma-Aldrich). Then, 0.5 M EDTA (Roth) was added during the last 5 min, and supernatant was filtered through a 70-µm cell strainer (ThermoFisher). Undigested gingiva tissue was peeled from the palate and mandible and mashed through the same filter to yield the gingiva cell suspension.

Tongue single-cell suspensions were prepared by finely mincing the tongue and digesting the tissue for three rounds of 15 min at 37 °C in RPMI supplemented with 0.1 mg ml$^{-1}$ Liberase (Sigma-Aldrich) and 2.5 µg ml$^{-1}$ DNase I (Sigma-Aldrich). After each round of digestion, the cell suspensions were filtered through a 70-µm cell strainer (ThermoFisher), and new enzyme solution was added to the tissue. All fractions were combined to yield the tongue single-cell suspension.

For isolation of stomach intraepithelial leucocytes, the stomach was cut open and food remnants were removed. Stomachs were incubated for 15 min at 37 °C in HBSS supplemented with 20 mM EDTA (Roth) to release epithelial layers from the connective tissue. The cell suspension was applied to a spin column (ThermoFisher) packed with 100-µm zirconia beads (Roth). After centrifugation, the flowthrough was collected, yielding an intraepithelial cell suspension containing mucosal stomach mast cells.

For preparation of small intestine cell suspensions, small intestines were cut open and food remnants were removed. Intestines were incubated for 15 min at 37 °C in HBSS supplemented with 2% FCS (Sigma-Aldrich), 5 mM EDTA (Roth), 1 mM DTT (Merck) and 10 mM HEPES (Life Technologies) to release epithelial layers from the connective tissue[27]. The cells in the soluble fraction (containing intraepithelial mast cells) were filtered through a 70-µm cell strainer (ThermoFisher). The remaining intestine tissue was washed in PBS and transferred into RPMI supplemented with 2% FCS (Sigma-Aldrich), 20 mM HEPES (Life Technologies), 0.2 mg ml$^{-1}$ collagenase IV (Sigma-Aldrich), 0.5 mg ml$^{-1}$ hyaluronidase I (Sigma-Aldrich) and 0.1 mg ml$^{-1}$ DNase I (Sigma-Aldrich). Digestion was carried out for 30 min at 37 °C, and digested tissue was filtered through a 100-µm cell strainer (ThermoFisher), yielding the lamina propria fraction (containing lamina propria mast cells).

Blood was drawn by cardiac puncture, followed by red blood cell lysis according to the manufacturer's protocol (RBC Lysis Buffer, BioLegend).

## Flow cytometry

Single-cell suspensions were centrifuged and incubated for 15 min with 200 µg ml$^{-1}$ mouse IgG (Jackson ImmunoResearch Laboratories) to block Fcγ receptors. After washing with PBS supplemented with 5% FCS (Sigma-Aldrich), cells were stained with fluorochrome-coupled antibodies (see list in the Antibodies section) for 20 min on ice and protected from light. Cells were washed and incubated with 100 nM SytoxBlue (Life Technologies) for dead cell exclusion. For absolute quantitation of cells, a defined number of 123 count eBeads (Life Technologies) were added to the samples before analysis with a BD LSRFortessa (Becton Dickinson). Data were analysed using FlowJo software (Treestar), using the gating strategies shown in Supplementary Fig. 1. Mast cells in the tongue and gingiva were gated as live CD45$^+$MHCII$^-$CD11b$^-$CD117$^+$FcεRI$^+$ cells. Stomach mast cells were gated as live CD45$^+$CD117$^+$FcεRI$^+$/IgE$^+$ cells. Intestinal mast cells were gated as live CD45$^+$CD3$^-$CD11b$^-$CD19$^-$Gr-1$^-$Ter119$^-$CD117$^+$FcεRI$^+$ cells. Neutrophils were gated as live CD45$^+$CD11b$^+$Siglec-F$^-$ Gr-1$^+$/Ly6G$^+$ cells. Basophils were identified as live CD45$^+$CD90.2$^-$CD11c$^-$Gr-1$^-$Siglec-F$^-$MHCII$^-$B220$^-$CD49b$^+$IgE$^+$ cells. For reagents, see list in the Antibodies section.

## Intracellular Ca$^{2+}$ measurement of mast cells

For analysis of Ca$^{2+}$ flux in stomach mast cells, stomach intraepithelial leucocytes were spun down and resuspended in calcium imaging buffer (125 mM NaCl, 3 mM KCl, 2.5 mM CaCl$_2$, 0.6 mM MgCl$_2$, 10 mM HEPES, 20 mM glucose, 1.2 mM NaHCO$_3$ and 20 mM sucrose, brought to pH 7.4 with NaOH) supplemented with 0.1% BSA (Roth), 2.5 mM probenecid (Biotinum), 0.01% Pluronic-F127 (Sigma-Aldrich) and 200 µg ml$^{-1}$ mouse IgG (Jackson ImmunoResearch Laboratories) to block Fcγ receptors. After washing with calcium imaging buffer, cells were stained with 4 µM Fluo-4 (Thermo Fisher Scientific) and CD45 BV421, CD117 PE and FcεRI APC antibodies for 30 min at room temperature in the dark. Cells were then washed with calcium imaging buffer supplemented with 0.1% BSA (Roth) and 100 nM SytoxBlue (Life Technologies). Cells were kept at 37 °C during measurements and analysed on a BD LSRFortessa (Becton Dickinson). After 30 s of baseline measurements, OVA (Sigma-Aldrich) was added to a final concentration of 1.25 mg ml$^{-1}$, and Fluo-4 fluorescence was acquired for 90 s. As a positive control, ionomycin (Sigma-Aldrich) was added to a final concentration of 16.4 mmol ml$^{-1}$ for the last 30 s of the measurement. Data were analysed using FlowJo software (BD Bioscience).

## Intracellular serotonin staining

Stomach intraepithelial leucocytes were prepared as described above. Cells were incubated with ZombieFITC (1:500, BioLegend) for dead/live discrimination and 10 µg ml$^{-1}$ anti-CD16/32 antibodies (39, BioLegend) to block Fcγ receptors for 15 min at room temperature. After washing with PBS supplemented with 5% FCS (Sigma-Aldrich), cells were stained with CD45 BV785, CD117 APC and IgE BV421 (R35-72, BD Bioscience) for 20 min on ice and protected from light. After washing and centrifugation, cells were fixed and permeabilized using a FoxP3-intracellular staining kit (BioLegend) according to the manufacturer's instructions. Cells were stained with 0.11 µg ml$^{-1}$ anti-5-HT (5HT-H209, Dako) or isotype control antibodies for 30 min, washed in PBS and stained with anti-mouse-IgG1 (RMG1-1, BioLegend) antibodies for 30 min before analysis with a BD LSRFortessa (Becton Dickinson).

## Serological analysis

OVA-specific IgE and IgG1 were measured by enzyme-linked immunosorbent assay as previously described[64]. Anti-OVA IgG1 (L71, Biozol) and anti-OVA IgE (2C6, Invitrogen) were used as standards. Rat anti-mouse IgG1-HRP (1:2000, X56, BD Pharmingen) and rat anti-mouse IgE-HRP (1:2000, 23G3, SouthernBiotech) were used for detection. Serum samples were diluted 1:40000 (IgG1) and 1:10 (IgE). IL-4 and IL-6 were measured by LEGENDplex Mouse Th Cytokine (BioLegend) assay according to the manufacturer's instructions.

## Ca$^{2+}$ imaging of full-thickness small intestine preparations

For ex vivo Ca$^{2+}$ imaging, the ileum of immunized adult Wnt1|GCaMP3 mice or immunized and AAV9-transduced (pENN.AAV.CamKII. GCaMP6f.WPRE.SV40, Addgene) Cpa3$^{+/+}$ and Cpa3$^{Cre/+}$ mice was isolated. Tissues were opened along the mesenteric border and pinned flat in a Sylgard-lined dish containing Krebs solution (120.9 mM NaCl, 5.9 mM KCl, 1.2 mM MgCl$_2$, 1.2 mM NaH$_2$PO$_4$, 14.4 mM NaHCO$_3$, 11.5 mM glucose and 2.5 mM CaCl$_2$), bubbled with 95% O$_2$/5% CO$_2$ at room temperature. The luminal contents were cleared away with Krebs washes. Tissues were mounted over an inox ring and stabilized using a matched rubber O-ring[65]. Ring preparations were placed on a glass-bottomed dish and imaged using a ×20 objective on an inverted Zeiss Axiovert 200M microscope equipped with a monochromator (Poly V) and a cooled CCD camera (Imago QE) (TILL Photonics). Preparations were constantly superfused with carbogenated Krebs solution at room

temperature using a local gravity-fed ($\pm 1$ ml min$^{-1}$) perfusion pipette. Krebs only, BSA (1% in Krebs) and OVA (1% in Krebs) were sequentially applied for 5 min each on to the mucosal surface using a perfusion pipette positioned above the imaged myenteric or submucosal plexus. Custom-written routines in Igor Pro (Wavemetrics)[66] were used for analysis. Heatmaps display normalized fluorescence ($F_i/F_0$) traces, and each row shows the fluorescence signal of an individual neuron under the control Krebs condition, followed by BSA (1%) and OVA (1%) mucosal perfusion. The signal of each neuron was normalized to baseline fluorescence under the Krebs condition. Traces depicted in the heatmaps are sorted top-down by the maximum amplitude of the signals detected. For percentages of activated neurons, regions of interest were drawn over each GCaMP-expressing neuron to calculate the average Ca$^{2+}$ signal intensity normalized to the baseline (displayed as $F_i/F_0$). Background subtraction was performed on some recordings where changes in background intensity were apparent. Neurons were considered to be active if at least one neuronal Ca$^{2+}$ peak was detected during each 5 min recording period.

## Vagotomy
Vagotomy was performed as previously described[67]. In brief, both vagal trunks were transected below the diaphragm. To ensure transection of all small vagal branches, neural and connective tissue surrounding the oesophagus was removed. Pyloroplasty was performed to avoid gastric dilatation due to vagotomy. Control mice underwent a sham operation, in which vagal trunks were exposed but not cut and pyloroplasty was performed.

## RNA isolation from stomach and small intestine
Naive, immunized and challenged (days 5, 7 or 11 of the two-bottle test; or 4×, 6× or 8× OVA gavage) BALB/c *Cpa3*$^{+/+}$ and *Cpa3*$^{Cre/+}$ mice were euthanized by CO$_2$ asphyxiation. Stomach and small intestine pieces (4 cm proximal to the duodenum) were cut open, and food remnants, fat and Peyer's plaques were removed. Tissues were immediately frozen in liquid nitrogen and ground in a mortar with a pestle. RNA isolation was performed according to the instructions of the PureLink RNA Mini kit (Invitrogen). Total RNA quality was determined by the RNA integrity number provided by the BioAnalyzer system (RNA 6000 Pico Kit, Agilent). Isolated RNA was stored at −80 °C until use.

## RNA sequencing
Library preparation was performed with a TruSeq Stranded RNA Kit (Illumina) according to the manufacturer's instructions. After library preparation, indexed samples were pooled and diluted to 2 nM with 2% PhiX spike in. The multiplexed library was then paired-end sequenced using NextSeq 1000/2000 P2 Reagents (200 Cycles) on the NextSeq 1000/2000 platform (conditions: Read1/Read2: 111 Cycles; Index1: 8 Cycles; Index2: 8 Cycles). Data were mapped using STAR aligner (v.2.5.2b)[68], and reads were annotated using the FeatureCounts algorithm from the subread package (v.1.5.1)[69]. Both mapping and annotation were performed on Genome Reference Consortium Mouse Build 38 (GRCm38)[70]. Count data normalization and differential expression analysis were performed using DESeq2 (ref. 71), comparing immunized unchallenged mice with immunized mice of the same genotype challenged by either the two-bottle test or OVA gavage.

Principal component analysis was performed on read counts normalized by the variant stabilizing transformation included in the DESeq2 package, based on the top 500 most variable genes. For subsequent data analysis and visualization, shrinkage towards zero of log$_2$ fold changes (lfcshrink) was computed using the apeglm implementation[72]. *P* values were adjusted using the Benjamini−Hochberg algorithm, and results were accepted as significant if adjusted *P* < 0.05.

Gene set enrichment analysis (GSEA) for each experimental group was performed on the complete dataset ranked with lfcshrink using ClusterProfiler[73]. All gene ontology (GO) terms related to 'biological

processes' in the org.Mm.eg.db database[74] were considered. Significantly (Benjamini−Hochberg adjusted *P* < 0.05) enriched GO terms were accepted for further processing. Based on the individual descriptions, enriched GO terms were manually annotated into four immune-related subgroups: innate immunity, adaptive immunity, mucosal immunity and chemotaxis (annotation of individual GO terms can be found in Supplementary Table 1). From subgrouped GO terms, core enrichment genes (genes that contribute most to the enrichment results for a GO term; GSEA documentation on the Broad website) were extracted and filtered (log$_2$ fold change <−1 or log$_2$ fold change >1.5 in at least one comparison). Fold changes of these genes were compared between experimental groups (Fig. 3a,c and Extended Data Fig. 7b,c).

Genes contained in the 'hallmark inflammatory response' gene set were retrieved from the Molecular Signatures Database (hallmark gene set collection: M5932, MSigDB)[14], and their fold changes in wild-type mice under avoidance and non-avoidance conditions were compared.

In the volcano plots, horizontal lines indicate the significance threshold and vertical lines indicate the log$_2$ fold change thresholds of −3 and 3, respectively. GSEA core enrichment genes are indicated in red, and manually curated mast-cell-related genes are depicted in blue (Fig. 3c–f). DEG with log$_2$ fold change >3 in the stomach (Fig. 3c,d) and intestine (Fig. 3e,f) were filtered and displayed as a heatmap (Fig. 3g). In this heatmap, genes not significantly differentially expressed in other comparisons are depicted as white squares. Genes with significant (*P* < 0.05) changes in expression have coloured squares. The colour scale indicates log$_2$ fold change.

## Analysis of anxiety-like behaviour in *Cpa3*$^{Cre/+}$ mice
Anxiety-like behaviour was tested using an elevated plus maze, open field test and light−dark test. Mice are averse to brightly lit open areas. However, they have a natural drive to explore a perceived threatening stimulus. Low levels of anxiety lead to increased exploratory behaviour, whereas high levels of anxiety lead to less locomotion and more time spent in enclosed areas. Tests for anxiety-like behaviour were performed between 09:00 and 13:00. C57BL/6 *Cpa3*$^{+/+}$ and *Cpa3*$^{Cre/+}$ mice were brought into the behavioural room 30 min before behavioural testing began. All experimental setups were cleaned with soap and 70% ethanol at the end of the measurements.

The elevated plus maze consisted of an opaque-grey plastic apparatus with four arms (6 cm wide and 35 cm long), two open (illuminated with 100 lux) and two closed (20 lux), set in a cross from a neutral central intersection (6 × 6 cm) and elevated 70 cm above the floor. The mice were placed in the centre of the maze, and 5-min test sessions were digitally recorded and analysed with Sygnis Tracker software (Sygnis).

For the open field test, mice were placed in the centre of a bright open arena (40 cm wide, 40 cm long and 40 cm high; 290 lux), and their behaviour was monitored with a digital camera and ANY-maze video tracking system (Stoelting Co.) for 10 min.

The behavioural test box used for the light−dark test consisted of two compartments, a 29 × 21-cm (21 cm high; 300 lux illumination) lit compartment and a 15 × 21-cm (21 cm high; 10 lux illumination) dark compartment, connected by an opening at floor level allowing transition between the compartments. Mice were placed in the dark compartment, and their behaviour was monitored with a digital camera and Sygnis Tracker software for 10 min.

## Analysis of general behaviour of *Cpa3*$^{Cre/+}$ mice
The LABORAS home cage observation system (Metris B.V.) consists of an adapted home cage placed on a carbon fibre platform that automatically detects behaviour-specific vibration patterns produced by the animal[75]. The LABORAS software (v.2.6.) processes the vibrations into various validated behaviours (climbing, grooming, locomotion immobility) and tracking information (distance travelled and speed). These behavioural parameters are automatically calculated as time duration or frequency counts. C57BL/6 *Cpa3*$^{+/+}$ and *Cpa3*$^{Cre/+}$ mice were individually placed in

these calibrated cages under standard housing conditions with free access to food and water for a 24-h period. Mice were not habituated to the LABORAS cages before the experiments were started.

## Antibodies

The following antibodies were used in flow cytometry: B220 FITC 1:50 (BD Pharmingen, RA3-6B2), CD3 BV421 1:200 (17A2, BioLegend), CD3 FITC 1:50 (17A2, BD Pharmingen), CD3 PE-Cy-7 1:25 (145-2C11, BD Pharmingen), CD11b PerCP-Cy5.5 1:400 (M1/70, eBioscience), CD11b BV421 1:400 (M1/70, BioLegend), CD11b PE-Cy-7 1:400 (M1/70, eBioscience), CD11c BV421 1:100 (N418, BioLegend), CD16/32 unconjugated 10 µg ml$^{-1}$ (93, BioLegend), CD19 BV421 (6D5, BioLegend), CD19 APC 1:400 (1D3, BD Pharmingen), CD45 BV421 1:400 (30-F11, BioLegend), CD45 BV785 1:400 (30-F11, BioLegend), CD49b APC 1:100 (DX5, BD Pharmingen), CD90.2 APC-Cy7 1:400 (30-H12, BioLegend), CD117 PE 1:800 (2B8, eBioscience), CD117 APC 1:800 (2B8, BD Pharmingen), CD117 BV711 1:800 (2B8, BioLegend), FcεRI APC 1:200 (MAR-1, eBioscience), Gr-1 BV421 1:800 (RB6-8C5, BioLegend), Gr-1 BV605 1:200 (RB6-8C5, BioLegend), IgE PE 1:100 (RME1, BioLegend), IgE BV786 1:100 (RME-1, BD Pharmingen), IgE BV421 1:100 (RME-1, BD Pharmingen), Ly6G PerCP-Cy5.5 1:100 (1A8, BD Pharmingen), MHCII A700 1:100 (M5/114.15.2, eBioscience), Siglec-F BV421 1:100 (E50-2440, BD Pharmingen), Siglec-F PE 1:100 (E50-2440, BD Pharmingen), Ter119 BV421 1:200 (Ter119, BioLegend), 5-HT unconjugated 0.11 µg ml$^{-1}$ (5HT-H209, Dako) and mouse-IgG1 PE 1:100 (RMG1-1, BioLegend).

## Schematics

Schematics shown in Figs. 4a and 5 and Extended Data Fig. 1a,e,j were made in Adobe Illustrator (v.25.0.1) using BioRender with permission to publish. The photograph of the IntelliCage was supplied by TSE Systems with permission to publish (Fig. 1b).

## Reporting summary

Further information on research design is available in the Nature Portfolio Reporting Summary linked to this article.

## Data availability

RNA-seq data have been deposited in NCBI Gene Expression Omnibus (GEO) and are publicly accessible through GEO series accession number GSE225054. The custom RNA-seq analysis pipeline is publicly available at https://github.com/robinthiele/AAPIA. Source data are provided with this paper.

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

**Acknowledgements** We thank E. Florsheim and R. Medzhitov for generously sharing and discussing data before publication, and K. Rajewsky, A. Roers, T. Höfer and B. Rodewald for discussions and comments on the manuscript. We also thank T. Holland-Letz, V. Weru and A. Kopp-Schneider for advice on statistical analyses, G. Küblbeck and S. Schäfer for excellent technical assistance and J. Hallgren for making *Mcpt6*$^{-/-}$ mice available. We are grateful for equipment and expert technical assistance from the Interdisciplinary Neurobehavioral Core (Heidelberg University), and from the Genomics and Proteomics Core Facility and the Center for Preclinical Research (both German Cancer Research Center). This work was supported by the Helmholtz Graduate School of Cancer Research (grants to R.B., D.P., F.S. and R.T.). P.V.B. acknowledges support for the Cell & Tissue Imaging Cluster from KU Leuven and FWO IRI I000321N. T.B.F. and H.R.R. received long-term support for mast cell research from ERC Advanced Grant 233074. H.R.R. is supported by ERC Advanced Grant 742883 and the Leibniz Award of the DFG.

**Author contributions** T. Plum and H.R.R. designed the study. T. Plum, R.B., Z.W., C.F., N.S., M.F. and A.T.T. performed experiments. J.S., S.T., C.X., J.V.B., D.V. and P.V.B. supplied mice. T. Plum, R.B., R.T., F.S., D.P., T. Poth, D.A.A.M., G.B., P.V.B., R.K., J.S., C.P., H.M., T.B.F. and H.R.R. analysed and interpreted data. T.Plum and H.R.R. wrote the manuscript with input from R.B. and T.B.F. All authors critically reviewed the manuscript.

**Funding** Open access funding provided by Deutsches Krebsforschungszentrum (DKFZ).

**Competing interests** The authors declare no competing interests.

**Additional information**
**Correspondence and requests for materials** should be addressed to Thomas Plum or Hans-Reimer Rodewald.

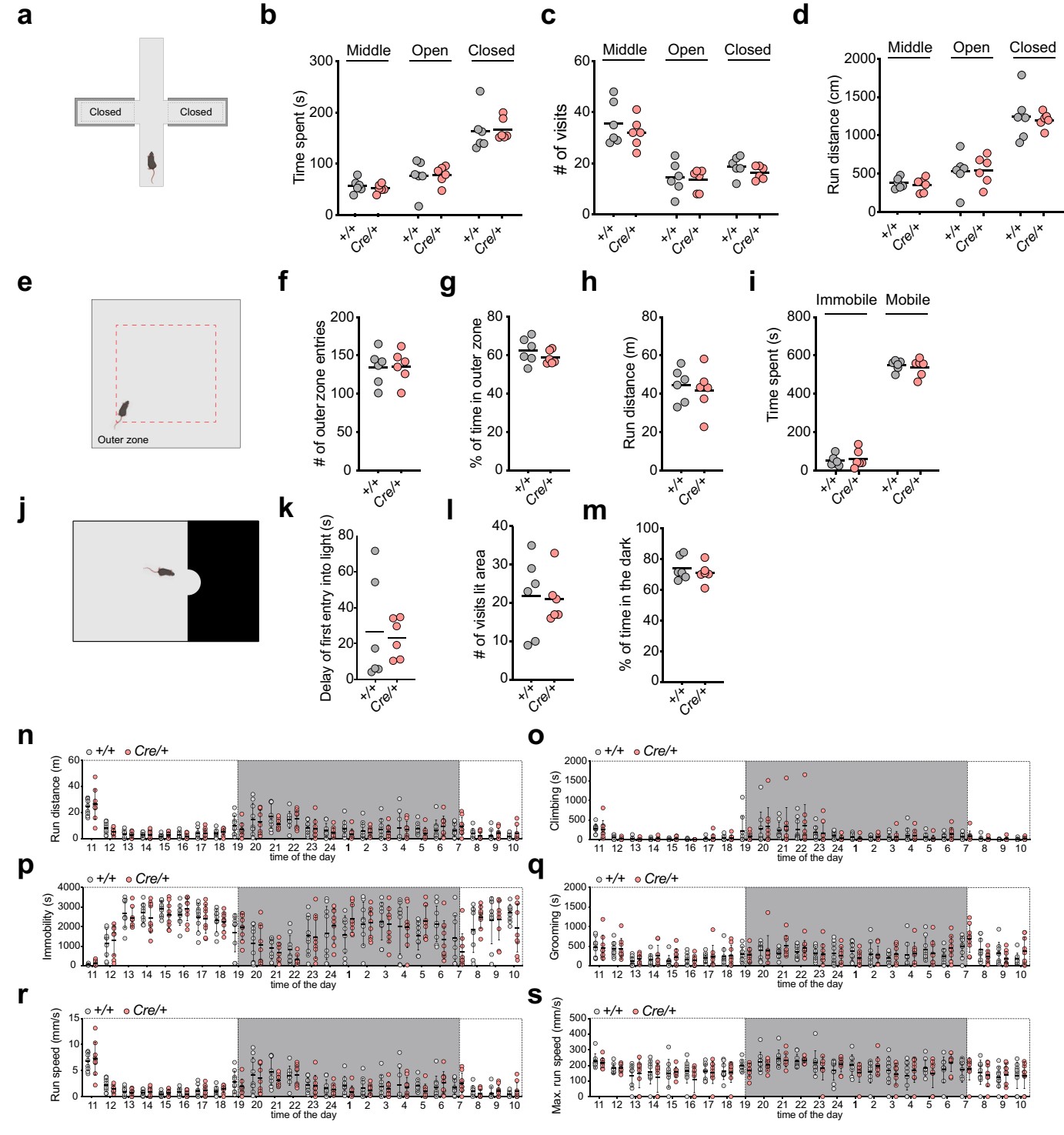

**Extended Data Fig. 1 | Anxiety and general behavioral tests of mast cell-deficient *Cpa3^Cre/+* mice.** C57BL/6 *Cpa3^+/+* and *Cpa3^Cre/+* mice were analyzed for behavioral deficits by elevated plus maze (**a**–**d**), open field (**e**–**i**), light dark chamber (**j**–**m**), and home cage monitoring (**n**–**s**). **a.** Model of the elevated plus maze. 300 s of behavior were digitally recorded, and automatically tracked using Sygnis tracker software. **b**, Time spent in middle, open, or closed arms of the plus maze. **c**, Number of visits into each arm. **d**, Run distance of mice within each arm. **e**, Model of the open field arena. 600 s of behavior were digitally recorded, and automatically tracked using ANY-maze video-tracking system. **f**, Number of outer zone entries. **g**, Time spent in outer zone as percentage of total observation time. **h**, Total run distance. **i**, Time spent immobile or mobile.

**j**, Model of the light-dark box. 600 s of behavior were digitally recorded, and automatically tracked using Sygnis tracker software. **k**, Time delay for mice for first entrance into the dark area. **l**, Number of visits into the lit area. **m**, Time spent in outer zone as percentage of total observation time. The bars represent the mean values, and each dot is a single mouse (n = 6 mice in **b**–**m**). **n**–**s**, Laboras home cage monitoring results are depicted in (**n**) run distance, (**o**) climbing duration, (**p**) duration of resting, (**q**) grooming duration, (**r**) run speed, and (**s**) maximal run speed in one-hour bins over 24-hour observation periods. Day and night phases are shaded white and grey, respectively. The bars represent the mean values (± SD), and each dot is a single mouse (n = 9 mice for **n**–**s**).

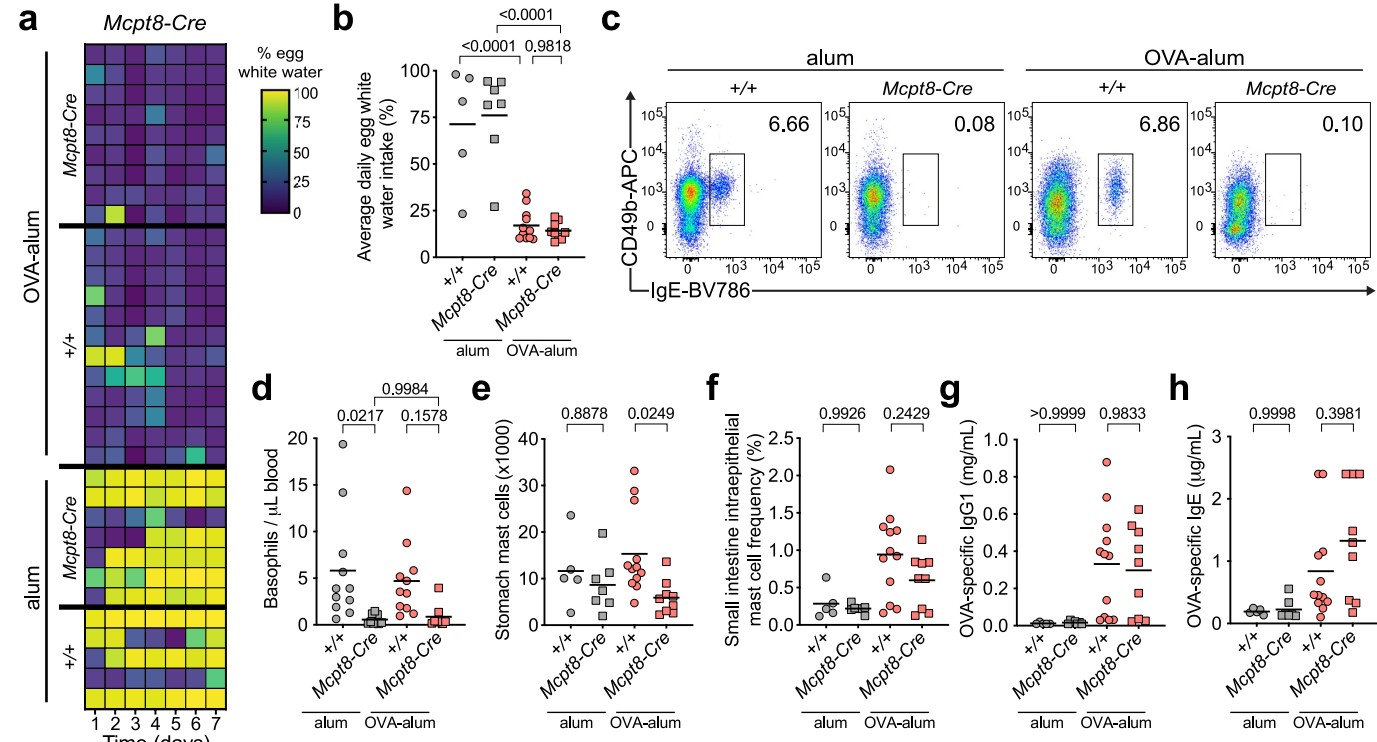

**Extended Data Fig. 2 | Antigen avoidance behavior independent of basophils. a** Basophil-deficient BALB/c *Mcpt8-Cre*, and BALB/c wild-type control (+/+) mice were immunized as in Fig. 1a, and analyzed in the two-bottle test. Egg white water preference of alum or OVA-alum immunized mice is displayed as percent egg white water intake over total water intake (color scale indicates percentages) over the course of the experiment. Each row is an individual mouse. **b**, Egg white water preference displayed as the fraction of egg white water intake over total water intake as an average per day for the duration of the experiment. **c**, Flow cytometric analysis of basophils in peripheral blood from the indicated mice. Red blood cell-lysed blood cells were stained for expression of CD49b versus IgE. Numbers in the gate indicate percentages

of basophils among live CD45$^+$CD90.2$^-$CD11c$^-$Gr-1$^-$Siglec-F$^-$MHCII$^-$B220$^-$ cells. **d**, Absolute numbers of basophils in peripheral blood from the indicated mice. Basophils were detected as in **c**, and total numbers calculated (Methods). **e,f,g,h**, Absolute numbers of stomach mast cells (**e**), and frequencies of small intestine intraepithelial mast cells among total live cells (**f**), and serum amounts of anti-OVA IgG1 (**g**), and anti-OVA IgE (**h**) at the end of the experiment. The bars represent mean values, and each dot a single mouse. +/+ alum (n = 5 mice for **a,b**, **e–h**, n = 11 for **d**), *Mcpt8-Cre* alum (n = 7 mice for **a,b**, **e–h**, n = 9 for **d**), +/+ OVA-alum (n = 12 mice for **a,b**, **e–h**, n = 11 for **d**), *Mcpt8-Cre* OVA-alum (n = 9 mice for **a,b**, **e–h**, n = 8 for **d**). Statistical analysis was performed using one-way ANOVA with Tukey multiple-comparison test (**b,d–h**). The exact P values are shown.

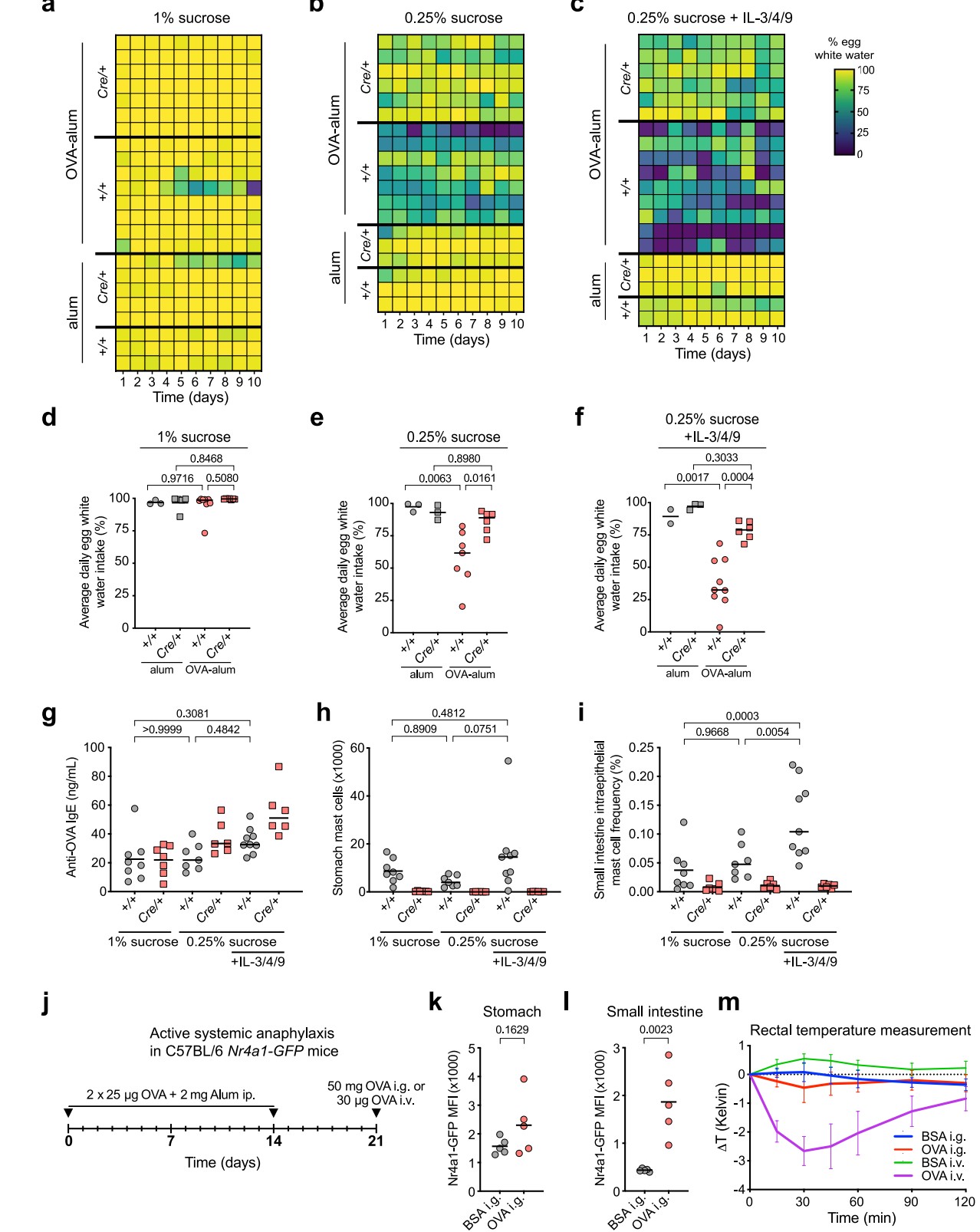

**Extended Data Fig. 3** | See next page for caption.

**Extended Data Fig. 3 | Type 2 cytokines enhance antigen avoidance behavior. a,b,c.** IntelliCage drink avoidance experiments with alum and OVA-alum immunized C57BL/6 *Cpa3*[+/+] (+/+) and *Cpa3*[Cre/+] (Cre/+) mice displayed as percent egg white water intake over total water intake for the course of the experiments. Each row is an individual mouse. **a**, Mice given free choice between water, and 20% egg white water with 1% sucrose. +/+ alum (n = 3 mice); +/+ OVA-alum (n = 8); Cre/+ alum (n = 5); Cre/+ OVA-alum (n = 7). **b**, Mice given free choice between water, and 20% egg white water with 0.25% sucrose. +/+ alum (n = 3 mice); +/+ OVA-alum (n = 7); Cre/+ alum (n = 3); Cre/+ OVA-alum (n = 6). **c**, Mice received two intraperitoneal injections of a cytokine cocktail consisting of IL-3, anti-mouse IL-3, IL-4, anti-mouse IL-4 and IL-9 (cocktail abbreviated as IL-3/4/9) on days 2 and 16 prior to the avoidance test as in (**b**). +/+ alum (n = 2 mice); +/+ OVA-alum (n = 9); Cre/+ alum (n = 3); Cre/+ OVA-alum (n = 6). **d,e,f**, Egg white water preferences (from **a,b,c**) displayed as the fraction of egg white water intake over total water intake as an average per day for the duration of the experiment. **g,h,i**, At the end of the experiment, OVA-alum immunized mice from **a**–**c** were analyzed for serum amounts of anti-OVA IgE (**g**), and absolute numbers of stomach- (**h**), and percentages of small intestine intraepithelial mast cells among total live cells (**i**). +/+ 1% sucrose (n = 8 mice), Cre/+ 1% sucrose (n = 7), +/+ 0.25% sucrose (n = 7), Cre/+ 0.25% sucrose (n = 6), +/+ 0.25 sucrose + IL3/4/9 (n = 9), Cre/+ 0.25% sucrose + IL3/4/9 (n = 6) (**g**–**i**). **j,k,l,m**, C57BL/6 *Nr4a1-GFP* mice were OVA-alum immunized. Type 2 immunization scheme and experimental timeline (**j**). **k,l**, On day 21 mice received OVA or BSA via intragastric gavage (i.g). After 2 h, GFP expression in stomach mast cells (**k**), and small intestine intraepithelial mast cells (**l**) were analyzed by flow cytometry. **m**, On day 21, mice received OVA/BSA by intragastric gavage or by intravenous injection. Body temperature of mice was monitored via rectal thermometer. BSA i.g. (n = 5) (**k**–**m**); OVA i.g. (n = 5) (**k**–**m**); BSA i.v.(n = 4) (**m**); OVA i.v. (n = 5) (**m**). Horizontal bars represent the mean values, and each dot represents a single mouse (**d-i; k,l**). Data are mean (± SEM) in (**m**). Statistical analysis was performed using one-way ANOVA with Tukey multiple-comparison test for (**d**–**i**), and two-sided students t-tests for (**k,l**). The exact P values are shown.

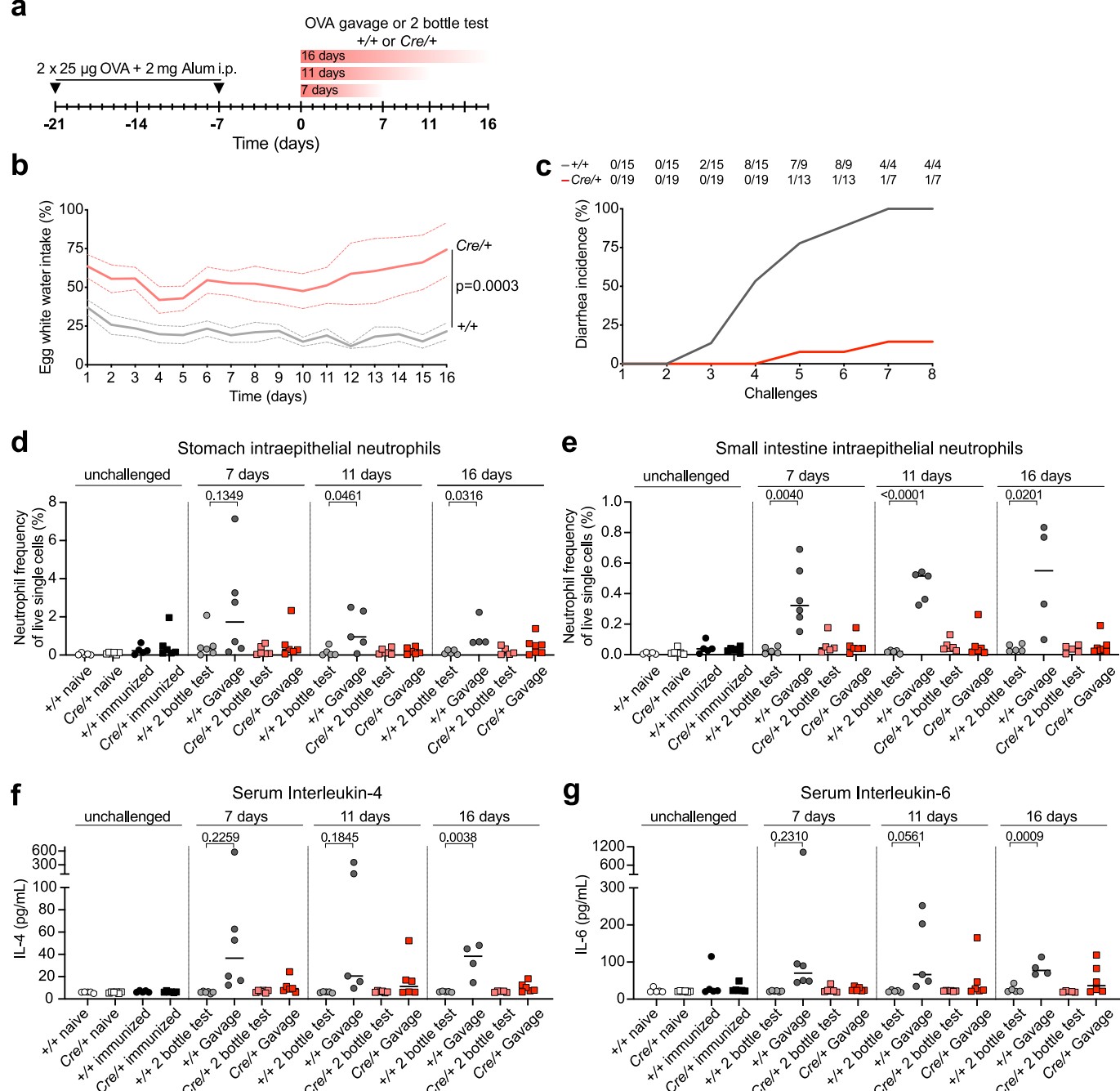

**Extended Data Fig. 4 | Ignoring avoidance results in local and systemic inflammation. a**, Experimental timeline of immunization, two-bottle test, and intragastric gavage. Groups of *Cpa3*[+/+] (+/+) and *Cpa3*[Cre/+] (Cre/+) mice were immunized, challenged, and analyzed as indicated in the figure. **b**, Egg white water intake in mice subjected to the two-bottle test. Total numbers of mice at the start of the experiment were n = 16 (+/+) and n = 17 (Cre/+). Data are mean (± SEM). Groups of mice were sacrificed on day 7 (+/+ n = 6; Cre/+ n = 6), day 11 (+/+ n = 5; Cre/+ n = 6), and day 16 (+/+ n = 5; Cre/+ n = 5) for RNA-sequencing (Fig. 3; Extended Data Fig. 5, 6) and for analyses shown in d–g. **c**, Diarrhea incidence of +/+ and Cre/+ mice on the indicated OVA gavage regimen (Methods). Total numbers of mice at the start of the experiment were n = 15 (+/+) and n = 19 (Cre/+). After 4 times gavage(day 7) n = 6 (+/+) and n = 6 (Cre/+), after 6 times

gavage (day 11) n = 5 (+/+) and n = 6 (Cre/+), and after 8 times gavage (day 16) n = 4 (+/+) and n = 7 (Cre/+)mice were sacrificed for RNA-sequencing (Fig. 3; Extended Data Fig. 5) and for analyses shown in d–g. **d,e**, Quantification of stomach neutrophils (**d**), and small intestine intraepithelial neutrophils (**e**) of animals from the indicated experimental groups and time points (a). **f,g**, Quantification of serum levels of IL-4 (**f**), and IL-6 (**g**) for the indicated experimental groups and time points (a). Control mice were naive (+/+ n = 5; Cre/+ n = 6) or immunized (+/+ n = 5; Cre/+ n = 6) but unchallenged (**d–g**). The bars represent mean values (**d–g**), and each dot a single mouse with n = 4-6 for all groups. Statistical analysis was performed using two-way ANOVA for (**b**), and two-sided students t-tests for (**d–g**). The exact P values are shown.

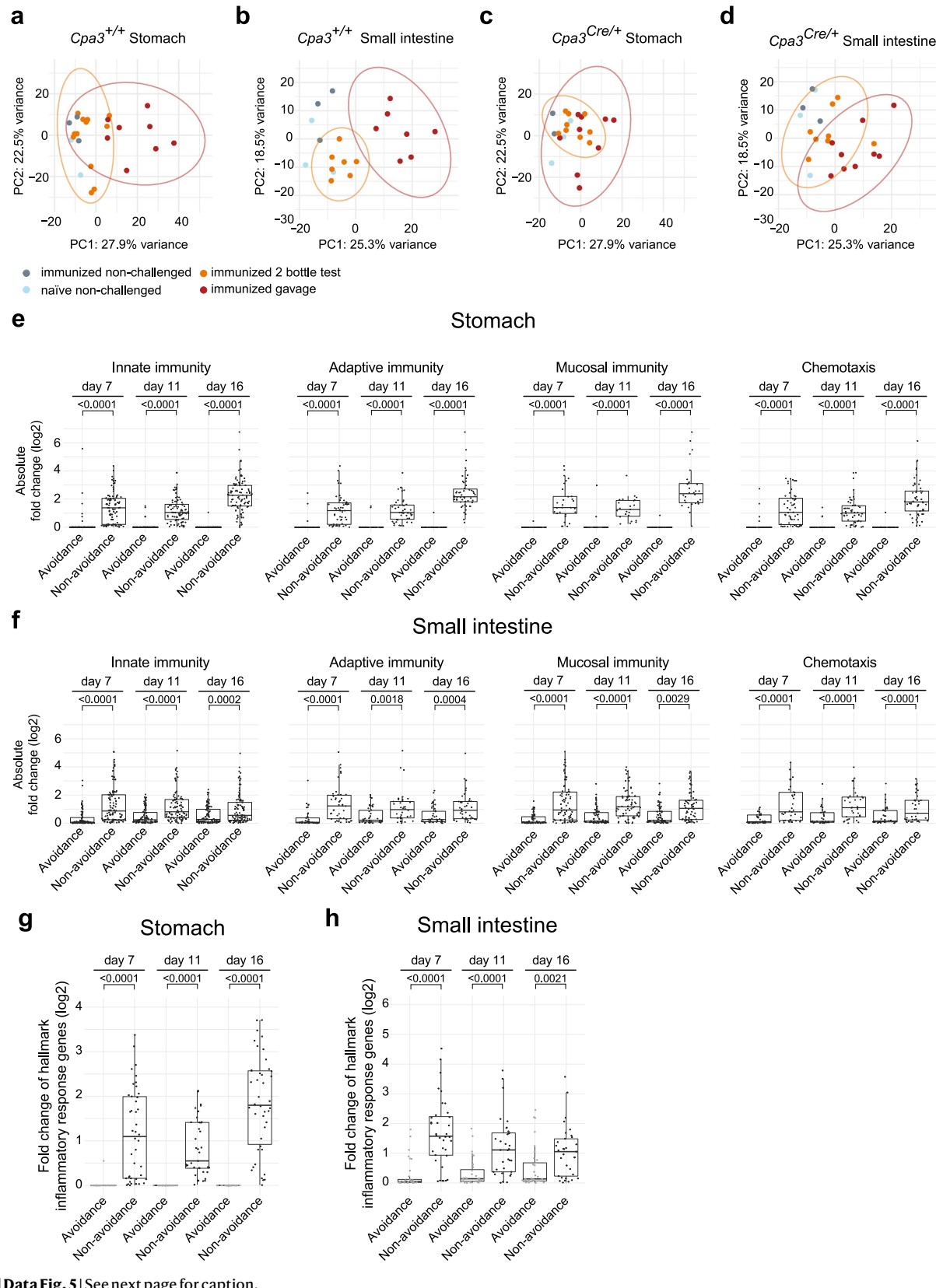

**Extended Data Fig. 5 |** See next page for caption.

**Extended Data Fig. 5 | Transcriptional analysis of gastrointestinal tissues under conditions of avoidance versus non-avoidance in BALB/c mice.**
**a**–**d**, Principal component analysis considering the 500 most variable genes in (**a**,**c**) stomach- and (**b**,**d**) small intestine tissue bulk RNA-Seq data of BALB/c *Cpa3*[+/+] and *Cpa3*[Cre/+] mice undergoing the experiment outlined in Extended Data Fig. 4a. The clustering of the samples per experimental condition attests reproducibility between mice. **e**,**f**, For experimental outline, see Extended Data Fig. 4a. Stomach (**e**) and small intestine (**f**) were analyzed by RNA sequencing on the indicated days. Based on log2Foldchange-ranked gene lists, gene set enrichment analysis revealed a strong contribution of immunological Gene Ontology (GO)-terms. We annotated these GO terms according to their description into 4 subgroups: "adaptive immunity", "innate immunity", "mucosal immunity" and "chemotaxis" (Supplementary Table 1). The underlying most impactful "core enrichment genes" (Methods) for each GO-term were categorized according to these immunological subgroups (Supplementary Table 2). Shown are absolute log2Foldchanges of core enrichment genes contained in significantly enriched immune-related GO-pathways for stomach (**e**) and small (**f**) intestine tissues of avoiding (two-bottle test) and non-avoiding (OVA gavage) mice. Genes were annotated manually as 'innate immunity', 'adaptive immunity', 'mucosal immunity', and 'chemotaxis'. **g**,**h**, Log2Foldchanges of 'Hallmark inflammatory response'[14] genes in stomach (**g**) and small intestine (**h**) tissues of avoiding (two-bottle test) and non-avoiding (OVA gavage) wild-type mice. Boxplots (**e**–**h**) show median and quantiles.

Statistical analysis was performed using two-sided Wilcoxon-rank-sum tests for (**e**–**h**). Numbers of mice in **a**–**d** were (**a**) *Cpa3*[+/+] stomach naive non-challenged (n = 2), immunized non-challenged (n = 3), immunized gavage (n = 8), immunized two-bottle test (n = 12); (**b**) *Cpa3*[+/+] small intestine naive non-challenged (n = 3), immunized non-challenged (n = 3), immunized gavage (n = 7), immunized two-bottle test (n = 7), (**c**) *Cpa3*[Cre/+] stomach naive non-challenged (n = 3), immunized non-challenged (n = 2), immunized gavage (n = 9), immunized two-bottle test (n = 9); (**d**) *Cpa3*[Cre/+] small intestine naive non-challenged (n = 3), immunized non-challenged (n = 3), immunized gavage (n = 9), immunized two-bottle test (n = 8). Numbers of mice in **e**–**h** were (**e**,**g**) *Cpa3*[+/+] stomach immunized non-avoidance (gavage) day 7 (n = 3), day 11 (n = 2), day 16 (n = 3), immunized avoidance (two-bottle test) day 7 (n = 4), day 11 (n = 4), day 16 (n = 4) (data pooled per group and compared to reference immunized non-challenged (n = 3)), (**f**,**h**) *Cpa3*[+/+] small intestine immunized non-avoidance (gavage) day 7 (n = 3), day 11 (n = 2), day 16 (n = 2), immunized avoidance (two-bottle test) day 7 (n = 3), day 11 (n = 2), day 16 (n = 2) (data pooled per group and compared to reference immunized non-challenged (n = 3)). For numbers of genes in **e**,**f**, (log2FoldChange <−1 or log2FoldChange >1.5 in at least one comparison), see Supplementary Table 1. For numbers of genes in **g**,**h**, see Supplementary Table 4. **e**–**h**, Box boundaries delineate the 1st and 3rd quartiles of the data, the center line represents the median, whiskers represent the furthest points within 1.5× the interquartile range.

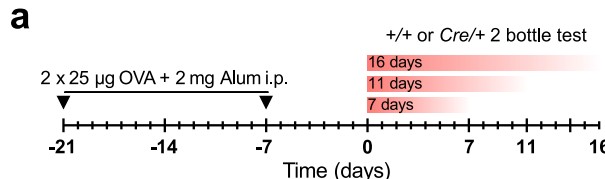

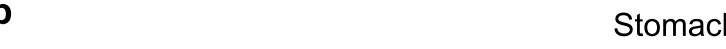

## Stomach

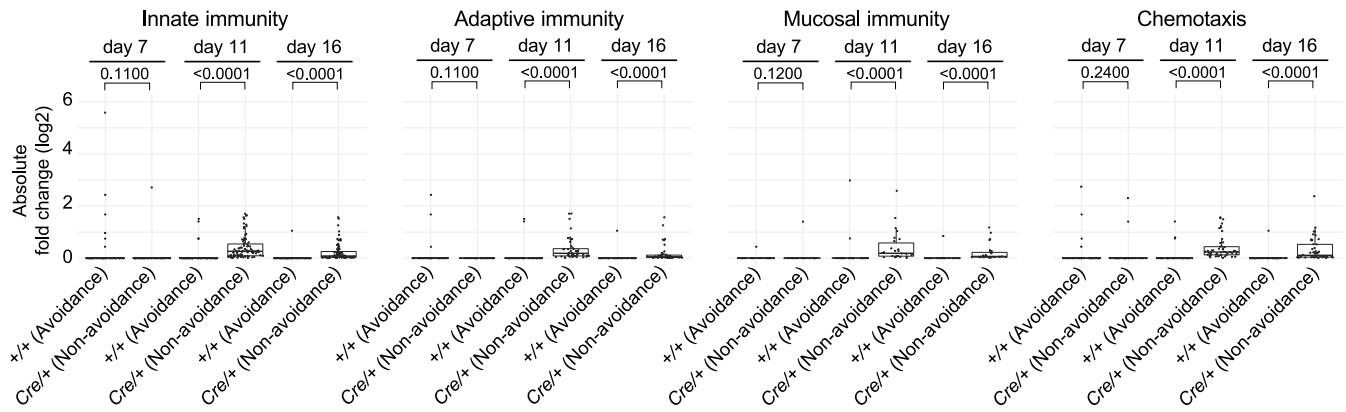

## Small intestine

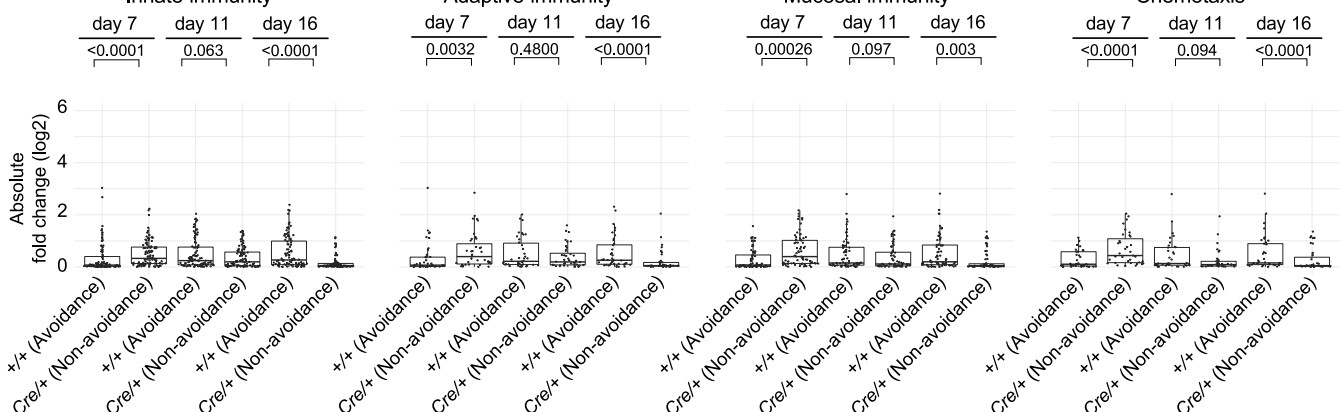

**Extended Data Fig. 6 | Mast cell-independent gastrointestinal inflammation.**
**a**, Experimental timeline of immunization and two-bottle test (Methods).
**b,c**, Absolute fold changes of core-enrichment genes contained in immune-related GO-pathways (Methods) for stomach (**b**) and small intestine (**c**) tissues of avoiding *Cpa3*[+/+] and non-avoiding *Cpa3*[Cre/+] mice from the two-bottle test. Genes were grouped into the manually curated categories 'innate immunity', 'adaptive immunity', 'mucosal immunity', and 'chemotaxis'. Boxplots show median and quantiles. Statistical analysis was performed using two-sided Wilcoxon-rank-sum tests (**b,c**). Numbers of mice in **b,c** were (**b**) *Cpa3*[Cre/+] stomach immunized non-avoidance (gavage) day 7 (n = 3), day 11 (n = 3), day 16 (n = 3),

immunized avoidance (two-bottle test) day 7 (n = 3), day 11 (n = 3), day 16 (n = 3) (data pooled per group and compared to reference immunized non-challenged (n = 2)), (**c**) *Cpa3*[Cre/+] small intestine immunized non-avoidance (gavage) day 7 (n = 3), day 11 (n = 3), day 16 (n = 3), immunized avoidance (two-bottle test) day 7 (n = 3), day 11 (n = 3), day 16 (n = 2) (data pooled per group and compared to reference immunized non-challenged (n = 3)). For numbers of genes in **b,c**, (log2FoldChange <−1 or log2FoldChange >1.5 in at least one comparison), see Supplementary Table 1. **b,c**, Box boundaries delineate the 1st and 3rd quartiles of the data, the center line represents the median, whiskers represent the furthest points within 1.5× the interquartile range.

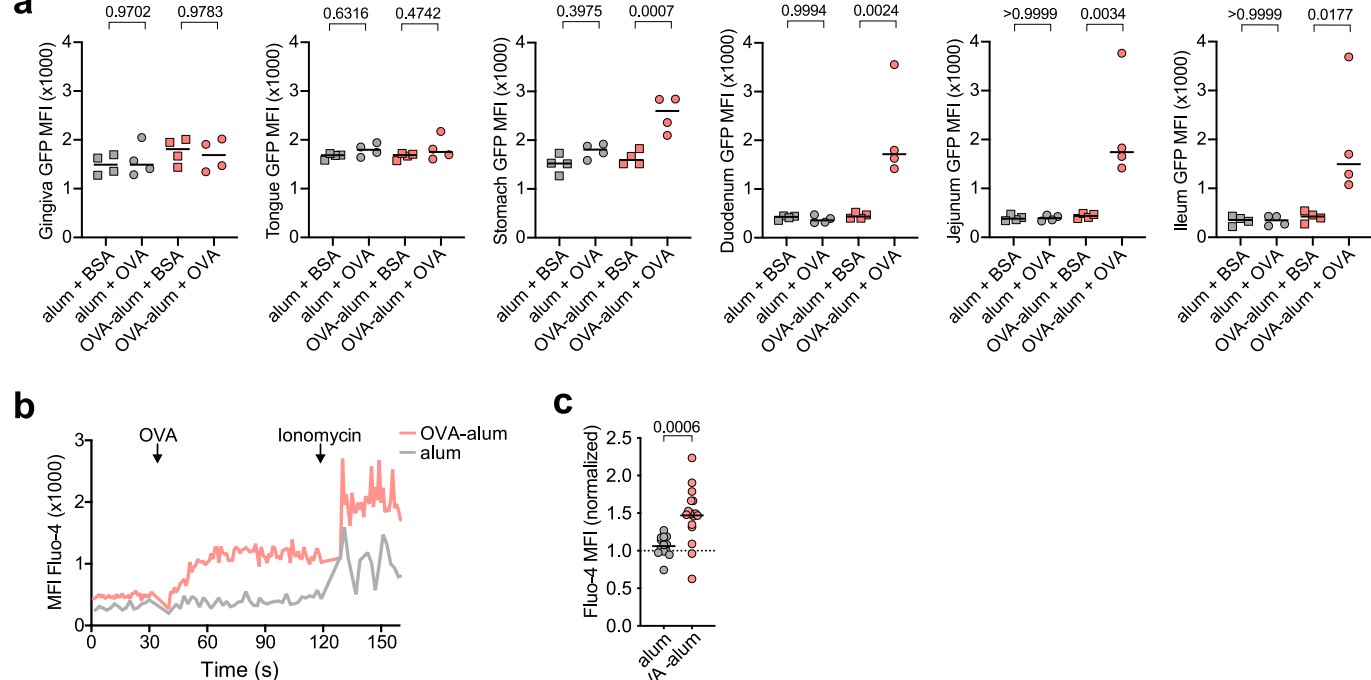

**Extended Data Fig. 7 | Freshly isolated stomach mast cells respond to OVA in vitro by intracellular Ca²⁺ flux. a**, (BALB/c x C57BL/6)F1 *Nr4a1-GFP* mice immunized with OVA-alum, or alum only, were given drinking water with OVA or BSA (Methods). GFP mean fluorescence intensity (MFI) of tissue mast cells (n = 4 for each group) isolated 3 h after mice consumed OVA- or BSA-containing water. **b**, Representative traces of Fluo-4 fluorescence (indicating intracellular Ca²⁺ concentration) in stomach mast cells from OVA-alum or alum immunized mice. Cells were stimulated with OVA and ionomycin at the indicated times (arrows). **c**, Quantification of Fluo-4 mean fluorescence intensity (as in b), normalized to the first 30 s of measurement (set as 1.0) (see Methods), comparing mast cells from OVA-alum and alum immunized mice. OVA-alum (n = 17), alum (n = 13). Statistical analysis was performed by one-way ANOVA with Tukey multiple-comparison test for (**a**), and two-sided students t-tests for (**c**). The exact P values are shown.

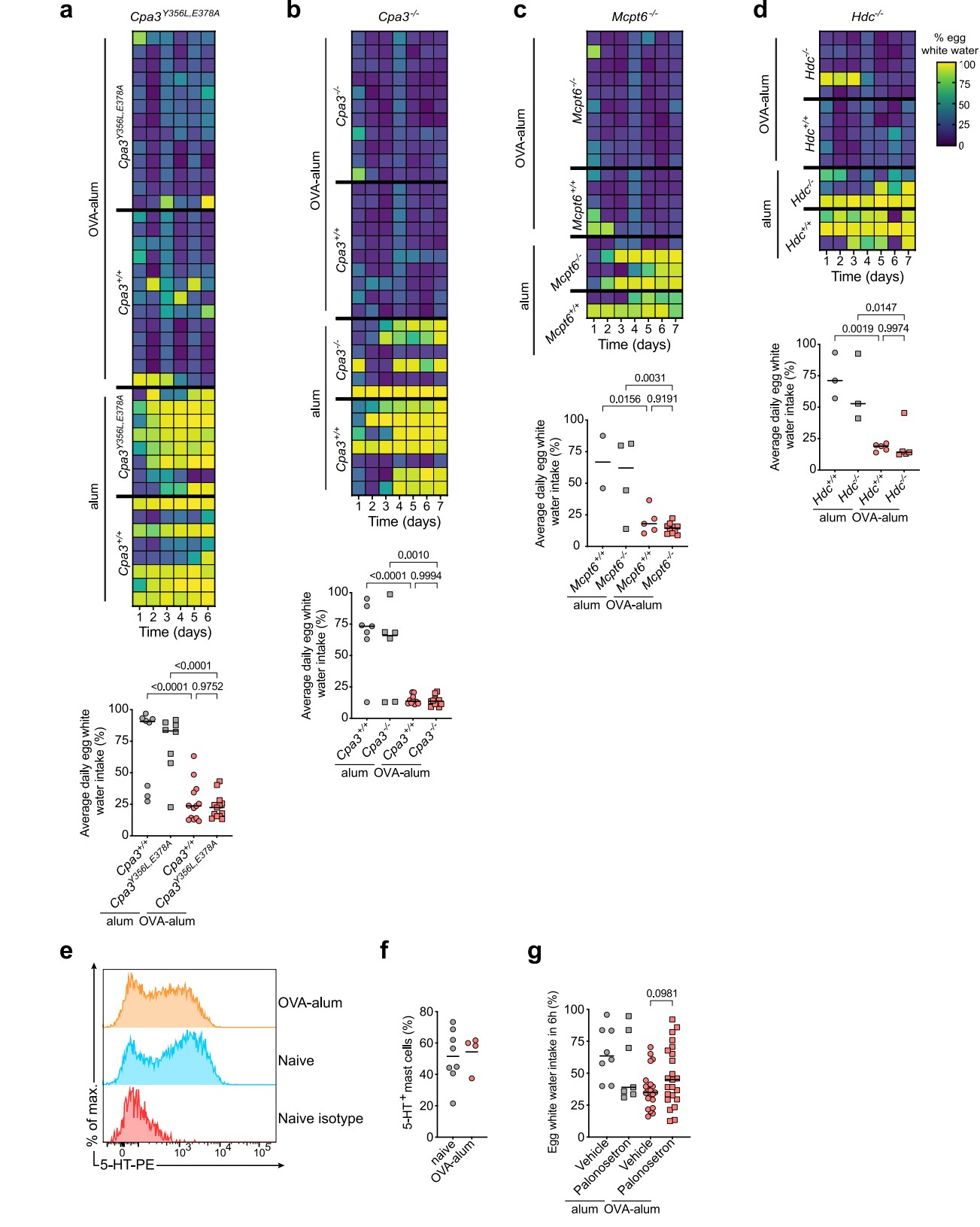

**Extended Data Fig. 8 |** See next page for caption.

**Extended Data Fig. 8 | Avoidance behavior after genetic- or pharmacologic blockade of mast cell mediators. a**–**d**, BALB/c mice lacking mast cell proteases (**a**,**b**,**c**), or histidine decarboxylase (**d**) were immunized as described in Fig. 1a. Egg white water preference is displayed as the fraction of egg white water intake over total water intake per day for the duration of the two-bottle test (heatmap; each row is an individual mouse), and displayed as average fraction of egg white water intake over total water intake for the entire experiment (lower panels). Mutant mice were *Cpa3*$^{Y356L,E378A}$ (**a**), *Cpa3*$^{-/-}$ mice (**b**), *Mcpt6*$^{-/-}$ mice (**c**), and *Hdc*$^{-/-}$ mice (**d**). *Cpa3*$^{+/+}$ alum (n = 8 mice), *Cpa3*$^{Y356L,E378A}$ alum (n = 8), *Cpa3*$^{+/+}$ OVA-alum (n = 13), *Cpa3*$^{Y356L,E378A}$ OVA-alum (n = 13) (**a**); *Cpa3*$^{+/+}$ alum (n = 7 mice), *Cpa3*$^{-/-}$ alum (n = 6), *Cpa3*$^{+/+}$ OVA-alum (n = 10), *Cpa3*$^{-/-}$ OVA-alum (n = 11) (**b**); *Mcpt6*$^{+/+}$ alum (n = 2 mice), *Mcpt6*$^{-/-}$ alum (n = 4), *Mcpt6*$^{+/+}$ OVA-alum (n = 5), *Mcpt6*$^{-/-}$ OVA-alum (n = 10) (**c**); *Hdc*$^{+/+}$ alum (n = 3 mice), *Hdc*$^{-/-}$ alum (n = 3), *Hdc*$^{+/+}$ OVA-alum (n = 5), *Hdc*$^{-/-}$ OVA-alum (n = 5) mice (**d**). **e**,**f**, Stomach mast cells of naive wild-type BALB/c mice (naive) and OVA-alum immunized wild-type BALB/c mice (OVA-alum) were stained intracellularly with anti-5-HT-specific or isotype control antibodies, and analyzed by flow cytometry (Methods). Histograms (representative for data in f) of intracellular 5-HT or isotype antibody staining (**e**), and frequencies of 5-HT$^+$ stomach mast (**f**) are shown. Naive (n = 8 mice), OVA- alum (n = 4) (**f**). **g**, BALB/c wild-type mice were immunized with OVA-alum, and 12 h prior to the two-bottle test mice were treated with PBS or palonosetron (Methods). Displayed are percentages of egg white water intake per total water intake over 6 h. Alum vehicle (n = 8 mice); alum palonosetron (n = 7); OVA-alum vehicle (n = 21); OVA-alum palonosetron (n = 23) (**g**). The bars (**a**–**d**, **f**,**g**) represent the mean values, and each dot is a single mouse. Statistical analysis was performed using one-way ANOVA with Tukey multiple-comparison test (**a**–**d**), and two-sided students t-test for (**g**). The exact P values are shown.

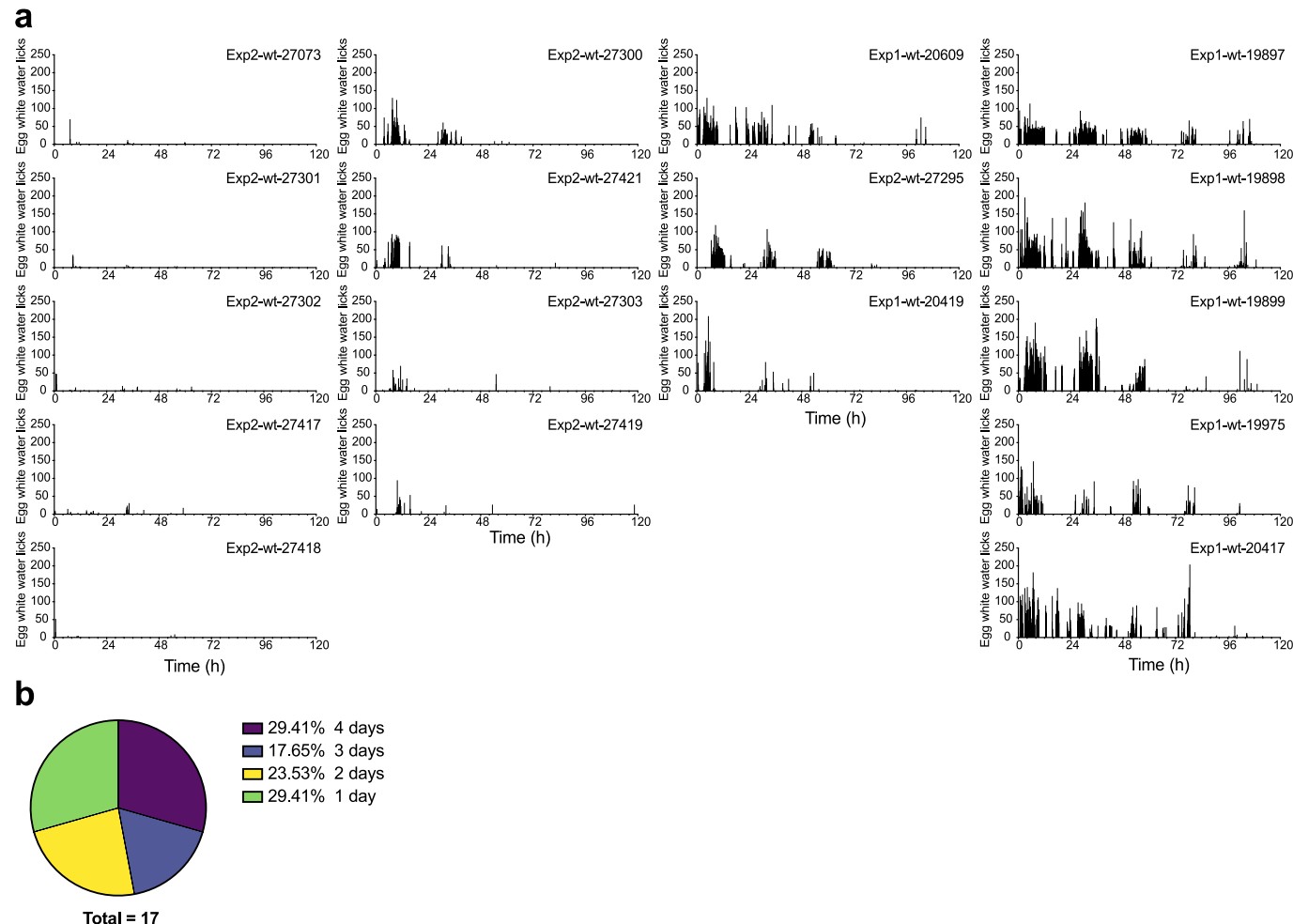

**a**

**b**

- 29.41%  4 days
- 17.65%  3 days
- 23.53%  2 days
- 29.41%  1 day

Total = 17

**Extended Data Fig. 9 | Kinetics of OVA avoidance. a**, Time-resolved egg white water lick counts of wild-type BALB/c mice (data taken from Fig. 1c, experiments 1 and 2). Egg white water licks of individual wild-type mice were recorded over the course of 12 days in the IntelliCage system, the first 120 h are shown. The experiment and mouse ID are stated in the top right corner of each graph (n = 17). **b**, Pie chart showing the distribution of the onset of egg white water avoidance in days in wild-type mice.

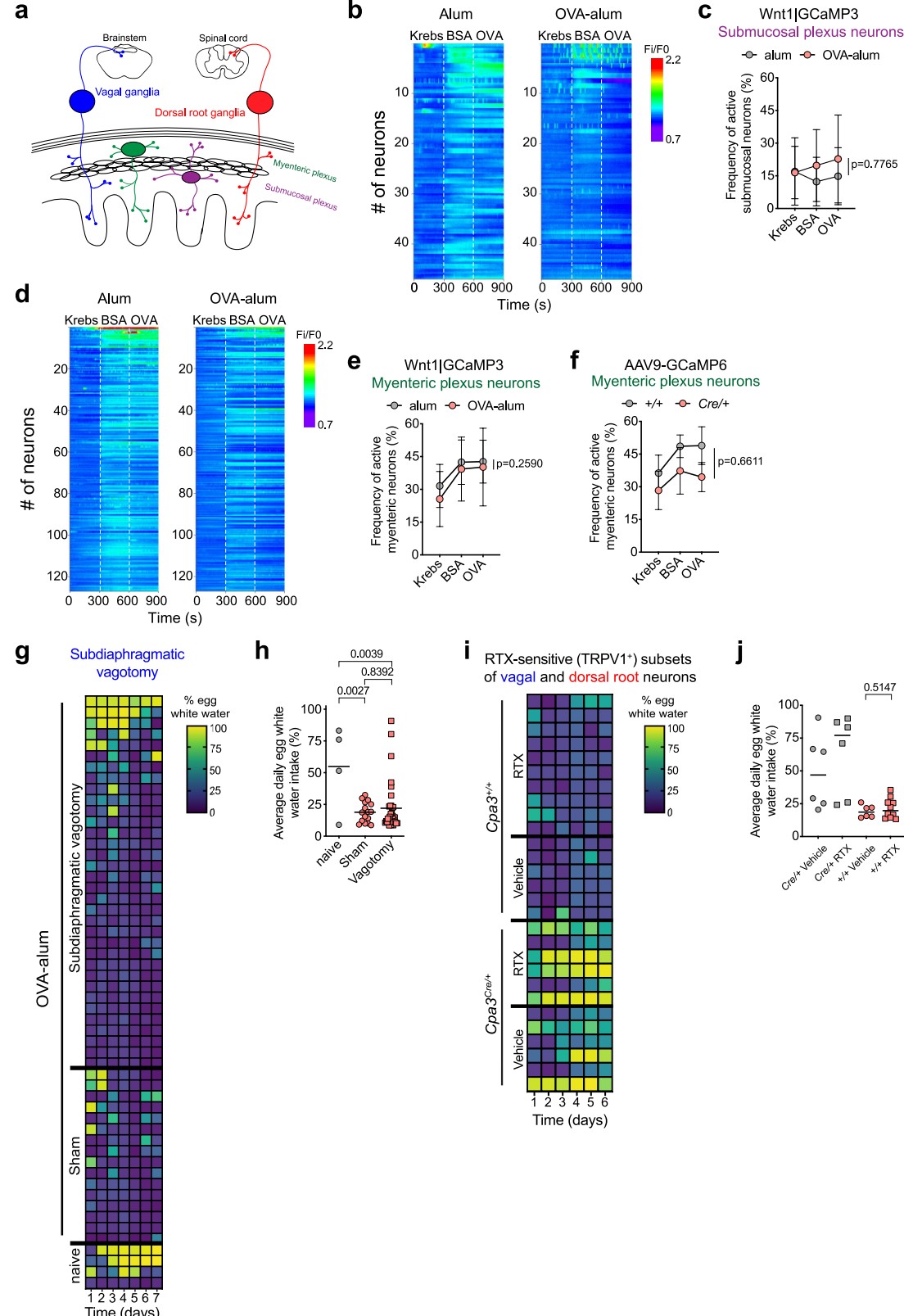

**Extended Data Fig. 10** | See next page for caption.

**Extended Data Fig. 10 | Enteric-, vagal-, and Trpv1-expressing neurons not involved in antigen avoidance. a**, Model for gut-brain signaling pathways. Extrinsic vagal ganglion and dorsal root ganglion neurons signal from gut to brainstem and spinal cord, respectively. Intrinsic primary afferent neurons reside within submucosal and myenteric plexi. **b**–**e**, Microscopic recordings of submucosal plexus (**b,c**), myenteric plexus (**d,e**) neuron $Ca^{2+}$ transients in full thickness gut preparations of alum and OVA-alum immunized Wnt1|GCaMP3 mice after consecutive mucosal superfusion of Krebs buffer (t = 0–299s), 1% BSA (t = 300–599s), and 1% OVA (t = 600–899s). **b,c**, based on analysis of 188 neurons from n = 5 alum mice, and of 178 neurons from n = 5 OVA-alum mice, calcium transients from 47 representative neurons (from n = 3 alum mice, and n = 3 OVA-alum mice) are shown (**b**), as well as proportions of active neurons among total neurons (**c**). **d,e**, based on analysis of 442 neurons from n = 5 alum mice, and of 419 neurons from n = 5 OVA-alum mice, calcium transients from 127 representative neurons (from n = 3 alum mice, and n = 3 OVA-alum mice) are shown (**d**), as well as proportions of active neurons among total neurons (**e**). **f**, $Cpa3^{+/+}$ (+/+) and $Cpa3^{Cre/+}$ (Cre/+) mice were transduced with the $Ca^{2+}$ sensor GCaMP6 from a viral vector construct (Methods), and immunized as in Fig. 1a. Recordings of $Ca^{2+}$ transients in myenteric plexus neurons in full thickness gut preparations (as in **b**). Shown is the proportion of active neurons among total neurons (n = 52 neurons from 2 alum mice, and 56 neurons from n = 2 OVA-alum mice). **g,h**, Egg white water preferences of naive or OVA-alum immunized BALB/c mice that underwent pyloroplasty (sham) or vagotomy (Methods) displayed as (**g**) the percentages of egg white water intake over total water intake over the course of the experiment, and (**h**) the fraction of egg white water intake over total water intake as average per day for the duration of the two-bottle test. In **g**, each row is an individual mouse (naive n = 4; sham n = 16; vagotomy n = 34). **i**, **j**, Egg white water preferences of OVA-alum immunized BALB/c $Cpa3^{+/+}$ and $Cpa3^{Cre/+}$ mice that underwent vehicle or resiniferatoxin (RTX)-injection (Methods). In **i**, shown are percentages of egg white water intake over total water intake over the course of the experiment. Each row is an individual mouse ($Cpa3^{Cre/+}$ vehicle n = 6; $Cpa3^{Cre/+}$ RTX n = 6; $Cpa3^{+/+}$ vehicle n = 6; $Cpa3^{+/+}$ RTX n = 10). In (**j**) the fraction of egg white water intake over total water intake as average per day over the duration of the two-bottle test. In **c,e,f**, dots represent mean values ± SD of microscopy recordings. In **h,j**, each dot is a single mouse, and bars represent mean values. Statistical analysis was performed using two-way ANOVA for (**c,e,f**,), and one-way ANOVA with Tukey multiple-comparison test for (**h,j**). The exact P values are shown.

| | |
|---|---|

# Reporting Summary

## Statistics

For all statistical analyses, confirm that the following items are present in the figure legend, table legend, main text, or Methods section.

| n/a | Confirmed | |
|---|---|---|
| ☐ | ☒ | The exact sample size (*n*) for each experimental group/condition, given as a discrete number and unit of measurement |
| ☐ | ☒ | A statement on whether measurements were taken from distinct samples or whether the same sample was measured repeatedly |
| ☐ | ☒ | The statistical test(s) used AND whether they are one- or two-sided<br>*Only common tests should be described solely by name; describe more complex techniques in the Methods section.* |
| ☒ | ☐ | A description of all covariates tested |
| ☐ | ☒ | A description of any assumptions or corrections, such as tests of normality and adjustment for multiple comparisons |
| ☐ | ☒ | A full description of the statistical parameters including central tendency (e.g. means) or other basic estimates (e.g. regression coefficient) AND variation (e.g. standard deviation) or associated estimates of uncertainty (e.g. confidence intervals) |
| ☐ | ☒ | For null hypothesis testing, the test statistic (e.g. *F*, *t*, *r*) with confidence intervals, effect sizes, degrees of freedom and *P* value noted<br>*Give P values as exact values whenever suitable.* |
| ☒ | ☐ | For Bayesian analysis, information on the choice of priors and Markov chain Monte Carlo settings |
| ☒ | ☐ | For hierarchical and complex designs, identification of the appropriate level for tests and full reporting of outcomes |
| ☒ | ☐ | Estimates of effect sizes (e.g. Cohen's *d*, Pearson's *r*), indicating how they were calculated |

*Our web collection on statistics for biologists contains articles on many of the points above.*

## Software and code

Policy information about availability of computer code

| Data collection | BD FACS Diva (v.6.1.2); IntelliCage Plus software (v.3.2.3); Sygnis Tracker software (v.4.1.4); LABORAS software (v.2.6) |
|---|---|
| Data analysis | Graphpad Prism (v.9.0); FlowJo (v.10.7.1); STAR aligner (v.2.5.2b); subread package (v.1.5.1); DESeq2 (v.1.38.3); clusterProfiler (v.3.1.6); The custom RNA-Seq analysis pipeline has been made publicly available via: https://github.com/robinthiele/AAPIA |

For manuscripts utilizing custom algorithms or software that are central to the research but not yet described in published literature, software must be made available to editors and reviewers. We strongly encourage code deposition in a community repository (e.g. GitHub). See the Nature Portfolio guidelines for submitting code & software for further information.

## Data

Policy information about availability of data

All manuscripts must include a data availability statement. This statement should provide the following information, where applicable:
- Accession codes, unique identifiers, or web links for publicly available datasets
- A description of any restrictions on data availability
- For clinical datasets or third party data, please ensure that the statement adheres to our policy

The RNA-Seq data, deposited in NCBI's Gene Expression Omnibus, are accessible through GEO Series accession number GSE225054 (https://www.ncbi.nlm.nih.gov/geo/query/acc.cgi?acc=GSE225054). There are no restrictions.

# Human research participants

Policy information about studies involving human research participants and Sex and Gender in Research.

| | |
|---|---|
| Reporting on sex and gender | n/a |
| Population characteristics | n/a |
| Recruitment | n/a |
| Ethics oversight | n/a |

Note that full information on the approval of the study protocol must also be provided in the manuscript.

# Field-specific reporting

Please select the one below that is the best fit for your research. If you are not sure, read the appropriate sections before making your selection.

☒ Life sciences          ☐ Behavioural & social sciences          ☐ Ecological, evolutionary & environmental sciences

For a reference copy of the document with all sections, see nature.com/documents/nr-reporting-summary-flat.pdf

# Life sciences study design

All studies must disclose on these points even when the disclosure is negative.

| | |
|---|---|
| Sample size | Sample sizes for antigen-avoidance experiments were calculated based on the variability in the data in ref. 4. For other experiments, sample sizes were chosen based on prior experience and pilot experiments for testing statistically significant differences between conditions. In compliance with the regulatory guidelines, a minimal number of animals for statistically significant data was used. |
| Data exclusions | Due to loss of transponder signal over several days, one mouse was excluded from the IntelliCage analysis of mast cell-deficient BALB/c mice (Fig. 1). |
| Replication | The main findings were reliably reproduced in independent experiments. All attempts at replication were successful. In the case where one experiment was done the data allowed significant conclusions on differences between genotypes and conditions. <br> Numbers of experiments: <br> Data in Fig. 1 are derived from two independent experiments (total number of 51 mice). <br> Data in Fig. 2a-e are derived from one experiment (total number of 30 mice) <br> Data in Fig. 2f-i are derived from one experiment (total number of 13 mice) <br> Data in Fig. 2 j-l are derived from two experiments (total number of 13 mice). <br> Data in Fig. 3 are derived from two experiments (total number of 89 mice). <br> Data in Fig. 4b,c are derived from two experiments (total number of 16 mice). <br> Data in Fig. 4e are derived from one experiment (total number of 30 mice). <br> Data in Fig. 4f are derived from two experiments (total number of 66 mice). <br> Data in Ext. Data Fig. 1 are derived from one experiment (total number of 30 mice). <br> Data in Ext. Data Fig. 2a,b,d-h are derived from two experiments (total number of 48 mice). <br> Data in Ext. Data Fig. 3a-i are derived from three experiments (total number of 62 mice). <br> Data in Ext. Data Fig. 3k-m are derived from two experiments (total number of 19 mice). <br> Data in Ext. Data Fig. 4, 5, 6 are derived from two experiments (total number of 89 mice). <br> Data in Ext. Data Fig. 7 are derived from two experiments (total number of 30 mice). <br> Data in Ext. Data Fig. 8a are derived from three experiments  (total number of 42 mice). <br> Data in Ext. Data Fig. 8b are derived from two experiments  (total number of 34 mice). <br> Data in Ext. Data Fig. 8c are derived from one experiment  (total number of 21 mice). <br> Data in Ext. Data Fig. 8d are derived from one experiment  (total number of 16 mice). <br> Data in Ext. Data Fig. 8e,f are derived from one experiment  (total number of 12 mice). <br> Data in Ext. Data Fig. 8g are derived from four experiments (total number of 59 mice). <br> Data in Ext. Data Fig. 9 are derived from two experiments (total number of 17 mice). <br> Data in  Ext. Data Fig. 10b-e are derived from two experiments (total number of 10 mice). <br> Data in  Ext. Data Fig. 10f are derived from one experiment (total number of 4 mice). <br> Data in  Ext. Data Fig. 10g,h are derived from two independent experiments (total number of 54 mice). <br> Data in  Ext. Data Fig. 10i,j are derived from one experiment (total number of 28 mice). |
| Randomization | Mice were genotyped, and prior to the experiments assigned to test and control groups. Because the genotype dictates the group assingment no randomization was possible. |
| Blinding | Investigators were not blinded in setting up experimental groups, as knowledge of genotypes was essential to plan the studies. Regarding data acquisition, scientists were not blinded to the genotypes and the treatment of the animals, except for automated data analysis, such as in IntelliCage experiments, RNA-Seq experiments, and anxienty and general behavioral assays. |

# Reporting for specific materials, systems and methods

We require information from authors about some types of materials, experimental systems and methods used in many studies. Here, indicate whether each material, system or method listed is relevant to your study. If you are not sure if a list item applies to your research, read the appropriate section before selecting a response.

## Materials & experimental systems

| n/a | Involved in the study |
|-----|------------------------|
| ☐ | ☒ Antibodies |
| ☒ | ☐ Eukaryotic cell lines |
| ☒ | ☐ Palaeontology and archaeology |
| ☐ | ☒ Animals and other organisms |
| ☒ | ☐ Clinical data |
| ☒ | ☐ Dual use research of concern |

## Methods

| n/a | Involved in the study |
|-----|------------------------|
| ☒ | ☐ ChIP-seq |
| ☐ | ☒ Flow cytometry |
| ☒ | ☐ MRI-based neuroimaging |

## Antibodies

**Antibodies used**

B220 FITC (RA3-6B2, BD Pharmingen, Cat. #553087) 1:50
CD3 BV421 (17A2, Biolegend, Cat. #100228) 1:200
CD3 FITC (17A2, BD Pharmingen, Cat. #555274) 1:50
CD3 Pe-Cy7 (145-2C11, BD Pharmingen, Cat. #552774) 1:25
CD11b PerCP-Cy5.5 (M1/70, eBioscience, Cat. #45-0112-82) 1:400
CD11b BV421 (M1/70, Biolegend, Cat. #101251) 1:400
CD11b PE-Cy-7 (M1/70, eBioscience, Cat. #25-0112) 1:400
CD11c BV421 (N418, Biolegend, Cat. #117330) 1:100
CD16/32 unconjugated (93, Biolegend, Cat. #101301) 10 µg/mL
CD19 BV421 (6D5, Biolegend, Cat. #115537) 1:400
CD19 APC (1D3, BD Pharmingen, Cat. #550992) 1:400
CD45 BV421 (30-F11, Biolegend, Cat. #103133) 1:400
CD45 BV785 (30-F11, Biolegend, Cat. #103149) 1:400
CD49b APC (DX5, BD Pharmingen, Cat. #560628) 1:100
CD90.2 APC-Cy7 (30-H12, Biolegend, Cat. #105328) 1:400
CD117 PE (2B8, eBioscience, Cat. #12-1171) 1:800
CD117 BV711 (2B8, Biolegend, Cat. #105835) 1:800
CD117 APC (2B8, BD Pharmingen, Cat. #553356) 1:800
FcεRI APC (MAR-1, eBioscience, Cat. #17-5898-82) 1:200
Gr-1 BV421 (RB6-8C5, BioLegend, Cat. #108445) 1:800
Gr-1 BV605 (RB6-8C5, Biolegend, Cat. #108439) 1:200
IgE PE (RME1, Biolegend, Cat. #406907) 1:100
IgE BV786 (RME-1, BD Pharmingen, Cat. #564206) 1:100
IgE BV421 (R35-72, BD Pharmingen, Cat. #564207) 1:100
Ly6G PerCP-Cy5.5 (1A8, BD Pharmingen, Cat. #560602) 1:100
MHCII A700 (M5/114.15.2, eBioscience, Cat. #56-5321-82) 1:100
Siglec-F BV421 (E50-2440, BD Pharmingen, Cat. #565934) 1:100
Siglec-F PE (E50-2440, BD Pharmingen, Cat. #552126) 1:100
Ter119 BV421 (Ter119, Biolegend, Cat. #116234) 1:200
5-HT unconjugated (5HT-H209, Dako, Cat. #M0758) 0.11 µg/mL
mouse-IgG1 PE (RMG1-1, Biolegend, Cat. #406607) 1:100
Anti-OVA IgG1 (L71, Biozol, Cat. #CHX-3013) 1:2000
Anti-OVA IgE (2C6, Invitrogen, Cat. #MA1-80396) 1:2000
Anti-mouse IgG1-HRP (X56, BD Pharmingen, Cat. #559626) 1:2000
Anti-mouse IgE-HRP (23G3, SouthernBiotech, Cat. #1130-01) 1:2000

**Validation**

Validation statement for each antibody is provided on the manufacturer's website:
B220 FITC https://www.bdbiosciences.com/en-de/products/reagents/flow-cytometry-reagents/research-reagents/single-color-antibodies-ruo/fitc-rat-anti-mouse-cd45r-b220.553087
CD3 BV421 https://www.biolegend.com/en-us/products/brilliant-violet-421-anti-mouse-cd3-antibody-7326
CD3 FITC https://www.bdbiosciences.com/en-de/products/reagents/flow-cytometry-reagents/research-reagents/single-color-antibodies-ruo/fitc-rat-anti-mouse-cd3-molecular-complex.555274
CD3 Pe-Cy7 https://www.citeab.com/antibodies/2414263-552774-bd-pharmingen-pe-cy-7-hamster-anti-mouse-cd3
CD11b PerCP-Cy5.5 https://www.thermofisher.com/antibody/product/CD11b-Antibody-clone-M1-70-Monoclonal/45-0112-82
CD11b BV421 https://www.biolegend.com/en-us/products/brilliant-violet-421-anti-mouse-human-cd11b-antibody-7163
CD11b PE-Cy-7 https://www.thermofisher.com/antibody/product/CD11b-Antibody-clone-M1-70-Monoclonal/25-0112-82
CD11c BV421 https://www.biolegend.com/en-us/products/brilliant-violet-421-anti-mouse-cd11c-antibody-7149
CD16/32 unconjugated https://www.biolegend.com/en-us/products/purified-anti-mouse-cd16-32-antibody-190
CD19 BV421 https://www.biolegend.com/en-us/products/brilliant-violet-421-anti-mouse-cd19-antibody-7160
CD19 APC https://www.bdbiosciences.com/en-ca/products/reagents/flow-cytometry-reagents/research-reagents/single-color-

antibodies-ruo/apc-rat-anti-mouse-cd19.550992
CD45 BV421 https://www.biolegend.com/en-us/products/brilliant-violet-421-anti-mouse-cd45-antibody-7253
CD45 BV785 https://www.biolegend.com/en-us/products/brilliant-violet-785-anti-mouse-cd45-antibody-10636
CD49b APC https://www.bdbiosciences.com/en-de/products/reagents/flow-cytometry-reagents/research-reagents/single-color-antibodies-ruo/apc-rat-anti-mouse-cd49b.560628
CD90.2 APC-Cy7 https://www.biolegend.com/en-us/products/apc-cyanine7-anti-mouse-cd90-2-thy1-2-antibody-6671
CD117 PE https://www.thermofisher.com/antibody/product/CD117-c-Kit-Antibody-clone-2B8-Monoclonal/12-1171-82
CD117 BV711 https://www.biolegend.com/en-us/products/brilliant-violet-711-anti-mouse-cd117-c-kit-antibody-12049
CD117 APC https://www.bdbiosciences.com/en-de/products/reagents/flow-cytometry-reagents/research-reagents/single-color-antibodies-ruo/apc-rat-anti-mouse-cd117.553356
FcεRI APC https://www.thermofisher.com/antibody/product/FceR1-alpha-Antibody-clone-MAR-1-Monoclonal/17-5898-82
Gr-1 BV421 https://www.biolegend.com/en-us/products/brilliant-violet-421-anti-mouse-ly-6g-ly-6c-gr-1-antibody-7201
Gr-1 BV605 https://www.biolegend.com/en-us/products/brilliant-violet-605-anti-mouse-ly-6g-ly-6c-gr-1-antibody-8724
IgE PE https://www.biolegend.com/en-us/products/pe-anti-mouse-ige-3267
IgE BV786 https://www.bdbiosciences.com/en-de/products/reagents/flow-cytometry-reagents/research-reagents/single-color-antibodies-ruo/bv786-rat-anti-mouse-ige.564206
IgE BV421 https://www.bdbiosciences.com/en-de/products/reagents/flow-cytometry-reagents/research-reagents/single-color-antibodies-ruo/bv421-rat-anti-mouse-ige.564207
Ly6G PerCP-Cy5 https://www.bdbiosciences.com/en-de/products/reagents/flow-cytometry-reagents/research-reagents/single-color-antibodies-ruo/percp-cy-5-5-rat-anti-mouse-ly-6g.560602
MHCII A700 https://www.thermofisher.com/antibody/product/MHC-Class-II-I-A-I-E-Antibody-clone-M5-114-15-2-Monoclonal/56-5321-82
Siglec-F BV421 https://www.bdbiosciences.com/en-de/products/reagents/flow-cytometry-reagents/research-reagents/single-color-antibodies-ruo/bv421-rat-anti-mouse-siglec.f.565934
Siglec-F PE https://www.bdbiosciences.com/en-de/products/reagents/flow-cytometry-reagents/research-reagents/single-color-antibodies-ruo/pe-rat-anti-mouse-siglec.f.552126
Ter119 BV421 https://www.biolegend.com/en-us/products/brilliant-violet-421-anti-mouse-ter-119-erythroid-cells-antibody-7259
5-HT unconjugated https://www.agilent.com/en/product/immunohistochemistry/antibodies-controls/primary-antibodies/serotonin-(concentrate)-76521
mouse-IgG1 PE https://www.biolegend.com/en-us/products/pe-anti-mouse-igg1-6494
Anti-OVA IgG1 https://www.biozol.de/de/product/CHX-3013
Anti-OVA IgE https://www.thermofisher.com/antibody/product/Ovalbumin-Antibody-clone-2C6-Monoclonal/MA1-80396
Anti-mouse IgG1-HRP https://www.bdbiosciences.com/en-de/products/reagents/immunoassay-reagents/elisa/hrp-rat-anti-mouse-igg1.559626
Anti-mouse IgE-HRP https://www.southernbiotech.com/rat-anti-mouse-ige-unlb-23g3-1130-01

# Animals and other research organisms

Policy information about studies involving animals; ARRIVE guidelines recommended for reporting animal research, and Sex and Gender in Research

| | |
|---|---|
| Laboratory animals | BALB/c wild type, C57BL/ 6 wild type, BALB/c Cpa3Cre/+, C57BL/6 Cpa3Cre/+, BALB/c Igh-7-/-, BALB/c Mcpt8-Cre, BALB/c Cpa3Y356L,E378A, BALB/c Cpa3-/-, BALB/c Hdc-/-, BALB/c Mcpt6-/-, C57BL/ 6 Nr4a1-GFP, BALB/c x C57BL/6 F1 Nr4a1-GFP, and C57BL/6 Wnt1|GCaMP3 (Wnt1-Cre;R26R-GCaMP3) mice. Adult male and female mice from 6-59 weeks of age were used for the study. |
| Wild animals | The study did not involve wild animals. |
| Reporting on sex | Both male and female mice were used. |
| Field-collected samples | The study did not involve samples collected in the field. |
| Ethics oversight | All animal experiments were performed in accordance with institutional and governmental regulations. Experiments in Heidelberg were approved by the Regierungspräsidium Karlsruhe, Germany. Experiments in Leuven were approved by the Animal Care and Animal Experiments Committee of the KU Leuven, Leuven, Belgium. |

Note that full information on the approval of the study protocol must also be provided in the manuscript.

# Flow Cytometry

## Plots

Confirm that:

☒ The axis labels state the marker and fluorochrome used (e.g. CD4-FITC).

☒ The axis scales are clearly visible. Include numbers along axes only for bottom left plot of group (a 'group' is an analysis of identical markers).

☒ All plots are contour plots with outliers or pseudocolor plots.

☒ A numerical value for number of cells or percentage (with statistics) is provided.

## Methodology

**Sample preparation**

Gingival single cell suspensions were prepared as previously described67. In brief, the palate and mandible were isolated, and tissues were digested for 1h at 37°C in RPMI supplemented with 10% FCS (Sigma-Aldrich), 0.15 µg DNase I, and 3.2 mg/mL collagenase IV (all enzymes from Sigma-Aldrich). 0.5M EDTA (Roth) was added during the last 5 min, and supernatant was filtered through a 70 µm cell strainer (ThermoFisher). Undigested gingiva tissue was then peeled from palate and mandible and mashed through the same filter to yield the gingiva cell suspension.

Tongue single cell suspensions were prepared by finely mincing the tongue and digesting the tissue for three rounds of 15-minutes at 37°C in RPMI supplemented with 0.1 mg/mL Liberase TM (Sigma-Aldrich) and 2.5 µg/mL DNase I (Sigma-Aldrich). After each round of digestion, the cell suspensions were filtered through a 70 µm cell strainer (ThermoFisher), and new enzyme solution was added to the tissue. All fractions were combined to yield the tongue single cell suspension.

For isolation of stomach intraepithelial leukocytes, the stomach was cut open and food remnants were removed. Stomachs were incubated for 15 min at 37°C in HBSS supplemented with 20 mM EDTA (Roth) to release the epithelial layers from the connective tissue. The cell suspension was applied onto a spin column (ThermoFisher) packed with 100 µm zirconia beads (Roth). After centrifugation, the flow through was collected yielding the intraepithelial cell suspension containing mucosal stomach mast cells.

For preparation of small intestine cell suspensions, small intestines were cut open and food remnants were removed. Intestines were incubated for 15 min at 37°C in HBSS supplemented with 2% FCS (Sigma-Aldrich), 5 mM EDTA (Roth), 1 mM DTT (Merck), and 10 mM HEPES (Life Technologies) to release the epithelial layers from the connective tissue. The cells in the soluble fraction (containing intraepithelial mast cells) were filtered through a 70 µm cell strainer (ThermoFisher). The remaining intestine tissue was washed in PBS and transferred into RPMI supplemented with 2% FCS (Sigma-Aldrich), 20 mM HEPES (Life Technologies), 0.2 mg/mL collagenase IV (Sigma-Aldrich), 0.5 mg/mL Hyaluronidase I (Sigma-Aldrich), and 0.1 mg/mL DNase I (Sigma-Aldrich). Digestion was carried out for 30 min at 37°C and digested tissue was filtered through a 100 µm cell strainer (ThermoFisher) yielding the lamina propria fraction (containing lamina propria mast cells).

Blood was drawn by cardiac puncture, followed by red blood cell lysis according to the manufacturer's protocol (RBC Lysis Buffer, BioLegend).

**Instrument**

LSR Fortessa (BD Bioscience)

**Software**

BD FACS Diva (v 6.1.2)

**Cell population abundance**

Cell populations were quantified as absolute numbers (per stomach or per microliter of blood), or frequencies of total live cells (intestine).

**Gating strategy**

Tongue and gingiva mast cells: live CD45+MHCII−CD11b−CD117+FcεRI+
Stomach mast cells: live CD45+CD117+FcεRI/IgE+
Intestinal mast cells: live CD45+CD3-CD19-Gr-1-Ter119-Siglec-F-CD117+FcεRI+
Basophils: live CD45+CD90.2-CD3- CD11c-Gr-1-Siglec-F-B220-MHCII-CD49b+IgE+
Stomach neutrophils: live CD45+CD11b+MHCII-Siglec-F-Ly6G+
Small intestine neutrophils: live CD45+CD11b+MHCII-Siglec-F-Gr-1+

☒ Tick this box to confirm that a figure exemplifying the gating strategy is provided in the Supplementary Information.

