## [Peer Review File · Nature]

Manuscript Title: Mast cells link immune sensing to antigen avoidance behavior

Reviewer Comments & Author Rebuttals

Reviewer Reports on the Initial Version:

Referee #1 (Remarks to the Author):

The authors investigated avoidance behavior (drinking behavior egg white 8% sucrose and 20% OVA in water) in allergic disease by using OVA-induced asthma Balb/c Cpa3 (mast cell deficient model). They demonstrated that antigen avoidance behavior was related with mast cell dependent and antigen-specific IgE. OVA induced mast cell accumulation from mouth to intestine was detected in the mast cell-deficient model. Surprisingly, mast cell deficiency did not trigger intestine pathology but allergen aversion. The authors confirmed these results in C57BL/6 Cpa3 mouse as well. To find a connection between neuronal induction, mast cells and OVA avoidance, subdiaphragmatic vagotomy was performed and did not show any connection. The authors found that intrinsic enteric neurons are not playing a crucial role in the model by measuring calcium. Extrinsic neuronal pathways molecules called TRPV1 were demonstrated and found that dorsal ganglion contribute to mast cell avoidance signaling to brain. The authors finally checked which mast cell molecules (Cpa3, Mcpt5, Mcpt6, hdc, 5-HTR3) play a role in the model. They suggested that OVA avoidance is partially mediated by signaling through 5- HTR3. There are some suggestions and criticisms that may improve the manuscript.

- 1- Can egg white 8% sucrose solution affect behavior and metabolism in mice model? (PMID: 27126968) please explain it in text.
- 2- In fig 2 Please explain type 2 cytokines enhances antigen avoidances ? Is it IL-4, IL-5, IL-9 or IL-13 ? or Please explain in legend and text IL-3/4/9 ? Is it cocktail ?
- 3- The authors must explain egg white is endotoxin free or not ? Please explain it.
- 4- The authors must explain cfos expression in their expression.
- 5- It would be proper to say that vagal and trpv1 neurons are not involved in this pathway since the signal generated by mast cells can reach the brain via both nerves and secreted mediators via systemic circulation. Therefore, the comment on line 314 should be reorganized.
- 6- The author must re-draw figure 4f. Please use molecules and receptors from this finding of manuscript.
- 7- L144 Which peptide specific is antigen specific IgE ?
- 8- FIG2. Separate figures should be prepared for experiments with 1% and 0.25% sucrose in FIG2d. The addition of alum control would be welcome for FIGa,b, and c. In addition, the p-value between +/+ and Cre/+ should be given in the experiment with 0.25% sucrose and it is necessary to prove that the experiment works as in the BALB/c mouse in FIG2d.
- 9- FIG3c-The authors propose that Nr4a1-GFP mice have increased numbers of mast cells in the stomach 3 hours after 25%OVA exposure. However, in FIG3c the p-value is not significant for stomach delta Nr4a1-GFP.
- 10- FIG4e -Alum control should be added to the experiment setup and should be shown in the figure.
- 11- Demonstrating the intracellular calcium increase in intestinal cells would be good in terms of

showing direct activation.

Referee #2 (Remarks to the Author):

In this manuscript, Plum et al. establish an automated behavioral assay that can track many animals in their natural home cage settings and monitor their preference for allergen-water (including OVA) versus water. They demonstrate avoidance behavior of the allergen-water requires mast cells and IgE. The authors analyze the gut phenotypes of the mast cell-deficient animals and show that they are not different from non-sensitized control animals. They use this absence of inflammatory phenotype to argue for the protective role of avoidance behaviors. However, this conclusion is based on correlative observation of negative data. To make such a conclusion, the authors should first show that continued consumption of allergen-water (as in mast cell-deficient background) promotes gut inflammation. Using calcium imaging, vagotomy, and pharmacologic means, the authors also show that the avoidance behaviors are not likely mediated by enteric neurons, the vagus nervous system, Trpv1+ peripheral nervous system neurons, or histamine. Lastly, they use pharmacology to block a serotonin receptor, which produced a statistically non-significant effect in blocking avoidance behavior. The study primarily reports negative results and lacks mechanistic explanations for the behavioral phenotype. To support the publication of this work, this reviewer believes that the authors need to provide positive results: e.g., identification of effectors downstream of mast cells in promoting avoidance behaviors. Otherwise, although the authors' behavioral quantifications are noteworthy, the study remains premature.

Referee #3 (Remarks to the Author):

A. Summary of the key results.

In studies of two inbred mouse strains (C57BL/6 and Balb/c Cpa3 +/+ mice) and the corresponding mast cell-deficient Cpa3 Cre/+ mice, the authors demonstrate that antigen-specific (OVA-specific) avoidance behavior in such mice is importantly dependent on mast cells. Evidence is presented that antigen avoidance is promoted by Th2 cytokines during immunization and by IgE in the execution phase. In this system, rapid antigen sensing was shown for mast cells in the stomach and small intestine but not in the oral cavity. Detailed studies of potential signaling between mast cells and the brain were done. These showed only that serotonin partially inhibited avoidance, but interruption of no single pathway completely eliminated avoidance. The authors concluded that "mast cells are predestined to recognize antigen immediately after ingestion, and to signal termination of antigen intake."

B. Originality and significance: if not novel, please include reference.

The key role of IgE in mediating acquired protection against the adverse effects, including death, of toxin exposure already has been reported in two papers: (ref. 43 [Immunity **39**, 963-975, 2013] and ref. 44 [Immunity **39**, 976-985, 2013]). The former paper also showed, in Fig. 6F-H, that passive immunization with serum from bee venom-immunized wild type mice failed to increase the resistance against challenge with a potentially lethal dose of bee venom in two different types of

mast cell-deficient mice. Accordingly, this paper should be cited as showing both the key role of IgE in mediating resistance to potentially lethal injections of bee venom and also for providing evidence concerning the importance of mast cells in mediating such IgE-dependent resistance.

However, the current paper uses IgE responses to a commonly employed antigen (OVA) to provide evidence that such IgE-dependent sensing of a presumably innocuous food allergen (OVA) can promote aversive behavior. The evidence that this response was partially dependent on serotonin (via one type of serotonin receptor: 5-HT₃) is of interest. The use of sophisticated methods to reduce the development of spurious results, including special cages (IntelliCages) for housing the mice, is appropriate. Finally, the authors comment at the end of the Discussion on the potential involvement of GDF15 in this response (they indicate that they were in communication with R. Medzhitov during the preparation of their manuscript).

In contrast to the Medzhitov manuscript, the Rodewald manuscript used OVA admixed with sucrose. Therefore, the two manuscripts reached somewhat similar conclusions using either OVA alone (Medzhitov group) or OVA plus sucrose (Rodewald group).

C. Data & methodology: validity of approach, quality of data, quality of presentation.

In general, data & methodology were appropriate. The use of IntelliCages to ensure that fewer potential biases were introduced during the experiments was particularly appropriate.

D. The use of statistics and treatment of uncertainties seem appropriate.

E. Conclusions: robustness, validity, reliability.

The data shown appear to be quite robust. Furthermore, the use of basophil-deficient *Mcpt8-Cre* mice to show that basophils were not involved in the avoidance behavior was appropriate, given the fact that the *Cpa3-Cre/+* mice have a 40% reduction in basophils (in addition to their striking deficiency in mast cells).

F. Suggested improvements: experiments, data for possible revision.

1) The next to last sentence needs revision. In lines 429-430, the authors write: "The present work identifies them as sensor cells of the nervous system,....". It isn't clear how mast cells, of hematopoietic origin, qualify as being "of the nervous system". The meaning of this sentence should be clarified.

Minor points:

1) Line 140-142: I suggest writing: "None of these assays distinguished mast cell-deficient mice from their wild type littermates, indicating that *Cpa3 Cre/+* mice had no behavioral deficits measured by these assays that could have confounded the drink avoidance experiments."

2) Lines 163-164: I suggest simply omitting: ", which may be testimony to the close relationship of basophils and mast cells in some tissues,".

3) Lines 381-382: "In light of this overall debatable role of mast cells in immunological protection," seems to be too general a statement, particularly since refs. 43 & 44 showed that IgE (ref. 43 & 44) and mast cells (ref. 43) could confer increased survival to mice injected with potentially lethal amounts of bee venom. I suggest the following text instead: "Given the complexities in

understanding the actual roles of mast cells in immunological protection, the role for mast cells observed here in antigen avoidance behavior is intriguing."

4) Lines 427-428: It seems a bit odd to cite only a paper concerned with human mast cells (ref. 32) when this manuscript was concerned solely with mouse mast cells. I suggest also citing a reference to the distinctive position of mouse mast cells among mouse hematopoietic cells (Dwyer DF, Barrett NA, Austen KF; Immunological Genome Project Consortium. Expression profiling of constitutive mast cells reveals a unique identity within the immune system. *Nat Immunol.* 2016 Jul;**17**(7):878-87.).

G. References: appropriate credit to previous work?

The key role of IgE in mediating acquired protection against the adverse effects, including death, of toxin exposure was reported in two of the cited papers: (ref. 43 [*Immunity* **39**, 963-975, 2013] and ref. 44 [*Immunity* **39**, 976-985, 2013]). However, the former paper also showed, in Fig. 6F-H, that passive immunization with serum from bee venom-immunized wild type mice failed to increase the resistance against challenge with a potentially lethal dose of bee venom in two different types of mast cell-deficient mice. Accordingly, this paper should be cited as showing both the key role of IgE in mediating resistance to potentially lethal injections of bee venom and also for providing evidence concerning the importance of mast cells in mediating such IgE-dependent resistance.

H. Clarity and context: lucidity of abstract/summary, appropriateness of abstract, introduction and conclusions.

The Abstract should be revised in line 36 to say: "Here we show that antigen-specific avoidance behavior in **inbred** mice....".

The authors also reported (lines 131-132): "...that a minor component of the OVA avoidance response is immunization-dependent and mast cell-independent (Fig. 1d). This is reflective of the detailed and careful approach used by the authors in the analysis of this complex behavior.

Referee #4 (Remarks to the Author):

Major

1. Mast Cell deficient mice may have secondary deficiencies; as such, to truly prove a role for mast cells, adoptive transfer of mast cells into the *cpa3* MC KO strain should be performed.
2. The passive sensitization experiment revealed only modest effect, in a fraction of the mice. This raises the question of whether the mast cells were sufficiently sensitized. The experiments are missing the positive control (allergen-induced anaphylactic response) as well as assessment of mast cell degranulation, on a single cell level. This is important as the investigators need to prove the role of IgE, and their data suggests that IgE-mediated mast cell activation (which may happen in their experiment), is not sufficient for the avoidance behaviour.

3. The experiments with C57Bl/6 mice (Figure 2), also need positive control (anaphylaxis) as per point above.
4. I have concerns about the experiments with the dorsal root ganglia. It is commendable that the authors indicate that the tracing experiments were derived from two mice; this is an insufficient number of mice to study, especially in view of the negative results. Related to this, the investigators, employed Wnt1|GCaMP3 engineered mice (plexus experiments); it remains unclear if similar negative results would be derived with other genetically engineered strains or best wild-type mice. This should be addressed.
5. The findings are very interesting, but the investigators are not providing a mechanism beyond mast cells. The link between mast cells and neurological pathways is purely speculative as they brain-axis experiments are largely negative. The single experiment with palonosetron does not prove peripheral afferent nerve involvement. Deeper experiments such as dose-response, use of additional inhibitors and/or KO mice, and ruling out off-target effects will be important.
6. The authors have not proven dependence on IgE.

Minor

1. Review of Figure 1 shows that some mice escape allergen avoidance. How is this explained? Do these mice exhibit intestinal pathology?
2. The investigators are implicating allergen activated stomach and intestinal mast cells, but not mast cells that reside in the oral cavity. They should also rule out a role for esophageal mast cells.
3. In view of the negative data in Figure 4, the authors are obliged to include additional controls. Please verify the KO at the protein level, including deficiency of histamine in the Hdc -/- mice.
4. Food allergic humans have 'ad-lib' diets, yet avoidance is not universal, such as in patients with eosinophilic esophagitis (that have abundant mastocytosis and food specific IgE). Please speculate on how this may occur.
5. The authors stated that they used the molecular signature of 200 inflammatory genes from the Libero et al. manuscript (cell systems, 2015). However, Liberzon et al. manuscript contains multiple data sets. Please explain from which dataset(s) the 200 genes were chosen and why these specific genes were chosen.
6. It is not clear why the genetic analysis of the 200 genes was performed only on the duodenum as the stomach, and small intestine demonstrated mast cell hyperplasia. Due to the differences between the organs, it is essential to perform the analysis on each organ separately. In addition, a full bulk RNA-sequencing will be more informative regarding the changes in the tissue, if any.
7. In the method- Immunization- the rationale for mixing the cytokines with their antibodies in one cocktail and injecting them into the mice is unclear.
8. Extended figure 8D- why did you stain for IgE? The lines in the text which refer to this figure are 346-347, which refer to mast cells and enteroendocrine cell 5-HTR expression, and IgE is not a marker for either of these. Please add the correct staining.
9. In extended figure 1, please add a legend for the grey and red marks.
10. Extended figure 3K- please make a readable table of the 200 genes you chose for the genetic analysis, as it is hard to read them in the heatmap.
11. Extended figure 3J- Why did only the duodenum in the representative figure as the stomach and small intestine demonstrate the mast cell hyperplasia.?

12. Extended Figures 4C and 4D- Is there an increased level of Mast cells in the Cre/+ group in the passive sensitized group compared to the unsensitized groups?

13. Figure 4C- indicate which color is what.

Author Rebuttals to Initial Comments:

Point-by-point response to the reviewers' comments (response in italics)

We are grateful to the reviewers for their insightful and very helpful comments. We feel that the manuscript has been significantly improved as a result of the reviewers' queries which importantly led us to make additional experiments. In the submitted revision, we now include comprehensive new data as requested and as described in detail below.

Reviewer #1:

We thank the reviewer for his/her positive and helpful comments. Please find below and in the manuscript the requested additional information and the requested control data.

1- Can egg white 8% sucrose solution affect behavior and metabolism in mice model? (PMID 27126968) please explain it in text.

Humans and mice have an innate preference for sweet solutions (PMID 25815979). Therefore, the 8% sucrose affects the behavior in creating a preference of mice for the sweetened egg white solution. Indeed, this preference behavior is a central element of the experiments because it makes mice prefer the sweetened egg white solution to study avoidance. The sucrose concentration was adjusted for the inbred strains used (BALB/c and C57BL/6).

Chronic consumption of sucrose over the span of several months can induce metabolic changes such as increased body weight, glucose intolerance, and adiposity (PMIDs 31255519, 15975159, 36229459, 35933635). We did not observe increased body weight in mice by the end of the IntelliCage experiments. This suggests that the short time frame of sugar consumption is insufficient to alter the metabolism of the mice.

We refer to this information in the Results (page 5, line 110): "The 8% sucrose affects the behavior in creating a preference of mice for the egg white water", and in Fig. 1 legend (page 35, line 1059): "The sugar consumption did not increase the body weight of the mice by the end of IntelliCage experiments."

2- In fig 2 Please explain type 2 cytokines enhances antigen avoidances ? Is it IL-4, IL-5, IL-9 or IL-13 ? or Please explain in legend and text IL-3/4/9 ? Is it cocktail ?

We thank the reviewer for pointing out this lack of clarity. We now provide information in the Results (page 9, line 216): "C57BL/6 mice were injected with a Th2- and mast cell-promoting cytokine cocktail (see Extended Data Fig. 3 legend) two days after each round of immunization (Methods)", in the Extended Data Fig. 3 legend (page 40, line 1194): "Mice received two intraperitoneal injections of a cytokine cocktail consisting of IL-3, anti-mouse IL-3, IL-4, anti-mouse IL-4 and IL-9 (cocktail abbreviated as IL-3/4/9) on days 2 and 16 prior to the avoidance test...", and in the Methods (page 19, line 529): "For Th2 cytokine treatment, mice were injected intraperitoneally with a cocktail composed of 2 µg IL-3 (Peprotech), 12 µg anti-mouse IL-3 (MP2-8F8, Biolegend), 2 µg IL-4 (Peprotech), 12 µg anti-mouse IL-4 (11B11, Biolegend) and 0.4 µg IL-9 (Peprotech) on days 2 and 16. Mixing IL-3 with the anti-IL-3 antibody MP2-8F8, and IL-4 with the anti-IL-4 antibody 11B11 generates cytokine-antibody complexes that display increased activity in vivo¹, which we exploited here to increase the magnitude and duration of cytokine effects in vivo. "

3- The authors must explain egg white is endotoxin free or not ? Please explain it.

We measured the endotoxin concentrations of 11 of our independent stock preparations of 20% egg white water that we used in our behavioral experiments from early 2020 to mid 2022 with the Limulus amoebocyte lysate test. Endotoxin concentrations of the egg white solutions were on

average 0.78 ± 0.17 EU/ml (see Figure 1 for reviewer below). This endotoxin level is considerably lower than the reported endotoxin concentration in standard mouse chow (approximately 20 EU/ μ g; PMID 18990206), or the median endotoxin concentration of commercially available pasteurized milk (102.5 EU/ml; PMID 25527628).

Fig. 1

Fig. 1: Endotoxin levels in egg white water

Measurement of endotoxin concentrations of 11 independent stock preparations of 20% egg white water used for behavioral experiments from early 2020 to mid 2022 with the Limulus ameobocyte lysate test. The bar represents the mean value.

We have revised the Methods accordingly (page 20, line 552): "The 20% egg white solutions used for experiments contained on average 0.78 ± 0.17 endotoxin units per ml, which is over 10-fold lower than standard mouse chow."

4- The authors must explain cFos expression in their expression.

To test mast cell-specific and allergen-induced cFos-expression in the central nucleus of amygdala (CeA), paraventricular nucleus (PVN), parabrachial nucleus (PBN), and nucleus tractus solitarius (NTS) of the brain, we immunized wild type (+/+) and mast cell-deficient (Cre/+) mice according to our standard regimen (Figure 2a for reviewer below). On day 21 mice were only given access to 20% egg white water containing 8% sucrose solution in place of their normal water bottle (Figure 2a for reviewer below). After three hours of voluntary drinking, mice were sacrificed, and the brains were analyzed for cFos-expression by immunohistochemistry. Mice were not given a choice in this experiment, and we observed reduced and variable egg white water intake in wild type mice compared to mast cell-deficient mice (Figure 2b for reviewer below) which indicates initial antigen avoidance. At this time point (3h) and under these conditions, we did not observe increased cFos-expression in wildtype mice compared to mast cell-deficient mice (Figure 2c-f for reviewer below). In fact, the number of cFos+ neurons in the CeA was increased in mast cell-deficient compared to wild type mice (see Figure 2c for reviewer below). Mast cell-deficient mice which consumed more egg-white sugar solution may have increased palatability (sugar and protein) signaling (PMID 22815514). In any case, these experiments did not further illuminate cFos signals in brain areas associated with avoidance.

Fig. 2

Fig. 2: cFos-expression in the brain after antigen consumption

a, Type 2 immunization scheme, and experimental timeline for analysis of cFos induction. **b**, On day 21 mice were given 20% egg white water containing 8% sucrose in place of their regular drinking bottle, and total egg white water intake of OVA-alum immunized BALB/c Cpa3^{+/+} and Cpa3^{Cre/+} mice was recorded after three hours. **c, d, e, f**, Mice were perfused and the brains were isolated and processed for immunohistochemistry staining with an antibody specific for cFos. Shown is the quantification of cFos-positive neurons in the central nucleus of the amygdala (CeA) (**d**), the paraventricular nucleus (PVN) (**e**), the parabrachial nucleus (PBN) (**f**), and the nucleus tractus solitarius (NTS) (**g**). The bars represent the mean values, and each dot is a single mouse. Statistical analysis was performed using students T-test (**b–g**). The exact P values are shown.

5- It would be proper to say that vagal and trpv1 neurons are not involved in this pathway since the signal generated by mast cells can reach the brain via both nerves and secreted mediators via systemic circulation. Therefore, the comment on line 314 should be reorganized.

We apologize for lack of clarity here. We have reorganized this text to read (page 14, line 375): “The fast avoidance reaction of some immunized mice (29% of wild type mice within the first day, and some immediately after their first licks) (Extended Data Fig. 9a, b) could be due to rapidly acting humoral factors in the blood circulation or to direct signaling from mast cells to neurons. [...] Avoidance signaling via extrinsic vagal neurons was assessed by sub-diaphragmatic vagotomy of immunized mice (Extended Data 10g). In addition, we tested the effect of Resiniferatoxin (RTX)-mediated depletion of extrinsic Trpv1-expressing vagal- and dorsal root ganglion sensory neurons on antigen avoidance (Extended Data Fig. 10i, j). None of these experiments revealed evidence for a neuronal route transmitting the avoidance signal (Extended Data Fig. 10b-j). However, since Trpv1-expressing (RTX-sensitive) neurons represent only a subset of all dorsal root ganglion neurons, a function for dorsal root ganglion neurons, that are insensitive to RTX ablation, in signaling antigen avoidance cannot be ruled out.”

6- The author must re-draw figure 4f. Please use molecules and receptors from this finding of manuscript.

We have redrawn this figure (now Figure 5), and indicate, among other changes, IgE and 5-lipoxygenase activating protein (FLAP). We also added our new findings on immune activation and inflammation under non-avoidance conditions.

7- L144 Which peptide specific is antigen specific IgE ?

The IgE-antibody used for passive sensitization is the clone E-C1 (also known as OE-1) purchased from Chondrex Inc. This antibody has been raised against full length OVA protein, and possibly recognizes a repetitive epitope (https://www.chondrex.com/documents/3006-3008_IgE_Animal_Model.pdf). To our knowledge, the epitope has not been mapped.

8- FIG2. Separate figures should be prepared for experiments with 1% and 0.25% sucrose in FIG2d. The addition of alum control would be welcome for FIGa,b, and c. In addition, the p-value between ^{+/+} and Cre/+ should be given in the experiment with 0.25% sucrose and it is necessary to prove that the experiment works as in the BALB/c mouse in FIG2d.

The revised data from former Fig. 2 is now shown in Extended Data Fig. 3. We now present separate figures for 1% sucrose (panel d), 0.25% sucrose (panel e), and 0.25% sucrose with cytokine cocktail (panel f). p-values are given. The C57BL/6 alum controls are shown in

Extended Data Fig. 3a-f. We refer to these experiments in the results (page 8, line 213): "As in BALB/c mice (Fig. 1c-e), antigen avoidance was not observed in alum only immunized C57BL/6 mice (Extended Data Fig. 3a, b, d, e)."...."This treatment further enhanced the avoidance response which, however, remained mast cell- and OVA immunization-dependent (Extended Data Fig. 3c, f)."

9- FIG3c-The authors propose that Nr4a1-GFP mice have increased numbers of mast cells in the stomach 3 hours after 25% OVA exposure. However, in FIG3c the p-value is not significant for stomach delta Nr4a1-GFP.

The original Figure 3c showed the difference (delta) in Nr4a1-GFP expression in tissue mast cells of OVA-alum or alum immunized mice challenged by OVA or BSA. We chose this comparison for normalization because we noticed differences in baseline GFP fluorescence in mast cells derived from different tissues. We now show mean fluorescence intensities for mast cells from each organ. p-value for stomach mast cells is significant (Figure 4c).

10- FIG4e -Alum control should be added to the experiment setup and should be shown in the figure.

We repeated this experiment, and included alum controls (Extended Data Fig. 8g). We initially observed a partial response of OVA-alum immunized mice towards higher egg white water consumption after palonosetron treatment compared to vehicle controls. To more definitively clarify the role of 5-HTR3a in antigen avoidance behavior, we consulted with statisticians, increased the number of tested mice to n = 21 for vehicle and n = 23 for palonosetron, and included the requested alum controls in these new experiments (Extended Data Fig. 8g). The data do not confirm an increased egg white water consumption after 5-HTR3a blockade. In addition, palonosetron-treatment altered the egg white water preference of alum immunized mice (Extended Data Fig. 8g), suggesting complex involvement of 5-HTR3a in antigen avoidance.

In summary, it is unlikely that 5-HTR3a has a major role in antigen-avoidance behavior. For clarity, we have removed the mast cell-deficient groups from the figure (Extended Data Fig. 8g), and have changed the text on page 13, line 365 to read: "Hence, stomach mast cells are loaded with 5-HT which upon release may signal via its ionotropic serotonin 3 receptor (5-hydroxytryptamine receptor 3; 5-HTR3), which is a key regulator of visceral malaise, nausea and emesis. We tested the role of 5-HTR3 in antigen avoidance by treatment of mice with the specific inhibitor palonosetron. Palonosetron did not significantly decrease antigen avoidance behavior (Extended Data Fig. 8g)."

11- Demonstrating the intracellular calcium increase in intestinal cells would be good in terms of showing direct activation.

We thank the reviewer for this interesting suggestion. In new experiments we attempted to measure, as proposed, the intracellular calcium increase in intestinal mast cells. The frequency of mast cells in the intestine epithelial cell fraction is much lower than in the stomach, therefore it was necessary to first enrich mast cells by Percoll gradient centrifugation. The subsequent protocol was performed as described for stomach mast cells (Figure 4d, e). However, such isolated intestinal mast cells did not increase intracellular calcium levels during antigen-stimulation, albeit ionomycin-induced calcium increases were apparent (see Figure 3 for reviewer below). Thus, we cannot distinguish whether intestinal epithelial mast cells are insensitive to ex-vivo antigen stimulation or whether the cells became refractory due to the gradient purification.

Fig. 3

Fig. 3: Freshly isolated intestinal mast cells do not respond to OVA in vitro by intracellular Ca^{2+} flux.

a, Representative traces of Fluo-4 fluorescence (indicating intracellular Ca^{2+} concentration) in small intestinal intraepithelial mast cells from OVA-alum or alum immunized mice. Cells were stimulated consecutively with OVA and ionomycin at the indicated times (arrows). **b**, Quantification of Fluo-4 mean fluorescence intensity (as in a), normalized to the first 30 seconds of measurement (set as 1.0), comparing mast cells from OVA-alum and alum immunized mice (OVA-alum $n = 7$; alum $n = 6$).

Reviewer #2

We thank the reviewer for his/her insightful comments. We addressed the key requests to demonstrate an inflammatory gut phenotype following continuous allergen intake, and to provide data on effectors downstream from mast cells. We now provide new data from comprehensive experiments on the first point (Fig. 3; Extended Data Fig. 4; Extended Data Fig. 6, Extended Data Fig. 7), and experiments inhibiting leukotriene synthesis (Fig. 4).

The authors analyze the gut phenotypes of the mast cell-deficient animals and show that they are not different from non-sensitized control animals. They use this absence of inflammatory phenotype to argue for the protective role of avoidance behaviors. However, this conclusion is based on correlative observation of negative data. To make such a conclusion, the authors should first show that continued consumption of allergen-water (as in mast cell-deficient background) promotes gut inflammation.

We agree that this is a central aspect of this work that was only indirectly addressed in the first version. The reviewer asks whether continued consumption of allergen-water (as in mast cell-deficient background) promotes gut inflammation. While we have done this (see below), this experiment is confounded by the possibility that the absence of mast cells may ameliorate the development of gut inflammation (PMID 14660743). Wild type mice, on the other hand, avoid continued consumption of allergen-water when given the choice, and drink less and variable amounts when not given the choice (see Figure 4a, b for reviewer below). A comparison of the antigen amounts for wild type mice given the choice, or treated with gavage, and mast cell deficient mice under continued consumption of allergen-water are shown in Figure 4c for reviewer below. This comparison shows that the amount of antigen given by gavage is modest, and in fact considerably lower than the amount of antigen consumed voluntarily by mast cell-deficient mice.

Fig. 4

Fig. 4: Antigen doses in various experimental conditions. a, Type 2 immunization scheme, and experimental timeline for consumption of egg white water only. b, On day 21 mice were given 20% egg white water containing 8% sucrose in place of their regular drinking bottle, and total egg white water intake of immunized BALB/c Cpa3^{+/+} and Cpa3^{Cre/+} mice was

recorded after three hours. Immunized wild type mice given only egg white water (allergen-water) immediately drank less than immunized mast cell-deficient mice. This resulted in a trend towards reduced OVA consumption (average = 0.5g difference) and a strongly increased standard deviation in wild type mice (0.708g) compared to mast cell-deficient mice (0.346g). **c**, Estimated average OVA intake of immunized BALB/c Cpa3^{+/+} and Cpa3^{Cre/+} mice per day over the course of the IntelliCage experiment (in Fig. 1). For comparison, OVA amounts given to immunized mice by gavage is shown. Avoiding wild type mice consumed between 1.5 and 36 mg OVA, and mast cell-deficient mice between 380 and 1340 mg OVA. We administered 50 mg OVA per gavage (8 times over 16 days).

Regarding the new experiments, we studied consequences of continued antigen uptake via oral gavage in wild type mice, and of continued consumption of egg white water in mast cell deficient mice (outlined in Extended Data Fig. 4a). To briefly summarize these experiments, we analyzed stomach and small intestine by flow cytometry for inflammatory cells (Extended Data Fig. 4d, e), and by RNA sequencing (Fig. 3; Extended Data Fig. 6; Extended Data Fig. 7, Supplementary Data Tables 1-4) of tissue lysates which captured changes in gene expression in an unbiased manner, i.e. without restricting analyses to cell subsets. We also analyzed serum cytokines (Extended Data Fig. 4f, g). All of these approaches revealed broad immune activation and inflammation in immunized wild type mice. In mast cell-deficient mice that were drinking antigen voluntarily, we also found an induction of immune response genes, however, its magnitude was reduced and its kinetics altered (Extended Data Fig. 7). These data suggest that the immunological and inflammatory response that is prevented by mast cell-mediated antigen avoidance behavior is largely but not exclusively driven by mast cells. Nonetheless, the observed immune activation and the inflammatory phenotype argue in our view for the protective role of avoidance behaviors.

These experiments are described in a new paragraph "Antigen avoidance behavior prevents immune activation and inflammation" in the main text page 9, line 235. Immune activation in non-avoiding mast cell-deficient mice is presented in a new paragraph on page 11, line 301.

Finally, we note that the comparison of plain water (no antigen), free choice (only voluntary amount of antigen) and gavage (defined larger amount of antigen) was key to uncover gene expression profiles for each condition (Fig. 3). These data may also define a threshold for the amount of voluntary antigen uptake.

To support the publication of this work, this reviewer believes that the authors need to provide positive results: e.g., identification of effectors downstream of mast cells in promoting avoidance behaviors.

In further search for effectors, we addressed the role of lipid mediators known to be released from IgE-activated mast cells. We tested FLAP-dependent leukotrienes (Fig. 4f) in antigen avoidance by pharmacologically inhibiting FLAP using the specific inhibitor MK-886. To this end, BALB/c wild type mice were alum- or OVA-alum immunized, and treated, one hour before the avoidance test, with MK-886. FLAP-inhibition significantly reduced avoidance. The results have been added to a paragraph "Antigen avoidance behavior depends on 5-lipoxygenase-activating protein (FLAP)" beginning on page 13, line 336. The data are shown in Fig. 4g, h.

Reviewer #3:

We thank the reviewers for their very positive and insightful comments, the suggested text changes, and the helpful discussion on protective roles of IgE.

1) The next to last sentence needs revision. In lines 429-430, the authors write: "The present work identifies them as sensor cells of the nervous system,....". It isn't clear how mast cells, of hematopoietic origin, qualify as being "of the nervous system". The meaning of this sentence should be clarified.

We clarified this sentence to read (page 17, line 472): "The present work identifies them as sensor cells linking antigen-recognition elicited by type 2 immune responses to behavior.

Minor points:

1) Line 140-142: I suggest writing: "None of these assays distinguished mast cell-deficient mice from their wild type littermates, indicating that Cpa3 Cre/+ mice had no behavioral deficits measured by these assays that could have confounded the drink avoidance experiments."

The sentence has been changed accordingly (page 6, line 150).

2) Lines 163-164: I suggest simply omitting: ", which may be testimony to the close relationship of basophils and mast cells in some tissues."

This part of the sentence has been deleted.

3) Lines 381-382: "In light of this overall debatable role of mast cells in immunological protection," seems to be too general a statement, particularly since refs. 43 & 44 showed that IgE (ref. 43 & 44) and mast cells (ref. 43) could confer increased survival to mice injected with potentially lethal amounts of bee venom. I suggest the following text instead: "Given the complexities in understanding the actual roles of mast cells in immunological protection, the role for mast cells observed here in antigen avoidance behavior is intriguing."

The sentence has been changed as proposed (page 16, line 427).

4) Lines 427-428: It seems a bit odd to cite only a paper concerned with human mast cells (ref. 32) when this manuscript was concerned solely with mouse mast cells. I suggest also citing a reference to the distinctive position of mouse mast cells among mouse hematopoietic cells (Dwyer DF, Barrett NA, Austen KF; Immunological Genome Project Consortium. Expression profiling of constitutive mast cells reveals a unique identity within the immune system. *Nat Immunol.* 2016 Jul;17(7):878-87.).

We thank the reviewers for pointing out this shortcoming. The Dwyer et al. reference is now cited (page 17, line 471)

G. References: appropriate credit to previous work?

The key role of IgE in mediating acquired protection against the adverse effects, including death, of toxin exposure was reported in two of the cited papers: (ref. 43 [*Immunity* 39, 963-975, 2013] and ref. 44 [*Immunity* 39, 976-985, 2013]). However, the former paper also showed, in Fig. 6F-H, that passive immunization with serum from bee venom-immunized wild type mice failed to increase the resistance against challenge with a potentially lethal dose of bee venom in two different types of mast cell-deficient mice. Accordingly, this paper should be cited as showing both the key role of IgE in mediating resistance to potentially lethal injections of bee venom

and also for providing evidence concerning the importance of mast cells in mediating such IgE-dependent resistance.

We agree, of course, that appropriate credit is due to previous work, and thank the reviewers for referring to the important specific findings regarding protective roles of mast cells shown in ref 45 (Marichal et al Immunity 2013). We have revised this sentence (page 15, line 423), and hope that this specific findings for the role mast cells in ref 45 will satisfy the reviewers' request.

H. Clarity and context: lucidity of abstract/summary, appropriateness of abstract, introduction and conclusions.

The Abstract should be revised in line 36 to say: "Here we show that antigen-specific avoidance behavior in inbred mice....".

We inserted "inbred" into the sentence (page 3, line 44).

Reviewer #4:

We thank the reviewer for his/her excellent and insightful suggestion to clarify the role of IgE in antigen-avoidance, and to interrogate immunological changes in stomach RNA-expression following antigen intake. We have now made new experiments with IgE-deficient mice and performed sequencing experiments to address these point directly as described below. Towards identifying a mechanism downstream of mast cells, we provide evidence for a role of leukotrienes, mediators known to be released from IgE-activated mast cells, in antigen avoidance behavior.

Major

1. Mast Cell deficient mice may have secondary deficiencies; as such, to truly prove a role for mast cells, adoptive transfer of mast cells into the *Cpa3* MC KO strain should be performed.

*The only known secondary deficiency of *Cpa3Cre* mice is a reduction in basophils (PMIDs 22101159; 27411001), which we addressed in this manuscript by showing OVA-avoidance in basophil-deficient *Mcpt8-Cre* mice (Extended Data Fig. 2a-h).*

Moreover, our conclusion that antigen-specific avoidance behavior in inbred mice is critically dependent on mast cells is now additionally supported by our new data showing an essential role for IgE in antigen avoidance behavior (Fig. 2a-c), given that mast cells are the major effector cells for IgE in antigen-exposed tissues.

*The reviewer requests adoptive transfer of mast cells into *Cpa3-Cre* mice. However, this approach suffers from several shortcomings: Adoptive transfer of in vitro-derived mast cells (BMMC) does not reconstitute a normal physiologic distribution of mast cells, resulting in a higher than normal concentration of mast cells in the stomach (PMIDs 16127161, 8613053, and 12217411), and a lower than normal concentration of mast cells in intestines (PMID 24516385, 24416383, and 28264908). Second, the phenotypes of adoptively transferred mast cells have been reported to differs from the corresponding native mast cell populations (PMID 15771585, 25727288, and 23127755). Finally, adoptively transferred mast cells may undergo phenotypic changes that are not fully consistent with their anatomical sites (e.g. protease expression pattern) (PMID 11337367). In a recent review from Stephen J Galli, Nicolas Gaudenzio, and Mindy Tsai in *Annu Rev Immunol* ('Mast Cells in Inflammation and Disease: Recent Progress and Ongoing Concerns') (PMID 32340580) the authors summarized these pitfalls and raised a note of caution in the area of adoptive transfer of mast cells. We do share this conclusion, and have hence refrained from the suggested adoptive transfer of BMMC into *Cpa3-Cre* mice.*

2. The passive sensitization experiment revealed only modest effect, in a fraction of the mice. This raises the question of whether the mast cells were sufficiently sensitized. The experiments are missing the positive control (allergen-induced anaphylactic response) as well as assessment of mast cell degranulation, on a single cell level. This is important as the investigators need to prove the role of IgE, and their data suggests that IgE-mediated mast cell activation (which may happen in their experiment), is not sufficient for the avoidance behaviour.

The essential role of IgE has now been demonstrated (see response to point 6 below). We agree that passive transfer of IgE only partially led to gain of avoidance (Fig. 2f, g). The reviewer raises the interesting question whether mice sensitized for avoidance are also sensitized for anaphylaxis. We injected BALB/c mice with IgE monoclonal antibody E-C1 (as in Fig. 2f, g), and measured passive systemic anaphylaxis following antigen challenge by OVA gavage.

Compared to OVA-alum immunized BALB/c mice OVA gavage could not induce a temperature drop in passively IgE-sensitized mice (Fig. 2l). As requested by the reviewer, we also analyzed mast cell degranulation on a single cell level based CD63 expression (PMID 34233046). In line with the partial avoidance response, only half of the mice showed activation of stomach mast cells upon OVA contact (Fig. 2j), and none of the mice showed activation in small intestinal epithelial mast cells (Fig. 2k). We conclude that IgE transfer can partially induce avoidance behavior, however, immunization is required to attain full mast cell activation and avoidance.

We present and consider these findings in the results on page 8, line 195.

3. The experiments with C57Bl/6 mice (Figure 2), also need positive control (anaphylaxis) as per point above.

In response to this request, we immunized C57BL/6 mice (bearing the Nr4a1-GFP reporter) with OVA-alum as in the avoidance experiments, followed by anaphylaxis assay. Mice were challenged by OVA or BSA gavage, or by intravenous injection of OVA or BSA (Extended Data Fig. 3j). Only intravenous injection of OVA induced a temperature drop (Extended Data Fig. 3m). Analysis of mast cell activation on the single cell level (GFP-expression) by flow cytometry revealed increased GFP expression of stomach (Extended Data Fig. 3k) and intestinal mast cells (Extended Data Fig. 3l) in mice receiving OVA-gavage compared to BSA-controls, albeit reaching statistical significance only for intestinal mast cells.

We present these findings in the results on page 9, line 226.

Hence, type2 immunization in C57BL/6 mice can lead to gastro-intestinal mast cell activation (Extended Data Fig. 3k, l) and avoidance (Extended Data Fig. 3b, e) without sensitization for anaphylaxis by gavage (Extended Data Fig. 3m). This separation was not evident in BALB/c mice (see point 2 above).

4. I have concerns about the experiments with the dorsal root ganglia. It is commendable that the authors indicate that the tracing experiments were derived from two mice; this is an insufficient number of mice to study, especially in view of the negative results. Related to this, the investigators, employed Wnt1|GCaMP3 engineered mice (plexus experiments); it remains unclear if similar negative results would be derived with other genetically engineered strains or best wild-type mice. This should be addressed.

We have now analyzed a total of $n = 5$ alum immunized Wnt1|GCaMP3 mice, and $n = 5$ OVA-alum immunized Wnt1|GCaMP3 mice. Repetition confirmed the first data set. We also modified the way the data are displayed. Instead of snap shots of recorded ganglia, we now show representative calcium traces of individual neurons (Extended Data Fig. 10b, d). The new total number of recorded submucosal plexus neurons is $n = 178$ for alum, and $n = 188$ for OVA-alum. The new number of recorded myenteric plexus neurons is $n = 442$ for alum, and $n = 419$ for OVA-alum.

Information on numbers of mice and neurons are provided in the legend of Extended Data Fig. 10 (page 43, lines 1292).

Regarding the model, the Wnt1|GCaMP mouse is widely used and well established in studies of the enteric nervous system with the advantage of having broad expression of the calcium

indicator in all enteric neurons throughout the GI tract. Some studies have used different promoters to drive GCaMP expression in targeted subpopulations of enteric neurons (e.g. nNOS-GCaMP3, nitrergic neurons; and ChAT-GCaMP3, cholinergic neurons). However, to our knowledge there are no reported differences between these genetically engineered strains (where GCaMP expression is driven by different promoters) that could impact gut physiology or potentially influence the results. We do report similar results using BALB/c wild type and mast cell-deficient mice when the calcium sensor GCaMP6 was transduced using a viral vector construct (AAV9-CaMKII-GCaMP6; Extended Data Fig. 10f). However, using this approach, the viral vector only transduced GCaMP6 expression in a fraction of enteric neurons (mostly myenteric plexus neurons). Again, this highlights the advantage of using the transgenic Wnt1|GCaMP mouse model for pan-neuronal expression of the reporter.

5. The findings are very interesting, but the investigators are not providing a mechanism beyond mast cells. The link between mast cells and neurological pathways is purely speculative as the brain-axis experiments are largely negative. The single experiment with palonosetron does not prove peripheral afferent nerve involvement. Deeper experiments such as dose-response, use of additional inhibitors and/or KO mice, and ruling out off-target effects will be important.

We thank the reviewer for his/her interest in our work, and we agree, of course, that a mechanism beyond mast cells is desirable. Towards a mechanism, we have performed several new experiments. First, we demonstrate that antigen-avoidance behavior is fully IgE-dependent (using IgE-deficient mice; see below under point 6). Regarding serotonin, we initially observed a partial response of OVA-alum immunized mice towards higher egg white water consumption after palonosetron treatment compared to vehicle controls. To more definitively clarify the role of 5-HTR3a in antigen avoidance behavior, we consulted with statisticians, increased the number of tested mice to $n = 21$ for vehicle and $n = 23$ for palonosetron, and included the requested alum controls in these new experiments (Extended Data Fig. 8g). The data do not confirm an increased egg white water consumption after 5-HTR3a blockade. In addition, palonosetron-treatment altered the egg white water preference of alum immunized mice (Extended Data Fig. 8g), suggesting complex involvement of 5-HTR3a in antigen avoidance. In summary, it is unlikely that 5-HTR3a has a major role in antigen-avoidance behavior. For clarity, we have removed the mast cell-deficient groups from the figure (Extended Data Fig. 8g), and have changed the text on page 13, line 361.

In further search for effectors, we addressed the role of lipid mediators known to be released from IgE-activated mast cells. We tested FLAP-dependent leukotrienes (Fig. 4f) in antigen avoidance by pharmacologically inhibiting FLAP using the specific inhibitor MK-886. To this end, BALB/c wild type mice were alum- or OVA-alum immunized, and treated, one hour before the avoidance test, with MK-886. FLAP-inhibition significantly reduced avoidance (Fig. 4g, h). We interpret these results cautiously, given that not all mice responded to the inhibitor, and given that the effect was mostly seen at early timepoints.

These experiments are presented in a new paragraph in the results section (page 13, line 336). We also discuss these findings (page 16, line 447) in the context of the work on GDF-15 by Florsheim et al. (PMID 36712030).

We acknowledge that our data regarding the link between mast cells and neurological pathways are negative. However, we still consider our comprehensive analyses useful. We excluded in KO mice roles for major mast cell mediators, including histamine and several proteases (including Mcpt6), which can directly stimulate sensory neurons (PMIDs 27793571; 12388180; 7810655). Moreover, we analyzed major branches of potential gut-brain signaling pathways (enteric neurons, vagal neurons, and RTX-sensitive neurons). Taken together, we and others

may build on this negative data to identify the mast cell- and antigen-responsive neuron populations, if any.

6. The authors have not proven dependence on IgE.

We analyzed IgE-deficient (Igh-7^{-/-}) mice, and found that antigen avoidance is critically dependent on IgE (new data in Fig. 2a-c). We describe these data in a new paragraph on page 7, line 178.

Minor

1. Review of Figure 1 shows that some mice escape allergen avoidance. How is this explained? Do these mice exhibit intestinal pathology?

As shown in Fig. 1c, all immunized wild type animals avoided antigen, albeit with variable kinetics. Therefore, we find no evidence for escape.

The question regarding pathology is important. We studied immune activation and inflammation under conditions of avoidance versus non-avoidance (experiments outlined in Extended Data Fig. 4a; new data in Fig. 3; Extended Data Fig. 6; Extended Data Fig. 7, Supplementary Tables 1-4). We found broad immune activation and inflammation in immunized wild type mice under non-avoidance conditions (Fig. 3a, b, d, f). We also compared plain water (no antigen) to free choice (only voluntary amount of antigen), and identified mild gene expression changes that may define a threshold for the amount of voluntary antigen uptake (Fig. 3c, e). By contrast, the differences avoidance versus non-avoidance were drastic (Fig. 3a, b, d, f).

These experiments are described in a new paragraph "Antigen avoidance behavior prevents immune activation and inflammation" in the main text page 9, line 235.

2. The investigators are implicating allergen activated stomach and intestinal mast cells, but not mast cells that reside in the oral cavity. They should also rule out a role for esophageal mast cells.

To address this point, we prepared single cell suspensions from esophageal and gingival tissues of OVA-alum immunized mice. Tissues were analyzed by flow cytometry for CD45⁺ CD11b⁻ MHCII⁻ cells expressing Kit and FcεRI (i.e. mast cells). This revealed a small population of mast cells in the gingiva (0.51% of live CD45⁺ CD11b⁻ MHCII⁻ cells), whereas mast cells were undetectable in the esophagus of mice (see Fig. 5 for reviewer below).

Fig. 5**Fig. 5: Analysis of mast cells in the esophagus**

Esophagus (upper panels) and gingiva (positive control, lower panels) of immunized BALB/c Cpa3^{+/+} mice were digested and single cell suspensions were prepared, as described in the methods section of the main manuscript. Cells were stained with fluorescent antibodies and mast cells were analyzed by flow cytometry. Under these conditions, gingiva cell suspension contained 0.51% mast cells within the live CD45⁺ CD11b⁺ MHCII⁻ gate, whereas mast cells were undetectable among live CD45⁺ CD11b⁺ MHCII⁻ cells from the esophagus.

We now refer to esophagus on page X: "We also analyzed esophagus and colon but detected no (esophagus), or only minute numbers (colon) of mast cells which precluded their further analysis (not shown)."

3. In view of the negative data in Figure 4, the authors are obliged to include additional controls. Please verify the KO at the protein level, including deficiency of histamine in the Hdc ^{-/-} mice.

Data on loss of protein or loss of function (KO), or gain of function (Cpa3-Cre^{+/+} and Mcpt8-Cre) have been published for all mutants used in this study. We include in Fig. 6 for reviewer below genomic PCR data from our own laboratory on all mutant mouse strains involved.

Fig. 6
Fig. 6: Genotyping PCRs of KO and transgenic strains used in this study

Mutant mice of the indicated strains were identified by genomic PCR amplification and gel agarose electrophoresis. Mutant strains were: Mc-cpa^{Y356L,E378A} (Schneider et al., 2007); Cpa3^{-/-} (Feyerabend et al., 2005); Mcpt6^{-/-} (Shin et al., 2008); Hdc^{-/-} (Ohtsu et al., 2001); Igh7^{-/-} (Oettgen et al., 1994); Mcpt8-Cre (Ohnmacht et al., 2010); and Cpa3^{Cre/+} (Feyerabend et al., 2011). The used oligonucleotide pairs were taken from the cited articles. Control PCR for Mcpt8-Cre was specific for beta-2microglobulin.

Lack of Cpa3 activity in Mc-cpa^{Y356L,E378A} mice has been reported in Figure 3e of Schneider et al., 2007 (PMID 17923505). Western blot analysis proving lack of Cpa3 and Mcpt5 in Cpa3^{-/-} mice has been reported in Figure 1d of Feyerabend et al., 2005 (PMID 15988029). Western blot analysis proving lack of Mcpt6 in Mcpt6^{-/-} mice has been reported in Figure 2a of Shin et al., 2008 (PMID 18354212). Histamine deficiency in brain, skin, stomach, spleen, kidney and plasma of Hdc^{-/-} mice is reported in table 1 of Ohtsu et al., 2001 (PMID 11478947). Lack of IgE antibodies in Igh7^{-/-} mice is reported in Figure 2b of Oettgen et al., 1994 (PMID 8047141). Lack of basophils in Mcpt8-Cre animals has been reported in Figure 1b of Ohnmacht et al., 2010 (PMID 20817571). Lack of Mast cells in Cpa3^{Cre} animals has been reported in Figure 1b of Feyerabend et al., 2011 (PMID 22101159).

4. Food allergic humans have ‘ad-lib’ diets, yet avoidance is not universal, such as in patients with eosinophilic esophagitis (that have abundant mastocytosis and food specific IgE). Please speculate on how this may occur.

In eosinophilic esophagitis (EoE) it is known that “food allergens (e.g., milk, egg, wheat, soy) are well-established disease triggers, with dietary elimination of specific food allergen triggers or elemental diet therapy resulting in disease remission in a majority of subjects” (PMID 36351516). Further, the American Academy of Allergy, Asthma & Immunology states: “A person with EoE may have one or more foods triggering their EoE. Once the causative food(s) is (are) identified and removed from a person’s diet, esophageal inflammation and symptoms generally improve in a few weeks” (<https://www.aaaai.org/conditions-treatments/related-conditions/eosinophilic-esophagitis>). Hence, as long as the antigen has not been identified, patients may not be aware of the causative food component, and cannot avoid this food. Once identified, as in the prominent case of peanut allergy, patients do avoid consumption of peanuts: “The cornerstone of management of patients with nut allergy has been avoidance of the incriminated nut as well as other potentially related nuts” (PMIDs 34150446; 12517578).

Our understanding is that, in general, allergen avoidance is considered a mainstay for allergy including EoE management.

5. The authors stated that they used the molecular signature of 200 inflammatory genes from the Libero et al. manuscript (cell systems, 2015). However, Liberzon et al. manuscript contains multiple data sets. Please explain from which dataset(s) the 200 genes were chosen and why these specific genes were chosen.

Liberzon et al. developed 50 hallmark datasets, encompassing broad biological functions, not only immunology, by both computational approaches based on founder gene sets, expert curation and validation of robustness by test datasets. For all 50, see Table 1 in the Liberzon et al. reference (PMID 26771021). We chose the category 'Inflammatory Response'

http://www.gsea-msigdb.org/gsea/msigdb/cards/HALLMARK_INFLAMMATORY_RESPONSE.html),

because it contains 200 inflammation-specific genes which would allow us to objectively probe our datasets for signs of inflammation comparing avoidance and non-avoidance conditions (see new Extended Data Fig. 6). For a table of the genes and their fold changes see Supplementary Table 4. The other 49 hallmark datasets were less relevant to solve this question.

We now refer to this analysis on page 11, line 293.

6. It is not clear why the genetic analysis of the 200 genes was performed only on the duodenum as the stomach, and small intestine demonstrated mast cell hyperplasia. Due to the differences between the organs, it is essential to perform the analysis on each organ separately. In addition, a full bulk RNA-sequencing will be more informative regarding the changes in the tissue, if any. Due to the differences between the organs, it is essential to perform the analysis on each organ separately.

We thank the reviewer for the suggestion to analyze, in addition to the small intestine, also the stomach by RNA-sequencing. This experiment yielded interesting new data (Fig.3; Extended Data Fig. 4, 5, 6, 7; Supplementary Table 1-4).

These experiments are described in a new paragraph "Antigen avoidance behavior prevents immune activation and inflammation" in the main text page 11, line 293.

7. In the method- Immunization- the rationale for mixing the cytokines with their antibodies in one cocktail and injecting them into the mice is unclear.

We apologize for the lack of clarity. We provide this information in the Methods (page 19, line 532): "Mixing IL-3 with the anti-IL-3 antibody MP2-8F8, and IL-4 with the anti-IL-4 antibody 11B11 generates cytokine-antibody complexes that display increased activity in vivo¹, which we exploited here to increase the magnitude and duration of cytokine effects in vivo."

8. Extended figure 8D- why did you stain for IgE? The lines in the text which refer to this figure are 346-347, which refer to mast cells and enteroendocrine cell 5-HTR expression, and IgE is not a marker for either of these. Please add the correct staining.

As detailed under major point 5 above, we have no evidence for a role of serotonin. We have therefore omitted the previous Extended Data Figure 8D, E.

9. In extended figure 1, please add a legend for the grey and red marks.

We have added the missing legends to the graphs in Extended Data Fig. 1.

10. Extended figure 3K- please make a readable table of the 200 genes you chose for the genetic analysis, as it is hard to read them in the heatmap.

The former heatmap (Extended Data Fig. 3k) has been removed. Based on our new RNA Seq data (Fig.3; Extended Data Fig. 4, 5, 6, 7; Supplementary Table 1-4), we analyzed gene

expression of the 200 genes from Liberzon et al. 'Hallmark inflammatory response' gene list. Supplementary Table 4 contains this information in a readable form.

11. Extended figure 3J- Why did only the duodenum in the representative figure as the stomach and small intestine demonstrate the mast cell hyperplasia.?

We have removed the previous histological analyses in favor of the more comprehensive, sensitive and accurate tissue analysis by total lysate RNA Seq (Fig. 3; Extended Data Fig. 4, 5, 6, 7; Supplementary Table 1-4). In addition to the augmented mast cell gene signature (Fig. 3, d, f, g), we analyzed mast cell hyperplasia by flow cytometry (Fig. 1h-j; Extended Data Fig. 2e, f; Extended Data Fig. 3h, i).

12. Extended Figures 4C and 4D- Is there an increased level of Mast cells in the Cre/+ group in the passive sensitized group compared to the unsensitized groups?

We thank the reviewer for this question which prompted us to re-calculate intestinal mast cell numbers. We display numbers now as percentages which is the more robust analysis that bypasses poorly controllable factors such as the efficiency of tissue digestion. Of note, for the stomach this was not necessary because the release of mast cells from the epithelium is robust. Regarding the specific reviewer question, we found no significant differences in intraepithelial stomach or intestinal mast cells comparing in passive sensitized Cre/+ mice compared to the unsensitized (alum) Cre/+ mice (Cre/+ mice data are directly compared in Figure 7 for reviewer below; corresponding data in the paper are shown in Fig. 1h, i; Fig. 2h, i).

Fig. 7

Fig. 7: Mast cell levels do not differ in passively sensitized Cre/+ mice compared to unsensitized Cre/+ mice

a, b, Absolute numbers of stomach- (**a**) and percentages of small intestine intraepithelial mast cell (**b**) were compared between E-C1 sensitized, unsensitized (alum-immunized), and OVA-alum immunized mice. The bars represent mean values, and each dot a single mouse. Statistical analysis was performed using one-way ANOVA with Tukey multiple-comparison test. The exact P values are shown.

13. Figure 4C- indicate which color is what.

The graphs describing the average daily egg white water intake in the tested mouse mutants have been moved to Extended Data Fig. 8a-d, and all panels contain detailed group descriptions below the x-axis.

Reviewer Reports on the First Revision:

Referee #1 (Remarks to the Author):

To enhance the discussion section, we suggest that the authors consider incorporating additional research papers that could provide further insights into the topic.

I recommend the following papers for consideration:

"Aldehyde-driven transcriptional stress triggers an anorexic DNA damage response" (PMID: 34819667)

"GDF15 mediates the effects of metformin on body weight and energy balance" (PMID: 31875646)

"A regulatory T cell Notch4-GDF15 axis licenses tissue inflammation in asthma" (PMID: 32929274)

"An airway-to-brain sensory pathway mediates influenza-induced sickness" (PMID: 36890237)

"Quantitative Transcriptome Analysis of Purified Equine Mast Cells Identifies a Dominant Mucosal Mast Cell Population with Possible Inflammatory Functions in Airways of Asthmatic Horses" (PMID: 36430453)

"Mast cell-derived serotonin enhances methacholine-induced airway hyperresponsiveness in house dust mite-induced experimental asthma" (PMID: 33486786)

Furthermore, it will be good if the authors create a table or figure that highlights the differences between the Balb/c and C57BL6 models following OVA induction (L100-115).

Referee #2 (Remarks to the Author):

The authors' new data, mostly on the gene expression analyses of mast-cell-induced inflammatory responses, fail to alleviate this reviewer's original concerns: the manuscript is mostly descriptive of the known role of mast cells and does not provide a mechanistic understanding of how mast cells generate avoidance behavior.

The authors showed:

- Mast cells and IgE are required for animals to develop an avoidance of the allergen. This point has already been previously demonstrated by multiple other groups.
- Mast cell deficiency (irrespective of the presence of the allergen) itself does not generate behavioral abnormalities.
- The avoidance behavior does not require basophils.
- C57BL/6, in addition to BALB/c, mice can develop mast-cell-dependent avoidance behavior.
- Exposure to the allergen, either by the forced gavage of the allergen or the elimination of mast-cell dependent avoidance behavior, induces inflammatory responses.
- Mast cell response to the allergen is mostly seen in the stomach and the small intestine.
- Avoidance behavior may be partly mediated by FLAP, an enzyme involved in leukotrienes production. Authors use pharmacological means to block FLAP, but the effect is minimal, short-lived, and variable from experiment to experiment (Figure 4g,h). No data linking this gene function to mast cells or the brain are included.

- While the manuscript tried to characterize the role of mast cells, their function in relation to the brain has not been explored at all.

These are descriptive, mostly control experiments, or negative results (e.g., showing the lack of contribution from basophils or factors released by mast cells). The authors claim that they 'interrogate the link between intestinal mast cells and the brain using genetic, pharmacological and neurobiological approaches," which this reviewer failed to recognize in the manuscript.

Referee #3 (Remarks to the Author):

The authors have done a very good job with their revised manuscript. All the points in our initial review were satisfactorily addressed. The new data have further improved the study, particularly the identification of a role for inhibition of leukotriene synthesis in partially inhibiting manifestations of avoidance behavior. We think that this revised study importantly advance our understanding of the role of mast cells and IgE in the immune sensing that results in antigen avoidance.

It would be helpful for the authors to address the following minor points:

1. Lines 199-201: We suggest the following rewording, since only one method of monoclonal IgE transfer was tested: "Taken together, monoclonal IgE transfer can partially induce avoidance behavior, however, immunization **may be** required to attain full mast cell activation and avoidance."
2. Since both BALB/c and C57BL/6 mice were used, the authors should carefully report (in the main text and the figure legends), which strain was used for the results being discussed. For example, in line 252 (referring to Extended Data Fig. 4a), they should state whether BALB/c or C57BL/6 mice are being used. Also, in lines 262-263, and in the legends of Fig. 3, Extended Data Figs. 4-7 and 9.
3. In Fig. 4 legend, please define "EW" (presumably, egg white).
4. In line 299: add "nearly" so it reads: "...intestine, **nearly** all of which was prevented....".
5. The last sentence of the paragraph ending on lines 413-415 is not entirely clear.

Referee #4 (Remarks to the Author):

Plum et al should include their negative CNS data in the manuscript rather than just in the response

letter.

Author Rebuttals to First Revision:

Response to reviewers (reviewers' comments in italics)

Reviewer #1

1. To enhance the discussion section, we suggest that the authors consider incorporating additional research papers that could provide further insights into the topic.

We are grateful for pointing out these interesting references. We have now added one of these references (ref. 47) to the discussion:

'...which signals avoidance of the paired food to the PBN via GFRA1⁺ brainstem neurons (ref. 45-47)

2. Furthermore, it will be good if the authors create a table or figure that highlights the differences between the Balb/c and C57BL/6 models following OVA induction (L100-115).

We do agree that the differences in the experiments using the two strains are complex. Rather than comparing the data from the two strain in a new large table, we felt it more helpful for the reader to follow the request from Rev. #3 to 'carefully report (in the main text and in the figure legends) which strain was used for the results being discussed'. In the revised manuscript, we now state throughout the manuscript at the beginning of each paragraph whether BALB/c or C57BL/6 mice are being used.

Reviewer #3

1. Lines 199-201: We suggest the following rewording, since only one method of monoclonal IgE transfer was tested: "Taken together, monoclonal IgE transfer can partially induce avoidance behavior, however, immunization may be required to attain full mast cell activation and avoidance."

As suggested by the reviewer, we changed the following sentence to read:

'...may be required to attain full mast cell activation and avoidance'.

2. Since both BALB/c and C57BL/6 mice were used, the authors should carefully report (in the main text and the figure legends), which strain was used for the results being discussed. For example, in line 252 (referring to Extended Data Fig. 4a), they should state whether BALB/c or C57BL/6 mice are being used. Also, in lines 262-263, and in the legends of Fig. 3, Extended Data Figs. 4-7 and 9.

In the revised manuscript, we now state throughout the manuscript at the beginning of each paragraph whether BALB/c or C57BL/6 mice are being used.

3. In Fig. 4 legend, please define "EW" (presumably, egg white).

We thank the reviewer for pointing out this error. Mice received 25% OVA in water and not egg white solution (EW). We corrected this in Fig. 4.

4. In line 299: add "nearly" so it reads: "...intestine, nearly all of which was prevented....".

We changed this sentence to read:

'...uncovered broadly enhanced immune activation and inflammation in stomach and small intestine, nearly all of which was prevented by antigen avoidance behavior.'

5. The last sentence of the paragraph ending on lines 413-415 is not entirely clear.

We changed this sentence to read:

'The extent of immune gene induction was lower in non-avoiding mast cell-deficient mice compared to force-fed wild type mice, suggesting that immune activation is largely, but not completely, mast cell-dependent.'

Reviewer #4

Plum et al should include their negative CNS data in the manuscript rather than just in the response letter.

While we provided a comprehensive reply to the question of Fos expression in our previous response to reviewer #1, an integration of this data requires an appropriate explanation of the experiment and interpretation of the data. Given the space constraints, we suggest that the histology data, which likely reflected sugar consumption rather than avoidance in our model, shall be publicly available (previous point-by-point response to reviewer #1) within the transparent peer review process.